# An atlas of gene regulatory elements in adult mouse cerebrum

Yang Eric Li[1,12], Sebastian Preissl[2,12], Xiaomeng Hou[2], Ziyang Zhang[1], Kai Zhang[1], Yunjiang Qiu[1], Olivier B. Poirion[2], Bin Li[1], Joshua Chiou[3,4], Hanqing Liu[5], Antonio Pinto-Duarte[6], Naoki Kubo[1], Xiaoyu Yang[7], Rongxin Fang[1], Xinxin Wang[2], Jee Yun Han[2], Jacinta Lucero[6], Yiming Yan[1], Michael Miller[2], Samantha Kuan[1], David Gorkin[2], Kyle J. Gaulton[4], Yin Shen[7,8], Michael Nunn[5], Eran A. Mukamel[9], M. Margarita Behrens[6], Joseph R. Ecker[5,10] & Bing Ren[1,2,11 ✉]

The mammalian cerebrum performs high-level sensory perception, motor control and cognitive functions through highly specialized cortical and subcortical structures[1]. Recent surveys of mouse and human brains with single-cell transcriptomics[2–6] and high-throughput imaging technologies[7,8] have uncovered hundreds of neural cell types distributed in different brain regions, but the transcriptional regulatory programs that are responsible for the unique identity and function of each cell type remain unknown. Here we probe the accessible chromatin in more than 800,000 individual nuclei from 45 regions that span the adult mouse isocortex, olfactory bulb, hippocampus and cerebral nuclei, and use the resulting data to map the state of 491,818 candidate *cis*-regulatory DNA elements in 160 distinct cell types. We find high specificity of spatial distribution for not only excitatory neurons, but also most classes of inhibitory neurons and a subset of glial cell types. We characterize the gene regulatory sequences associated with the regional specificity within these cell types. We further link a considerable fraction of the *cis*-regulatory elements to putative target genes expressed in diverse cerebral cell types and predict transcriptional regulators that are involved in a broad spectrum of molecular and cellular pathways in different neuronal and glial cell populations. Our results provide a foundation for comprehensive analysis of gene regulatory programs of the mammalian brain and assist in the interpretation of noncoding risk variants associated with various neurological diseases and traits in humans.

The cerebral cortex and cerebral nuclei or basal ganglia in adult mice are made up of tens of millions of neurons and glial cells[9]. The neurons are classified into many different types of excitatory projection neurons and inhibitory interneurons, defined by their distinct morphology, neurotransmitters, and synaptic connections[10–12]. The glial cells include astrocytes, oligodendrocytes, oligodendrocyte precursor cells, microglia and other less abundant non-neuronal cell types. Understanding how the identity and function of each neural cell type is established during development and modified by experience is one of the fundamental challenges in brain research. Despite recent advances in analysis of gene expression patterns using single-cell transcriptomics and spatial transcriptomics assays[2–8], we still lack comprehensive maps of the transcriptional regulatory elements in each cell type. Transcriptional regulatory elements recruit DNA-binding transcription factors to exert control of target gene expression in *cis* in a cell-type-dependent manner[13,14]. Activation of these elements is accompanied by open chromatin,

specific histone modifications and DNA hypomethylation[13,14]. To exploit these epigenetic features, candidate *cis*-regulatory elements (cCREs) have been mapped by techniques such as DNase I hypersensitive sites sequencing (DNase-seq), assay for transposase-accessible chromatin using sequencing (ATAC-seq), chromatin immunoprecipitation followed by sequencing (ChIP–seq) and whole-genome bisulfite sequencing[15,16]. Conventional assays performed using bulk tissue samples are unable to resolve the cCREs in individual neural cell types owing to the extreme cellular heterogeneity of the brain. To overcome this limitation, single-cell genomic technologies have been developed to enable detection of open chromatin in individual cells[17–21] and identify cell-type-specific transcriptional regulatory sequences in several mouse brain regions[19,22–27].

As part of the BRAIN Initiative Cell Census Network (BICCN), we performed single-nucleus (sn)ATAC-seq assays using single-cell combinatorial indexing (sci)ATAC-seq[17,25] for more than 800,000 cells from

[1]Ludwig Institute for Cancer Research, La Jolla, CA, USA. [2]Department of Cellular and Molecular Medicine, Center for Epigenomics, University of California San Diego, School of Medicine, La Jolla, CA, USA. [3]Biomedical Sciences Graduate Program, University of California San Diego, La Jolla, CA, USA. [4]Department of Pediatrics, University of California San Diego, La Jolla, CA, USA. [5]Genomic Analysis Laboratory, The Salk Institute for Biological Studies, La Jolla, CA, USA. [6]Computational Neurobiology Laboratory, Salk Institute for Biological Studies, La Jolla, CA, USA. [7]Institute for Human Genetics, University of California San Francisco, San Francisco, CA, USA. [8]Department of Neurology, University of California San Francisco, San Francisco, CA, USA. [9]Department of Cognitive Science, University of California, San Diego, La Jolla, CA, USA. [10]Howard Hughes Medical Institute, The Salk Institute for Biological Studies, La Jolla, CA, USA. [11]Institute of Genomic Medicine, Moores Cancer Center, School of Medicine, University of California San Diego, La Jolla, CA, USA. [12]These authors contributed equally: Yang Eric Li, Sebastian Preissl. ✉e-mail: biren@health.ucsd.edu

45 dissected regions in the adult mouse brain to produce comprehensive maps of cCREs in distinct cerebral cell types. We also integrated the chromatin accessibility data with brain single-cell RNA sequencing (scRNA-seq) data to characterize the gene regulatory programs of different brain cell types, and used the cCREs to interpret genetic risk variants that are associated with neurological diseases and traits.

## snATAC-seq analysis of mouse brain cells

We dissected 45 brain regions from the isocortex, olfactory bulb (OLF), hippocampus (HIP) and cerebral nuclei (including the striatum and pallidum) in 8-week-old male mice (Fig. 1a, Extended Data Fig. 1, Supplementary Table 1). Each dissection was made from 600-µm-thick coronal brain slices starting at the frontal pole according to the Allen Brain Reference Atlas[28] (Extended Data Fig. 1). For each region, we performed snATAC-seq with two independent biological replicates (Fig. 1a, Extended Data Fig. 2a–f). A total of 813,799 nuclei with a median number of 4,929 fragments per nucleus passed rigorous quality control measures (Supplementary Table 2, Extended Data Fig. 2g–k, Supplementary Note, Methods). Among them, 381,471 nuclei were from the isocortex, 123,434 were from the olfactory area, 147,338 were from cerebral nuclei and 161,556 were from the hippocampus (Fig. 1a, Extended Data Fig. 2l).

We performed iterative clustering with the software package SnapA-TAC[26] and classified the 813,799 nuclei into distinct cell groups (Fig. 1b–e, Extended Data Figs. 3–5, Supplementary Tables 2, 3, Supplementary Note, Methods). First, we classified the cells into glutamatergic neurons (387,060 nuclei, 47.6%), GABAergic (γ-aminobutyric acid-producing) neurons (167,181 nuclei, 20.5%) and non-neuronal cells (259,588 nuclei, 31.9%) (Fig. 1b–d). Next, the three cell classes were further divided into 43 subclasses (also referred to as major types in the accompanying paper[29]) (Fig. 1b, d) and annotated these on the basis of chromatin accessibility at promoters and the gene bodies of at least three marker genes of known neural cell types[2,4] (Fig. 1e, Extended Data Fig. 6, Supplementary Table 4). Finally, we performed another round of clustering for each subclass and identified a total of 160 cell types at optimal resolutions (also referred to as subtypes in the accompanying paper[29]) (Extended Data Figs. 3b–g, 7, Supplementary Table 3, Supplementary Note, Methods). For example, *Lamp5*+ neurons (LAMGA) and *Sst*+ neurons (SSTGA)[4,6] were further divided into several cell types, including a chandelier-like cell type[4] and the *Chodl* cell type[6] (Fig. 1b, e). Notably, the detected clusters and cell type proportions from snATAC-seq were comparable between the combinatorial barcoding (sci) and the droplet-based 10x Genomics platform[20] (Extended Data Fig. 8, Supplementary Note).

We constructed a dendrogram to capture the hierarchical organization of chromatin landscapes among the 43 subclasses (Fig. 1d, Extended Data Fig. 9). This dendrogram shows known organizing principles of mammalian brain cells: the non-neuronal class is separated from the neuronal class, which is further separated on the basis of the neurotransmitter types (GABAergic versus glutamatergic) and developmental origins[4] (Fig. 1d). These chromatin-defined cell types matched well with the taxonomy based on transcriptomes[2] and DNA methylomes[29] (Extended Data Fig. 10, Supplementary Note, Supplementary Table 5).

## Regional specificity of brain cell types

Taking advantage of our high-resolution brain dissections, we examined the regional specificity of each brain cell type (Extended Data Figs. 11, 12). We calculated a regional specificity score for each subclass and cell type based on the contribution from different brain regions and showed that this score is highly consistent between biological replicates (Fig. 1f, g, Extended Data Fig. 12d, Methods). Overall, we found good agreement between the regional specificity of most neuronal cell types defined using snATAC-seq datasets and the normalized in situ hybridization (ISH) signals of marker genes in each cell type (Extended Data Fig. 13, Supplementary Table 6, Methods). As expected, most glial cell types were ubiquitously distributed throughout the different brain dissections and showed very low regional specificity (Fig. 1f), except for neuronal intermediate progenitor cells (NIPCs) and radial glia-like cells in the dentate gyrus or subventricular zone (labelled as subclass RGL in Fig. 1f). In contrast to the glial cell types, most GABAergic and glutamatergic neurons showed notable regional specificity (Fig. 1f, g). We found a marked separation on the basis of brain subregions for distinct neuron types such as granule cells in the dentate gyrus and matrix D1 neurons (MXD) in the pallidum. Glutamatergic neurons showed slightly higher regional specificity than GABAergic neurons, consistent with previous single-cell transcriptomic analysis[4] (Fig. 1g, bottom). We also observed distinct types of *Pvalb*+ neuron (PVGA), intra-telencephalic-projection neuron, and hippocampal cornu ammonis (CA1) neuron (CA1GL) that were highly restricted to individual brain regions or dissections (Extended Data Fig. 14, Supplementary Note).

## Mapping of cCREs in mouse brain cells

To delineate the gene regulatory programs that underlie the identity and function of each brain cell type, we next identified cCREs in each of the 160 brain cell types from the accessible chromatin landscapes. To account for different sequencing depth and/or the number of nuclei in individual clusters, we identified reproducible peaks based on a corrected integer score calculated by model-based analysis of ChIP–seq data (MACS2)[30] (Extended Data Fig. 15a–c, Supplementary Note, Methods). We further selected the elements that were determined as open chromatin regions in a significant fraction of cells in each subtype (false discovery rate (FDR) < 0.01, zero-inflated beta model) (Extended Data Fig. 15d, Supplementary Note, Methods), resulting in a union of 491,818 open chromatin regions. These cCREs together made up 14.8% of the mouse genome (Supplementary Tables 7, 8). Of these cCREs, 96.3% were located at least 1 kb away from annotated promoter regions of protein-coding and long noncoding RNA genes (Fig. 2a, Extended Data Fig. 15e). Several lines of evidence support the authenticity of the identified cCREs. First, they strongly (70.1%) overlapped with the DNase hypersensitive sites (DHSs) that were previously mapped in a broad spectrum of bulk mouse tissues and developmental stages[14] (Fig. 2b, Extended Data Fig. 15f). Second, they generally showed higher levels of sequence conservation than random shuffled genomic regions with similar GC content (Fig. 2c, Extended Data Fig. 15g). Third, they were enriched for active chromatin states or potential insulator protein-binding sites that were previously mapped by bulk analysis of mouse brain tissues[31–33] (Fig. 2d, Extended Data Fig. 15h).

To define the cell type specificity of the cCREs, we first plotted the median levels of chromatin accessibility against the range of variation for each element (Fig. 2e). We found that most cCREs exhibited highly variable chromatin accessibility across the brain cell types identified, except for 8,188 regions that showed accessible chromatin in almost all cell clusters (Fig. 2e). The invariant cCREs were highly enriched for promoters (81%), with the remainder including CCCTC-binding factor (CTCF)-binding sites (9%) and strong enhancers (Fig. 2f). To characterize the cell type specificity of the cCREs more explicitly, we used non-negative matrix factorization to group them into 42 modules, with elements in each module sharing similar cell type specificity profiles. Aside from the first module (M1) that included mostly cell type invariant cCREs, the remaining 41 modules showed high cell-type-restricted accessibility (Fig. 2g, Supplementary Tables 9, 10). These cell-type-restricted modules were enriched for distinct sets of motifs recognized by known transcriptional regulators (Supplementary Table 11), laying a foundation for investigating the gene regulatory programs in different brain cell types and regions.

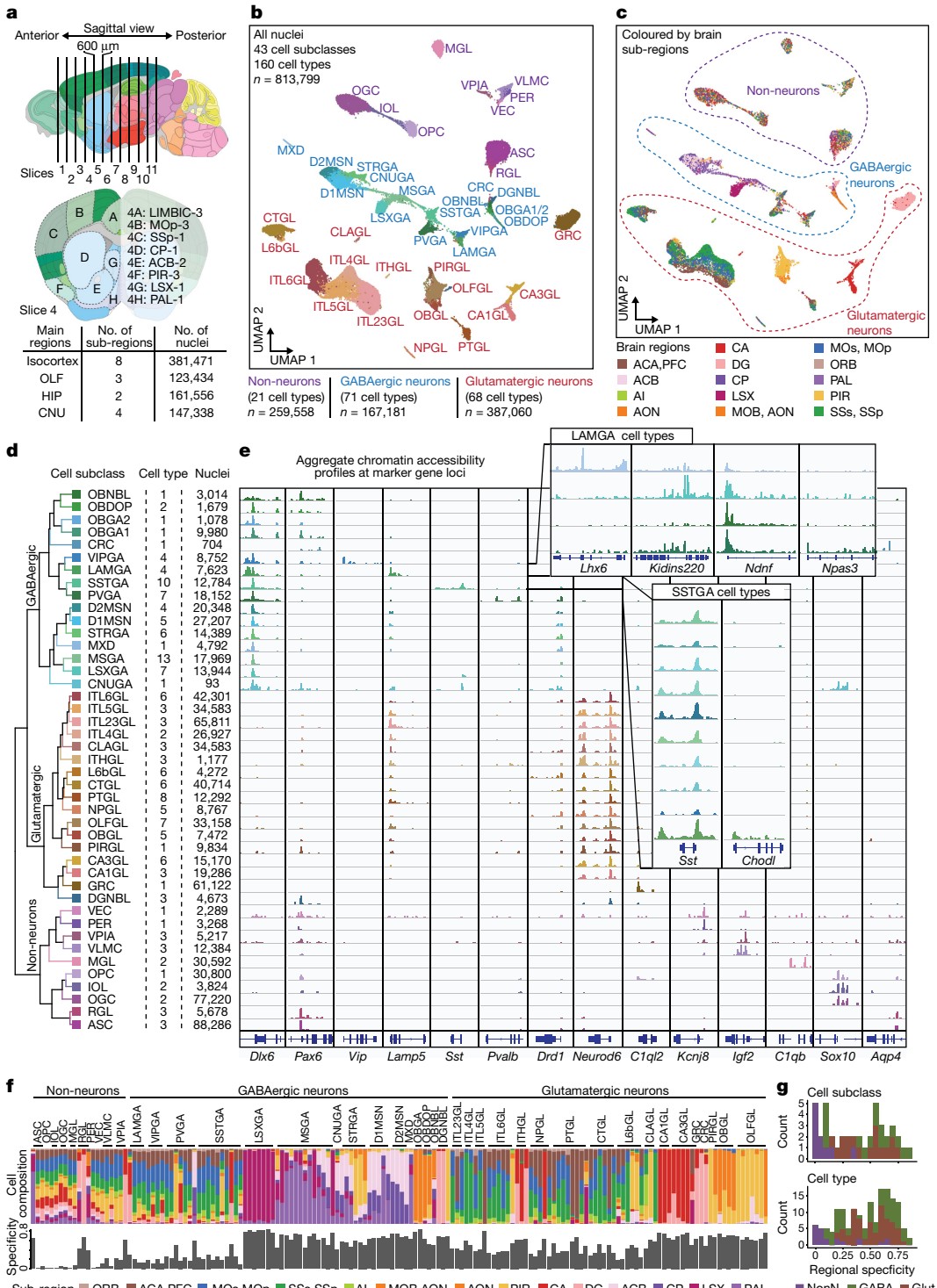

**Fig. 1 | Single-cell analysis of chromatin accessibility in the adult mouse cerebrum. a**, Schematic of sample dissection strategy. A detailed list of regions is in Supplementary Table 1. **b**, Uniform manifold approximation and projection (UMAP)[58] embedding and clustering analysis of snATAC-seq data. A full list and description of cluster labels are in Supplementary Tables 2, 3. **c**, UMAP embedding and analysis as in **b** but coloured by subregions. Dotted lines demark major cell classes. ACA, anterior cingulate area; ACB, nucleus accumbens; AI, agranular insular area; AON, anterior olfactory nucleus; CA, cornus ammonis; CP, caudoputamen; DG, dentate gyrus; LSX, lateral septal nucleus; MOB, main olfactory bulb; MOs, secondary motor area; MOp, primary motor area; ORB, orbital area; PAL, pallidum; PFC, prefrontal cortex; PIR,

piriform area; SSs, secondary somatosensory area; SSp, primary somatosensory area. **d**, Left, hierarchical organization of subclasses on chromatin accessibility. Middle, each subclass represents 1–13 cell types. Right, the number of nuclei in each subclass. **e**, Genome browser tracks of aggregate chromatin accessibility profiles for each subclass at selected marker gene loci that were used for cell cluster annotation. A full list and description of subclass annotations are in Supplementary Table 4. **f**, Bar chart representing the relative contribution of subregions to 160 cell types. **g**, Stacked histograms showing the regional specificity scores of subclasses and cell types of GAGAergic neurons (GABA), glutamatergic neurons (Glu) and non-neurons (NonN).

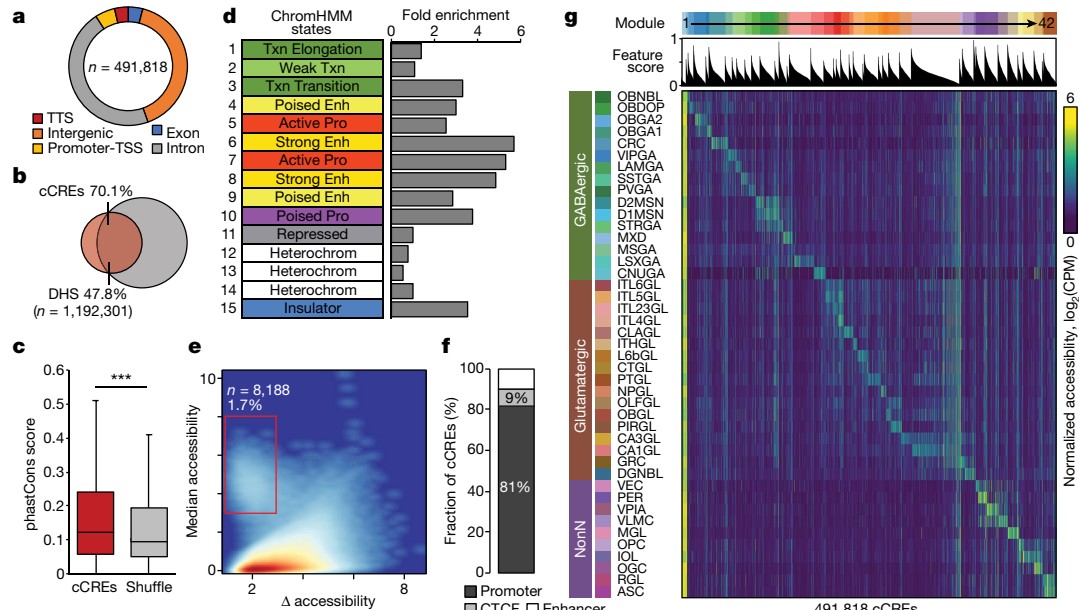

**Fig. 2 | Identification and characterization of candidate CREs across mouse cerebral cell types. a**, Pie chart showing the fraction of cCREs that overlaps with different classes of annotated sequences in the mouse genome. TSS, transcription start site; TTS, transcription termination site. **b**, Venn diagram showing the overlap between cCREs defined in this study (in red) and the DHSs in the SCREEN database (in grey)[14]. **c**, Box plots showing sequence conservation measured by PhastCons score[59]. ***$P < 0.001$, Wilcoxon rank-sum test. Boxes span the first to third quartiles (Q1 to Q3), horizontal line denotes the median, and whiskers show 1.5× the interquartile range (IQR). **d**, Bar chart showing the fold enrichment of cCREs within the different states of mouse brain

chromatin[31] annotated with a 15-state ChromHMM model[33]. **e**, Density map comparing the median and maximum variation of chromatin accessibility at each cCRE across cell types. Each dot represents a cCRE. Red box highlights elements with low chromatin accessibility variability across clusters. **f**, Stacked bar plot showing the fraction of cCREs with low variability overlapping with distinct genomic features. **g**, Heat map showing association of the 43 subclasses (rows) with 42 *cis*-regulatory modules (top, from left to right). Columns represent cCREs. A full list of subclass or module associations is in Supplementary Table 9, and the association of cCREs to modules is in Supplementary Table 10. CPM, counts per million.

## Differential chromatin states at cCREs

Because most neuronal types showed highly restricted distribution in the mouse cerebral cortex and basal ganglia, we hypothesized that the regional specificity of different cell types is accompanied by differences in chromatin accessibility at the cCREs, which drive cell-specific gene expression patterns. We performed integrative analysis to delineate these differentially accessible cCREs among different neuronal and glial cell types. We compared the open chromatin landscapes among different cell types using a likelihood ratio test (Methods), and identified a median number of 11,683 cCREs that exhibited differential accessibility (range: 360–31,608) (Extended Data Fig. 16a, b). We characterized the most diverse GABAergic neuron types in the medial septal nucleus (MSGA) (Extended Data Fig. 17, Supplementary Table 12, Supplementary Note). For SSTGA, we detected a total of 50,079 cCREs, 98% of which were promoter distal, that exhibited cell-type-restricted accessibility within the subclass (Figs. 1, 3a, b, Extended Data Fig. 17a–c, Supplementary Table 13). We found a strong motif enrichment of the zinc-finger transcription factor family KLF in cCREs in SSTGA10 cells (also known as Sst-*Chodl* cells) compared with other SST neurons (Fig. 3c). This observation, coupled with the finding that *Klf5*, a member of the KLF family, was expressed in *Chodl* cells, implicates KLF5 in the transcriptional control of *Chodl* cells (Fig. 3d).

We also identified three astrocyte cell types and performed differential chromatin accessibility analysis for cCREs between these (Fig. 3e, Extended Data Fig. 18a). Two cell types were predominantly found in the cortex and hippocampus, whereas the third cell type (ASCN) was detected mostly in the pallidum and lateral septum complex[34] (Fig. 3f). The cortical or hippocampal astrocyte cell types resembled previously defined fibrous astrocytes in white matter (ASCW) and protoplasmic astrocytes in grey matter (ASCG)[5,35] (Extended Data Fig. 18b–d). Consistent with the previous findings that astrocytes were organized into

distinct lineage-associated laminae[36], we detected a spatial gradient in ASCG based on chromatin accessibility at several gene loci in ASCG (Extended Data Fig. 18e). We further performed motif analysis for differentially accessible regions in the ASCN cell type, finding enrichment of the binding motif for the GLI family of zinc-finger transcription factors (Fig. 3g, h, Extended Data Fig. 18f, Supplementary Table 14), which mediate the sonic hedgehog (Shh) signalling pathway that maintains neural stem-cell and astrocyte functions[36]. Notably, we found a cCRE that contained the GLI motif upstream of the *Olig2* promoter. This is consistent with a potential role for Shh signalling in regulating *Olig2* expression (Fig. 3i, Extended Data Fig. 18g, h) in OLIG2-lineage-derived mature astrocytes in the globus pallidus[34]. We found that a high fraction of genes specific to OLIG2-lineage astrocytes were predominantly expressed in the pallidum (Fig. 3j). For example, *Itih3*, *Slc6a11*[34] and *Agt* were predominantly expressed in the pallidum, and cCREs at the gene locus were specifically accessible in ASCNs (Fig. 3k, l, Extended Data Fig. 18h–j). We also found enrichment of distinct transcription factor motifs from regional-specific cCREs in ASCGs sampled from different brain regions (Extended Data Fig. 18k, l, Supplementary Note, Supplementary Table 15).

## Integrative analysis of gene regulation

To investigate the transcriptional regulatory programs that are responsible for cell-type-specific gene expression patterns in the mouse cerebrum, we carried out integrative analysis that combines the snATAC-seq data collected in the current study with previously published scRNA-seq data from matched brain regions[2]. We first connected 261,204 distal cCREs to 12,722 putative target genes by measuring the co-accessibility using Cicero[37,38] (Fig. 4a, Methods). This analysis identified a total of 813,638 gene–cCRE pairs within 500 kb of each other (Supplementary Table 16). Next, we identified the subset of cCREs that might increase the

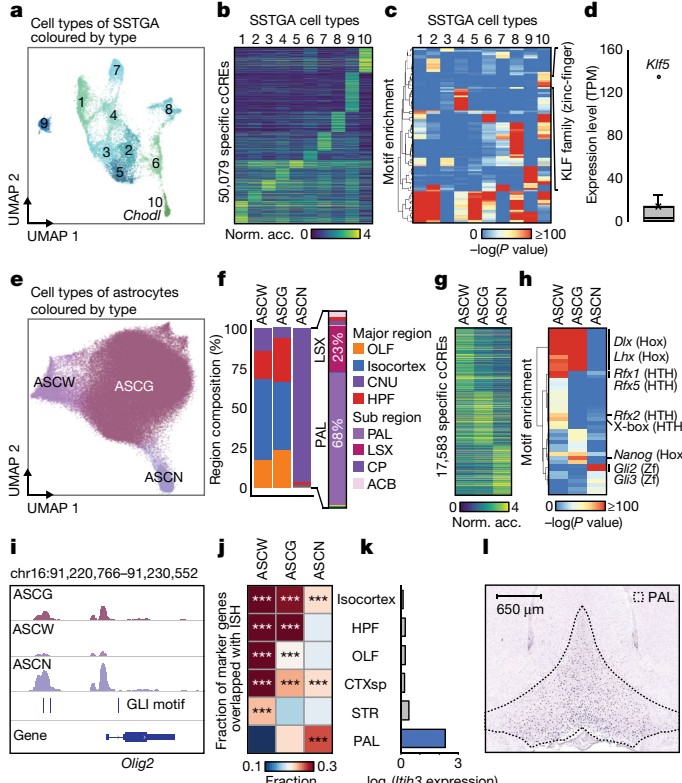

**Fig. 3 | Regional specificity of cell types correlates with chromatin accessibility at cCREs. a**, UMAP[58] embedding of SSTGA cell types. **b**, Normalized chromatin accessibility of 50,079 cell-type-specific cCREs. **c**, Motif enrichment analysis for cell-type-specific cCREs. **d**, Expression level of genes encoding members of the KLF transcription factor family in SSTGA10. Expression values of Sst-*Chodl* cells were extracted from the Allen Brain Atlas (atlas.brain-map.org)[44]. **e**, UMAP[58] embedding of astrocyte cell types. **f**, Contribution of brain regions to each astrocyte cell type. CNU, cerebral nuclei; OLF, olfactory area; HPF, hippocampus. **g**, Normalized accessibility of astrocyte cell-type-specific cCREs. **h**, Motif enrichment in astrocyte cell-type-specific cCREs. Hox, homeobox; HTH, helix-turn-helix; Zf, zinc-finger. **i**, Genome browser view[60] showing ASCN-specific cCREs at the *Olig2* locus. **j**, Heat map showing the fraction of overlap between spatially mapped genes from ISH in different brain structures and genes specific to astrocyte cell types. ***P < 0.0001, Fisher's exact test. **k**, Bar plot showing log2-transformed expression value of *Itih3* in each brain region from ISH experiments. CTXsp, cortical subplate. Images downloaded from © 2004 Allen Institute for Brain Science. Allen Mouse Brain Atlas. Available from: atlas. brain-map.org. **l**, Views of ISH experiment from the Allen Brain Atlas, showing spatial expression of *Itih3* in the pallidum.

expression of putative target genes and therefore function as putative enhancers in neuronal or non-neuronal types. To this end, we first identified distal cCREs for which chromatin accessibility was correlated with transcriptional variation of the linked genes in the RNA–ATAC joint cell clusters as defined above (Fig. 4a, b, Extended Data Fig. 10). This analysis revealed a total of 129,404 pairs of positively correlated cCRE (putative enhancers) and genes at an empirically defined significance threshold of FDR < 0.01 (Supplementary Table 16). These included 86,850 putative enhancers and 10,604 genes (Fig. 4b, Extended Data Fig. 19, Supplementary Note). To investigate how the putative enhancers may direct cell-type-specific gene expression, we further classified them into 38 modules, by applying non-negative matrix factorization to the matrix of normalized chromatin accessibility across the RNA–ATAC joint cell clusters (Supplementary Table 17). The putative enhancers in each module had a similar pattern of chromatin accessibility across cell clusters to the expression of putative target genes (Fig. 4c, Supplementary Table

18). This analysis revealed a large group of 12,740 putative enhancers that were linked to 6,373 genes expressed at a higher level in all neuronal cell clusters than in all non-neuronal cell types (module M1) (Fig. 4c). It also uncovered modules of enhancer–gene pairs that were active in a more restricted manner (modules M2–M38) (Fig. 4c, Extended Data Fig. 19, Supplementary Tables 19, 20, Supplementary Note).

Genes associated with module M1 are preferentially expressed in both glutamatergic and GABAergic neurons, but not in glial cell types (Fig. 4c). De novo motif enrichment analysis of the 12,740 cCREs or putative enhancers in this module showed strong enrichment of sequence motifs recognized by the transcription factors CTCF, RFX and MEF2 (Supplementary Table 21, Extended Data Fig. 19d), as well as many known motifs for other transcription factors (Fig. 4d, Supplementary Table 20). CTCF is a ubiquitously expressed DNA-binding protein with a well-established role in transcriptional insulation and chromatin organization[39]. CTCF has also been shown to promote neurogenesis by binding to promoters and enhancers of the proto-cadherin alpha gene cluster and facilitating enhancer–promoter contacts[40,41]. We found putative enhancers with one or more CTCF-binding motifs linked to 2,601 genes that were broadly expressed in both inhibitory and excitatory neurons (Fig. 4d, Extended Data Fig. 19e), the gene products of which are involved in several neural processes including axon guidance and synaptic transmission (Extended Data Figs. 19f, 20, Supplementary Tables 22, 23, Supplementary Note).

## Neurogenesis in the adult mouse brain

Neurogenesis in the adult mouse brain is spatially restricted to the subgranular zone (SGZ) in the dentate gyrus of the hippocampus where excitatory neurons are generated, and the subventricular zone (SVZ) of the lateral ventricles that give rise to GABAergic neurons[42]. The NIPCs that are involved in adult neurogenesis[42,43] could be identified as the cells lined up in trajectories between radial glia-like cells and neuroblasts in both brain regions (Fig. 4e) and their presence in the respective dissections was supported by ISH data from the Allen Brain Atlas[44] for several marker genes (Extended Data Fig. 21a, b). We predicted potential transcription factors that contribute to NIPCs as well as other cell types by integrating RNA expression and motif enrichment analysis using the Taiji pipeline[45] (Fig. 4f, Supplementary Table 24, Methods). Consistent with previous reports, NR2E1 was predicted to be a master regulator in both NIPC populations[46], and SOX2 was a regulator of the NIPCs from the SGZ[42], whereas E2F1 contributed to NIPCs from the SVZ[47]. Although chromatin landscapes in the NIPCs from both regions were very similar (Fig. 4g), we identified 200 differentially accessible regions in the NIPC population between the SGZ and the SVZ (Fig. 4h). Several cCREs at *Neurog2*, which encodes a protein that is crucial for glutamatergic granule neuron specification in the SGZ, were found to be accessible selectively in the SGZ but not in the SVZ[43] (Fig. 4i). By contrast, several cCREs with chromatin accessibility in SVZ NIPCs were located at the *Dlx2* locus—a gene that is important for the specification of GABAergic neurons[43,46] (Extended Data Fig. 21c). An active enhancer previously validated by mouse transgenics[48] was predicted to target the nearby *Trappc9* gene, which encodes a protein that is involved in nerve growth factor-induced neuronal differentiation[49] (Fig. 4i, j). These observations suggest that NIPCs in the SGZ and SVZ give rise to distinct neuronal cell types by engaging different cCREs involved in controlling region-specific gene expression of key regulator genes.

## Interpreting noncoding risk variants

Genome-wide association studies (GWASs) have identified genetic variants that are associated with many neurological diseases and traits (Supplementary Table 25), but interpreting the results has been challenging because most variants are located in noncoding parts of the genome that often lack functional annotations[50]. To test whether our maps of

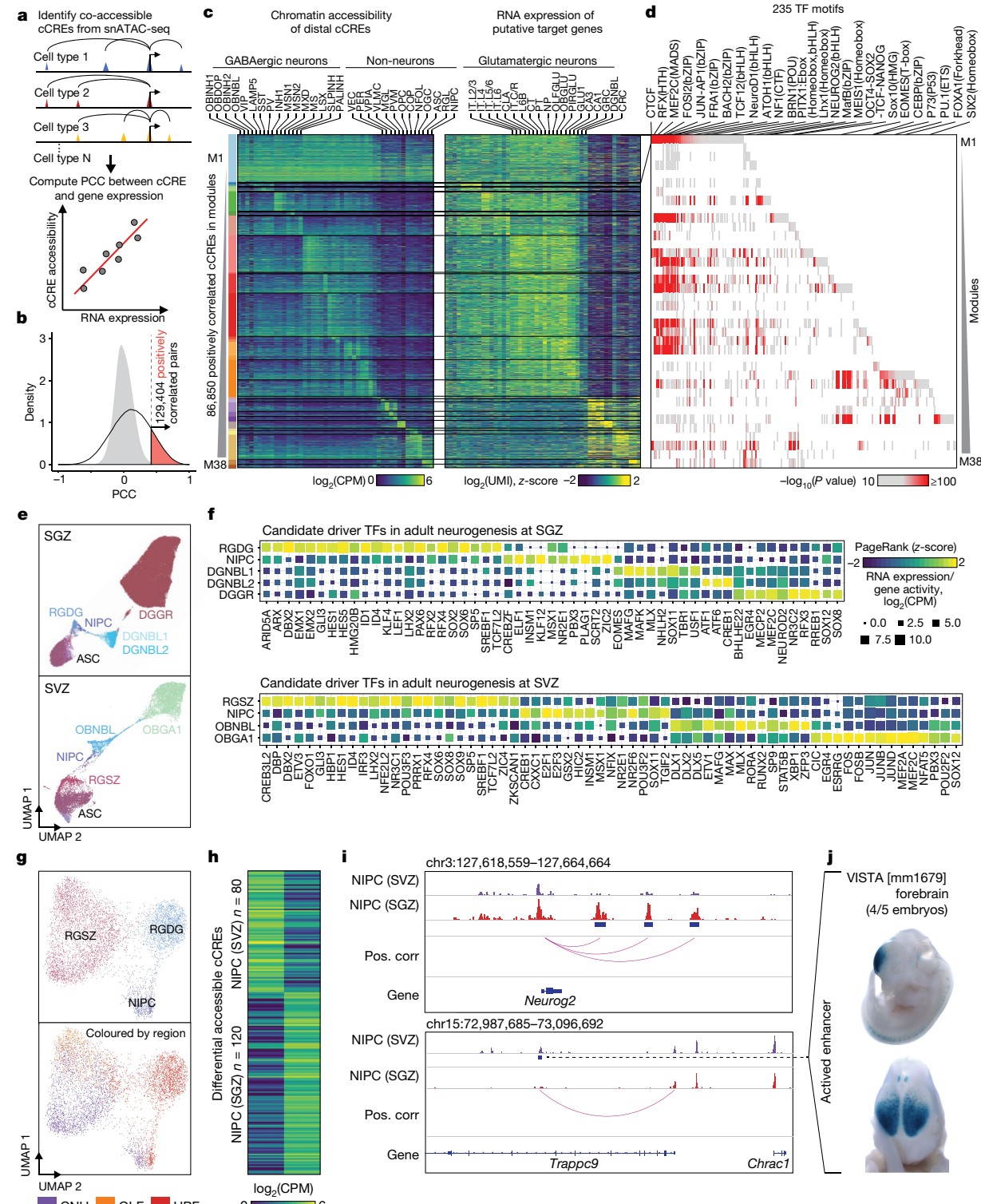

**Fig. 4 | Integrative analysis of gene regulatory programs in different cerebral cell types. a**, Schematic overview of the computational strategy used to identify cCREs that are positively correlated with transcription of target genes. **b**, In total, 129,404 pairs of positively correlated cCRE and genes (highlighted in red) were identified (FDR < 0.01). Grey filled curve shows distribution of Pearson's correlation coefficient (PCC) for randomly shuffled cCRE–gene pairs. **c**, Heat map showing chromatin accessibility of putative enhancers (left) and expression of linked genes (right). Genes are shown for each putative enhancer separately. UMI, unique molecular identifier. **d**, Enrichment of known transcription factor (TF) motifs in distinct enhancer–gene modules. Known motifs from HOMER[61] with enrichment $P$ value < $10^{-10}$ are

shown. **e**, UMAP[58] embedding of cell types involved in adult neurogenesis at the SGZ (top) and SVZ (bottom). **f**, Predicted transcription factors in different cell types involved in neurogenesis in the SGZ and SVZ. **g**, UMAP[58] embedding of NIPCs and radial glia-like cells coloured by cell type (top) and brain region (bottom). **h**, Heat map showing the differential cCREs of NIPCs between the SGZ and SVZ. **i**, Genome browser tracks[60] showing representative differential cCREs of NIPCs in the SGZ (top) and SVZ (bottom). **j**, Representative images of transgenic mouse embryos showing LacZ reporter gene expression under the control of the indicated enhancers that overlapped the differential cCRE in **i** (dotted line). Images were downloaded from the VISTA database (https://enhancer.lbl.gov)[48].

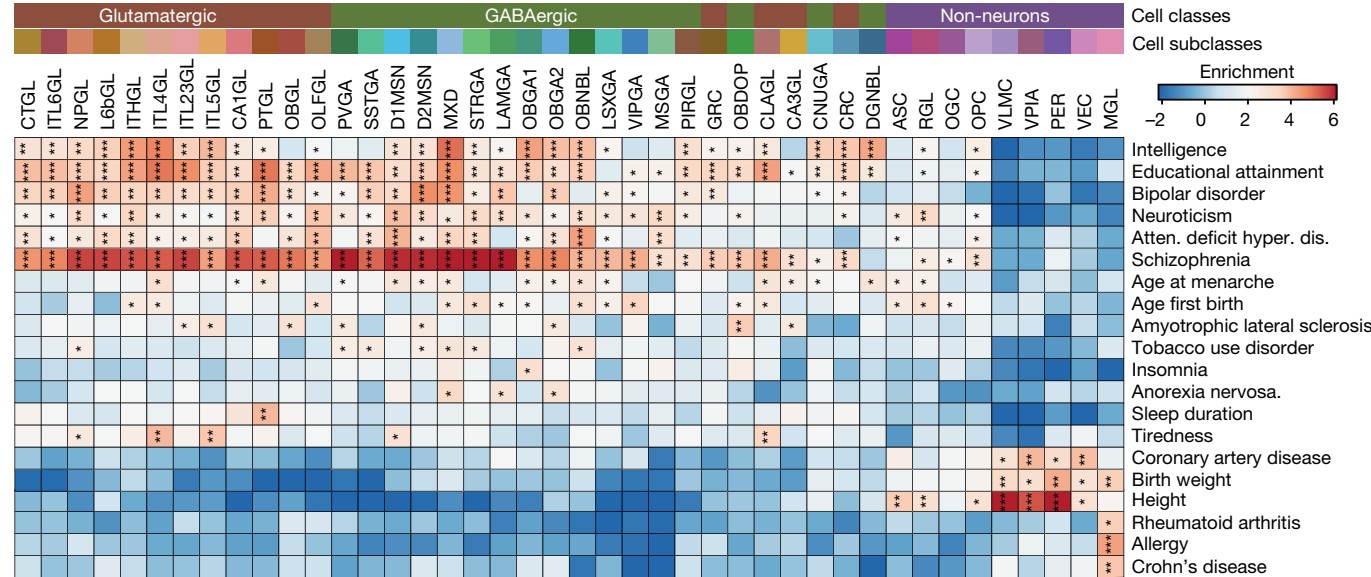

**Fig. 5 | Human orthologues of cerebral cCREs are enriched for noncoding risk variants for neurological diseases and traits in a cell-type-restricted manner.** Heat map showing the results of linkage disequilibrium score regression[52] analysis of the noncoding variants associated with the indicated traits or diseases in the human orthologues of cCREs identified from 43 subclasses of mouse cerebral cell. *FDR < 0.05, **FDR < 0.01, ***FDR < 0.001. Displayed are all the 43 subclasses and traits or diseases with at least one significant association by linkage disequilibrium score regression analysis (FDR < 0.05).

cCREs in different mouse brain cell types could assist the interpretation of noncoding risk variants of neurological diseases, we identified orthologues of the mouse cCREs in the human genome by performing reciprocal homology searches[51] (Methods). For this analysis, we found that for 69.2% of the mouse brain cCREs, human genome sequences with high similarity could be identified (more than 50% of bases lifted over to the human genome) (Extended Data Fig. 22a). Supporting the function of these human orthologues of the mouse brain cCREs, 83.0% of them overlapped with representative DNase hypersensitivity sites in the human genome[14].

We performed linkage disequilibrium score regression[52] analysis and found significant associations between 20 neurological traits (Supplementary Table 25) and the open chromatin landscapes in one or more subclasses of the brain cells we identified (Fig. 5, Methods, Supplementary Note). In particular, we observed widespread and strong enrichment of genetic variants linked to psychiatric and cognitive traits such as major depressive disorder, bipolar disorder and schizophrenia (SCZ) within accessible cCREs across various neuronal cell types (Fig. 5). Other neurological traits—such as attention deficit hyperactivity disorder and autism spectrum disorder—were associated with specific neuronal cell types in cerebral nuclei and the hippocampus (Fig. 5). Risk variants for schizophrenia were not only enriched in cCREs in all excitatory neurons, but also enriched in certain inhibitory neuron cell types[53] (Fig. 5). We also found that more than 25% of homologous sequences of SCZ causal variants reside in the mouse cCREs defined in this study (Extended Data Fig. 22b, Supplementary Note). The strongest enrichment of heritability for bipolar disorder was found in cCREs that mapped in the excitatory neurons from the isocortex (Fig. 5). Risk variants of tobacco use disorder showed significant enrichment in the cell types from the striatum—a cerebral nucleus previously implicated in addiction[54] (Fig. 5).

Understanding the cellular and molecular basis of brain circuits is one of the grand challenges of the twenty-first century[55,56]. In-depth knowledge of the transcriptional regulatory program in brain cells would not only improve our understanding of the molecular inner workings of neurons and non-neuronal cells, but could also shed light on the pathogenesis of a spectrum of neuropsychiatric diseases[57]. Here, we report a comprehensive profiling of chromatin accessibility at single-cell resolution in the mouse cerebrum. The chromatin accessibility maps of 491,818 cCREs, probed in 813,799 nuclei and 160 cell types, span several cerebral cortical areas and subcortical structures. Taking advantage of our high-resolution brain dissections, we examined the regional specificity in chromatin accessibility of cell types in the mouse cerebrum and showed that most brain cell types exhibit strong regional specificity. The described cCRE atlas (http://catlas.org/mousebrain) represents a rich resource for the neuroscience community to understand the molecular patterns that underlie diversification of brain cell types in complementation to other molecular and anatomical data.

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

# Methods

## Tissue preparation and nuclei isolation

All experimental procedures using live animals were approved by the SALK Institute Animal Care and Use Committee under protocol number 18-00006. Adult C57BL/6J male mice were purchased from Jackson Laboratories. Brains were extracted from 56–63-day-old mice and sectioned into 600-μm coronal sections along the anterior-posterior axis in ice-cold dissection media[62,63], by using a brain slicer from CNS muSlice LLC. Specific brain regions were dissected according to the Allen Brain Reference Atlas[28] (Extended Data Fig. 1) and nuclei isolated as previously described[63]. Regions were pooled from 3–31 of the same sex to obtain enough nuclei for snATAC-seq for each biological replica, and two biological replicas were processed for each dissection region.

## snATAC-seq using combinatorial indexing

snATAC-seq was performed as described with steps optimized for automation[25,26]. A step-by-step-protocol for library preparation is available at: https://www.protocols.io/view/sequencing-open-chromatin-of-single-cell-nuclei-sn-pjudknw/abstract.

Brain nuclei were pelleted with a swinging bucket centrifuge (500*g*, 5 min, 4 °C; 5920R, Eppendorf). Nuclei pellets were resuspended in 1 ml nuclei permeabilization buffer (5% BSA, 0.2% IGEPAL-CA630, 1 mM DTT and cOmpleteTM, EDTA-free protease inhibitor cocktail (Roche) in PBS) and pelleted again (500*g*, 5 min, 4 °C; 5920R, Eppendorf). Nuclei were resuspended in 500 μl high-salt tagmentation buffer (36.3 mM Tris-acetate (pH 7.8), 72.6 mM potassium-acetate, 11 mM Mg-acetate, 17.6% dimethylformamide) and counted using a haemocytometer. Concentration was adjusted to 1,000–4,500 nuclei per 9 μl, and 1,000–4,500 nuclei were dispensed into each well of a 96-well plate. For tagmentation, 1 μl barcoded Tn5 transposomes[26] were added using a BenchSmart 96 (Mettler Toledo, RRID:SCR_018093) (Supplementary Table 26), mixed five times, and incubated for 60 min at 37 °C with shaking (500 rpm). To inhibit the Tn5 reaction, 10 μl of 40 mM EDTA was added to each well with a BenchSmart 96 (Mettler Toledo, RRID:SCR_018093) and the plate was incubated at 37 °C for 15 min with shaking (500 rpm). Next, 20 μl 2× sort buffer (2% BSA, 2 mM EDTA in PBS) were added using a BenchSmart 96 (Mettler Toledo, RRID:SCR_018093). All wells were combined into a FACS tube and stained with 3 μM Draq7 (Cell Signaling) (Extended Data Fig. 23). Using a SH800 (Sony), 20 nuclei were sorted per well into eight 96-well plates (total of 768 wells, 30,720 nuclei total, 15,360 nuclei per sample) containing 10.5 μl EB (25 pmol primer i7, 25 pmol primer i5, 200 ng BSA (Sigma). If processing two samples per day, tagmentation was performed with different sets of barcodes in separate 96 well plates. After tagmentation nuclei from individual plates were pooled together. Preparation of sort plates and all downstream pipetting steps were performed on a Biomek i7 Automated Workstation (Beckman Coulter, RRID:SCR_018094). After the addition of 1 μl 0.2% SDS, samples were incubated at 55 °C for 7 min with shaking (500 rpm). Then, 1 μl 12.5% Triton X-100 was added to each well to quench the SDS. Next, 12.5 μl NEBNext High-Fidelity 2 × PCR Master Mix (NEB) were added and samples were amplified by PCR (72 °C 5 min, 98 °C 30 s, (98 °C 10 s, 63 °C 30 s, 72 °C 60 s) × 12 cycles, held at 12 °C). After PCR, all wells were combined. Libraries were purified according to the MinElute PCR Purification Kit manual (Qiagen) using a vacuum manifold (QIAvac 24 plus, Qiagen) and size selection was performed with SPRI Beads (Beckmann Coulter, 0.55x and 1.5x). Libraries were purified one more time with SPRI Beads (Beckmann Coulter, 1.5x). Libraries were quantified using a Qubit fluorimeter (Life Technologies, RRID:SCR_018095) and the nucleosomal pattern was verified using a Tapestation (High Sensitivity D1000, Agilent). Libraries generated with indexing version 1 (Supplementary Table 1) were sequenced on a HiSeq2500 sequencer (RRID:SCR_016383, Illumina) using custom sequencing primers, 25% spike-in library and following read lengths: 50 + 43 + 37 + 50 (Read1 + Index1 + Index2 + Read2). Libraries generated

with indexing version 2 (Supplementary Table 1) were sequenced on a HiSeq4000 (RRID:SCR_016386, Illumina) using custom sequencing primers with following read lengths: 50 + 10 + 12 + 50 (Read1 + Index1 + Index2 + Read2). Indexing primers and sequencing primers are in Supplementary Table 26. The nuclei indexing version (v1 or v2) used for each library is indicated in Supplementary Table 26.

## Nuclei indexing scheme

To generate snATAC-seq libraries we used initially an indexing scheme as previously described (version 1)[22,25]. Here, 16 p5 and 24 p7 indexes were combined to generate an array of 384 indexes for tagmentation and 16 i5 as well as 48 i7 indexes were combined for an array of 768 PCR indexes. Owing to this library design, it is required to sequence all four indexes to assign a read to a specific nucleus with long reads and a constant base sequence for both indices reads between i and p barcodes. Therefore, the resulting libraries were sequenced with 25% spike-in library on a HiSeq2500 (RRID:SCR_016383) and these read lengths: 50 + 43 + 37 + 50 (ref. [25]).

To generate libraries compatible with other sequencers and not requiring spike-in libraries or custom sequencing recipes, we modified the library scheme (Version 2). For this, we used 384 individual indices for T7 and combined with one T5 with a universal index sequence for tagmentation (for a total of 384 tagmentation indexes). For PCR, we used 768 different i5 indexes and combined with a universal i7 primer index sequence. Tagmentation indexes were 10 bp and PCR indexes 12-bp long. We made sure, that the hamming distance between every two barcodes was ≥ 4, the GC content between 37.5–62.5%, and the number of repeats ≤ 3. The resulting libraries were sequenced on a HiSeq4000 with custom primers and these read lengths: 50 + 10 + 12 + 50 (Supplementary Table 26).

## snATAC-seq data using the 10x Chromium platform

Brain nuclei were pelleted with a swinging bucket centrifuge (500*g*, 5 min, 4 °C; 5920R, Eppendorf). Nuclei pellets were resuspended in 1 ml nuclei permeabilization buffer (5% BSA, 0.2% IGEPAL-CA630, 1 mM DTT and cOmpleteTM, EDTA-free protease inhibitor cocktail (Roche) in PBS). Nuclei were pelleted again (500*g*, 5 min, 4 °C; 5920R, Eppendorf) and washed with wash buffer (10 mM Tris-HCl (pH 7.5), 10mM NaCl, 3 mM MgCl$_2$, 0.1% Tween-20, and 1% BSA (Proliant 7500804) in molecular biology-grade water). Nuclei were pelleted (500*g*, 5 min, 4 °C; 5920R, Eppendorf) and resuspended in 30 μl of 1× nuclei Buffer (10x Genomics). Nuclei were counted using a haemocytometer, and 15,360 nuclei were used for tagmentation. Single-cell ATAC-seq libraries were generated using the Chromium Single Cell ATAC Library & Gel Bead Kit (10x Genomics, 1000110), Chromium Chip E Single Cell ATAC kit (10x Genomics, 1000155) and Chromium i7 Multiplex Kit N, Set A (10x Genomics, 1000084) following manufacturer instructions. Final libraries were quantified using a Qubit fluorimeter (Life Technologies) and the nucleosomal pattern was verified using a Tapestation (High Sensitivity D1000, Agilent). Libraries were sequenced on NextSeq 500 and NovaSeq 6000 sequencers (Illumina) with following read lengths: 50 + 8 + 16 + 50 (Read1 + Index1 + Index2 + Read2). After demultiplexing, the Index2 (cell index) was transferred to the read name, in order to keep the same fastq format for downstream processing.

## Processing and alignment of sequencing reads

Paired-end sequencing reads were demultiplexed and the cell index transferred to the read name. Sequencing reads were aligned to mm10 reference genome using bwa[64]. After alignment, we used the R package ATACseqQC (1.10.2)[65] to check for fragment length contribution which is characteristic for ATAC-seq libraries. Next, we combined the sequencing reads to fragments, and for each fragment we performed the following quality control: (1) keep only fragments quality score MAPQ > 30; (2) keep only the properly paired fragments with length < 1,000 bp; (3) PCR duplicates were further removed with SnapTools

(https://github.com/r3fang/SnapTools, RRID:SCR_018097)[26]. Reads were sorted on the basis of the cell barcode in the read name.

## TSS enrichment calculation

Enrichment of ATAC-seq accessibility at TSSs was used to quantify data quality without the need for a defined peak set. The method for calculating enrichment at TSS was adapted from previously described. TSS positions were obtained from the GENCODE database v16 (RRID:SCR_014966)[38]. In brief, Tn5 corrected insertions (reads aligned to the positive strand were shifted +4 bp and reads aligned to the negative strand were shifted −5 bp) were aggregated ± 2,000 bp relative (TSS strand-corrected) to each unique TSS genome-wide. Then this profile was normalized to the mean accessibility ± 1,900–2,000 bp from the TSS and smoothed every 11 bp. The max of the smoothed profile was taken as the TSS enrichment.

## Doublet removal

We used a modified version of Scrublet (RRID:SCR_018098)[66] to remove potential doublets for every dataset independently. Peaks were called using MACS2 scores for aggregate accessibility profiles on each sample. Next, cell-by-peak count matrices were calculated and used as input, with default parameters. Doublet scores were calculated for both observed nuclei $\{x_i\}$ and simulated doublets $\{y_i\}$ using Scrublet (RRID:SCR_018098)[66]. Next, a threshold $\theta$ is selected based on the distribution of $\{y_i\}$, and observed nuclei with doublet score larger than $\theta$ are predicted as doublets. To determine $\theta$, we fit a two-component mixture distribution by using function normalmixEM from R package mixtools[67]. The lower component contained most embedded doublet types, and the other component contained majority of neo-typic doublets (collision between nuclei from different clusters. We selected the threshold $\theta$ where the $p_1 \cdot pdf(x, \mu_1, \sigma_1) = p_2 \cdot pdf(x, \mu_2, \sigma_2)$ in which $p$ denotes probability; pdf denotes probability density function. This value suggested that the nuclei have same chance of belonging to both classes.

### Cell clustering

We used an iterative clustering strategy using the snapATAC[26] package (RRID:SCR_018097) with slight modifications as detailed below. For round 1 clustering, we clustered and finally merged single nuclei to three main cell classes: non-neurons, GABAergic neurons, and glutamatergic neurons. For each main cell class, we performed another round of clustering to identify major cell subclasses. Last, for each subclass, we performed a third round of clustering to find cell types.

Detailed description for every step is as follows. (1) Nuclei filtering. Nuclei with ≥1,000 uniquely mapped fragments and TSS enrichment > 10 were filtered for individual dataset. Second, potential barcode collisions were also removed for individual datasets. (2) Feature bin selection. First, we calculated a cell-by-bin matrix at 5-kb resolution for every dataset independently and subsequently merged the matrices. Second, we converted the cell-by-bin count matrix to a binary matrix. Third, we filtered out any bins overlapping with the ENCODE blacklist (mm10, http://mitra.stanford.edu/kundaje/akundaje/release/black-lists/mm10-mouse/mm10.blacklist.bed.gz). Fourth, we focused on bins on chromosomes 1–19, X and Y. Last, we removed the top 5% bins with the highest read coverage from the count matrix. (3) Dimensionality reduction. SnapATAC applies a nonlinear dimensionality reduction method called diffusion maps, which is highly robust to noise and perturbation[26]. However, the computational time of the diffusion maps algorithm scales exponentially with the increase of number of cells. To overcome this limitation, we combined the Nyström method (a sampling technique)[68] and diffusion maps to present Nyström landmark diffusion map to generate the low-dimensional embedding for large-scale dataset.

A Nyström landmark diffusion maps algorithm includes three major steps:

(1) Sampling. We sampled a subset of $K$ ($K \ll N$) cells from $N$ total cells as 'landmarks'.

(2) Embedding. We computed a diffusion map embedding for $K$ landmarks.

(3) Extension. We projected the remaining ($N − K$) cells onto the low-dimensional embedding as learned from the landmarks to create a joint embedding space for all cells. Having more than 800,000 single nuclei at the beginning, we decided to apply this strategy on round 1 and 2 clustering. A total of 10,000 cells were sampled as landmarks and the remaining query cells were projected onto the diffusion maps embedding of landmarks. Later for the round 3 clustering, diffusion map embeddings were directly calculated from all nuclei.

(4) Eigenvector selection. To determine the number of eigenvectors of diffusion operator to include for downstream analysis, we generated an 'elbow plot', to rank all eigenvectors on the basis of the percentage of variance explained by each one. For each round of clustering, we selected the top 10–20 eigenvectors that captured most of the variance.

(5) Graph-based clustering. Using the selected significant eigenvectors, we next construct a $k$-nearest neighbour graph. Each cell is a node and the $k$-nearest neighbours of each cell were identified according to the Euclidian distance and edges were drawn between neighbours in the graph. Next, we applied the Leiden algorithm on the $k$-nearest neighbour graph using Python package leidenalg (https://github.com/vtraag/leidenalg)[69].

(6) Optimization on cluster resolution. We tested different 'resolution_parameter' parameters (step between 0 and 1 by 0.1) to determine the optimal resolution for different cell populations. For each resolution value, we tested whether there was clear separation between nuclei. To do so, we generated a cell-by-cell consensus matrix in which each element represents the fraction of observations two nuclei are part of the same cluster. A perfectly stable matrix would consist entirely of zeros and ones, meaning that two nuclei either cluster together or not in every iteration. The relative stability of the consensus matrices can be used to infer the optimal resolution. To this end, we generated a consensus matrix based on 300 rounds of Leiden clustering with randomized starting seed $s$. Let $M^s$ denote the $N \times N$ connectivity matrix resulting from applying Leiden algorithm to the dataset $D^s$ with different seeds. The entries of $M^s$ are defined as follows:

$$M^s(i,j) = f(x)$$
$$= \begin{cases} 1, & \text{if single nucleus } i \text{ and } j \text{ belong to the same cluster} \\ 0, & \text{otherwise} \end{cases}$$

Let $I^s$ be the $N \times N$ identicator matrix in which the $(i,j)$-th entry is equal to 1 if nucleus $i$ and $j$ are in the same perturbed dataset $D^s$, and 0 otherwise. Then, the consensus matrix $C$ is defined as the normalized sum of all connectivity matrices of all the perturbed $D^s$.

$$C(i,j) = \left( \frac{\sum_{s=1}^{S} M^s(i,j)}{\sum_{s=1}^{S} I^s(i,j)} \right)$$

The entry $(i,j)$ in the consensus matrix is the number of times single nucleus $i$ and $j$ were clustered together divided by the total number of times they were selected together. The matrix is symmetric, and each element is defined within the range [0, 1]. We examined the cumulative distribution function (CDF) curve and calculated proportion of ambiguous clustering (PAC) score to quantify stability at each resolution. The resolution with a local minimum of the PAC scores denotes the parameters for the optimal clusters. In the case these were multiple local minimal PACs, we picked the one with higher resolution. Another measurement is dispersion coefficient, which reflects the dispersion (ranges from 0 to 1) of the consensus matrix $M$ from the value 0.5. The closer to 1 is the dispersion coefficient, the more perfect is consensus matrix, and thus the more stable is the clustering. In a perfect

consensus matrix, all entries are 0 or 1, meaning that all connectivity matrices are identical. The dispersion coefficient is defined as:

$$\rho = \frac{1}{n^2} \sum_{i=1} \sum_{j=1} 4\left(M(i,j) - \frac{1}{2}\right)^2$$

Finally, for every cluster, we tested whether we could identify differential features compared to all other nuclei (background) and the nearest nuclei (local background) using the function 'findDAR'.

(7) Visualization. For visualization, we applied UMAP[58]. Using the cell embedding, we applied both $k$-nearest neighbor batch effect test (kBET) and local inverse Simpson's index (LISI) analysis to test the robustness of the clutering results to variation of sequencing depth, signal-to-noise ratios, and batches.

## Clustering for adult neurogenesis lineages

We performed separated cell clustering following above strategy for two lineages:

1. Adult neurogenesis in the SGZ: we extracted 83,583 nuclei from 8 brain dissections at or surrounding the SGZ, including CA-1, CA-2, CA-3, CA-4, DG-1, DG-2, DG-3, DG-4 (Supplementary Table 1). Then, we performed cell clustering on 83,583 nuclei for 6 cell types: astrocytes (ASC); dentate gyrus radial glia-like cells; NIPCs; granule neuroblasts (DGNBL1/2); and granule neurons.

2. Adult neurogenesis in the SVZ: we extracted 25,923 nuclei from 8 brain dissections at or surrounding the SVZ, including MOB, AON, ACB-1, ACB-2 CP-1, CP-2, LSX-1 and LSX-2 (Supplementary Table 1). Then, we performed cell clustering on 25,923 nuclei for 5 cell types: astrocytes; subventricular zone radial glia-like cells; neuronal intermediate progenitor cells; neuroblasts (OBNBL); and inhibitory neurons in olfactory (OBGA1).

## Dendrogram construction for mouse brain cell types

First, we calculated for cCRE the median accessibility per cluster and used this value as cluster centroid. Next, we calculated the coefficient of variant for the cluster centroid of each element across major cell types. Finally, we only kept variable elements with coefficient of variants that were larger than 1.5 for dendrogram construction.

We used the set of variable features defined above to calculate a correlation-based distance matrix. Next, we performed linkage hierarchical clustering using the R package pvclust (v.2.0)[70] with parameters method.dist = "cor" and method.hclust = "ward.D2". The confidence for each branch of the tree was estimated by the bootstrap resampling approach with 1,000 rounds.

## Regional specificity of cell types

The specificity score is defined as Jensen–Shannon divergence, which measures the similarity between two probability distributions. For each cell type, the contribution of different brain regions was first calculated. Then, we compared this distribution with the contribution of brain regions calculated from all sampled cells. We used the function 'JSD' from R package philentropy for this analysis[71].

## Identification of reproducible peak sets in each cell cluster

We performed peak calling according to the ENCODE ATAC-seq pipeline (https://www.encodeproject.org/atac-seq/). For every cell cluster, we combined all properly paired reads to generate a pseudo-bulk ATAC-seq dataset for individual biological replicates. In addition, we generated two pseudo-replicates that comprise half of the reads from each biological replicate. We called peak for each of the four datasets and a pool of both replicates independently. Peak calling was performed on the Tn5-corrected single-base insertions using the MACS2 score[30] with these parameters: −shift -75 −extsize 150 −nomodel −call-summits −SPMR -q 0.01. Finally, we extended peak summits by 250 bp on either side to a final width of 501 bp for merging and downstream analysis. To generate

a list of reproducible peaks, we kept peaks that (1) were detected in the pooled dataset and overlapped ≥50% of peak length with a peak in both individual replicates; or (2) were detected in the pooled dataset and overlapped ≥50% of peak length with a peak in both pseudo-replicates.

We found that when cell population varied in read depth or the number of nuclei, the MACS2 score varied proportionally owing to the nature of the Poisson distribution test in MACS2 scores[30]. Ideally, we would perform a reads-in-peaks normalization, but in practice, this type of normalization was not possible because we did not know how many peaks we would get. To account for differences in performance of MACS2 scores[30] on the basis of read depth and/or number of nuclei in individual clusters, we converted MACS2 peak scores ($-\log_{10}(q\text{-value})$) to 'score per million'[72]. We filtered reproducible peaks by choosing a score-per-million cut-off of 2 to filter reproducible peaks.

We only kept reproducible peaks on chromosome 1–19 and both sex chromosomes, and filtered ENCODE mm10 blacklist regions (mm10, http://mitra.stanford.edu/kundaje/akundaje/release/blacklists/mm10-mouse/mm10.blacklist.bed.gz). A union peak list for the whole dataset was obtained by merging peak sets from all cell clusters using BEDtools (RRID:SCR_006646)[73].

Lastly, because snATAC-seq data are very sparse, we selected only elements that were identified as open chromatin in a significant fraction of the cells in each cluster. To this end, we first randomly selected the same number of non-DHS regions (approximately 670,000 elements) from the genome as background and calculated the fraction of nuclei for each cell type that showed a signal at these sites. Next, we fitted a zero-inflated beta model, and empirically identified a significance threshold of FDR < 0.01 to filter potential false positive peaks. Peak regions with FDR < 0.01 in at least one of the clusters were included in downstream analysis.

## Computing chromatin accessibility scores

Accessibility of cCREs in individual clusters was quantified by counting the fragments in individual clusters normalized by read depth (CPM). For each gene, we summed up the counts within the gene body +2 kb upstream to calculate the gene activity score. The gene activity score were used for integrative analysis with scRNA-seq. For better visualization, we smoothed the gene activity score to 50 nearest neighbour nuclei using Markov affinity-based graph imputation of cells (MAGIC)[74].

## Integrative analysis of snATAC-seq and scRNA-seq datasets

For integrative analysis, we downloaded level 5 clustering data from the Mouse Brain Atlas website (http://mousebrain.org)[2]. First, we filtered brain regions that matched samples profiled in this study using these attributes for 'region': 'CNS', 'cortex', 'hippocampus', 'hippocampus,cortex', 'olfactory bulb', 'striatum dorsal', 'striatum ventral', 'dentate gyrus', 'striatum dorsal, striatum ventral', 'striatum dorsal, striatum ventral, dentate gyrus', 'pallidum', 'striatum dorsal, striatum ventral, amygdala', 'striatum dorsal, striatum ventral', 'telencephalon', 'brain' and 'sub ventricular zone, dentate gyrus'.

Second, we manually subset cell types into three groups by checking the attribute in 'taxonomy_group': non-neurons: 'vascular and leptomeningeal cells', 'astrocytes', 'oligodendrocytes', 'ependymal cells', 'microglia', 'oligodendrocyte precursor cells', 'olfactory ensheathing cells', 'pericytes', 'vascular smooth muscle cells', 'perivascular macrophages', 'dentate gyrus radial glia-like cells', 'subventricular zone radial glia-like cells', 'vascular smooth muscle cells', 'vascular endothelial cells', 'vascular and leptomeningeal cells'; gabaergic neurons: 'non-glutamatergic neuroblasts', 'telencephalon projecting inhibitory neurons', 'olfactory inhibitory neurons', 'glutamatergic neuroblasts', 'cholinergic and monoaminergic neurons', 'di- and mesencephalon inhibitory neurons', 'telencephalon inhibitory interneurons', 'peptidergic neurons'; glutamatergic neurons: 'dentate gyrus granule neurons', 'di- and mesencephalon excitatory neurons', 'telencephalon projecting excitatory neurons'.

To directly compare our single-nucleus chromatin accessibility-derived cell clusters with the single-cell transcriptomics defined taxonomy of the mouse brain[2], we first used the snATAC-seq data to impute RNA expression levels (gene activity scores) according to the chromatin accessibility of gene promoter and gene body as previously described[75]. We performed integrative analysis with scRNA-seq using Seurat 3.0 (RRID:SCR_016341) to compare cell annotation between different modalities[75]. We randomly selected 200 nuclei (and used all nuclei for cell cluster with fewer than 200 nuclei) from each cell cluster for integrative analysis. We first generated a Seurat object in R by using previously calculated gene activity scores, diffusion map embeddings and cell cluster labels from snATAC-seq. Then, variable genes were identified from scRNA-seq and used for identifying anchors between these two modalities. Finally, to visualize all the cells together, we co-embedded the scRNA-seq and snATAC-seq profiles in the same low dimensional space.

To quantify the similarity between cell clusters from two modalities, we calculated an overlapping score as the sum of the minimum proportion of cells or nuclei in each cluster that overlapped within each co-embedding cluster[5]. Cluster overlaps varied from 0 to 1 and were visualized as a heat map with snATAC-seq clusters in rows and scRNA-seq clusters in columns. We found strong correspondence between the two modalities, which was evidenced by co-embedding of both transcriptomic (T-type) and chromatin accessibility (A-type) cells in the same RNA–ATAC joint clusters (Extended Data Fig. 10a, Supplementary Table 5). For this analysis, we examined GABAergic neurons, glutamatergic neurons and non-neuronal cell classes separately (Extended Data Fig. 10, Supplementary Table 5).

### Identification of *cis*-regulatory modules

We used non-negative matrix factorization[76] to group cCREs into *cis*-regulatory modules on the basis of their relative accessibility across major clusters. We adapted non-negative matrix factorization (Python package: sklearn[77]) to decompose the cell-by-cCRE matrix $V$ ($N \times M$, $N$ rows: cCRE, $M$ columns: cell clusters) into a coefficient matrix $H$ ($R \times M$, $R$ rows: number of modules) and a basis matrix $W$ ($N \times R$), with a given rank $R$:

$$V \approx WH$$

The basis matrix defines module related accessible cCREs, and the coefficient matrix defines the cell cluster components and their weights in each module. The key issue to decompose the occupancy profile matrix was to find a reasonable value for the rank $R$ (that is, the number of modules). Several criteria have been proposed to decide whether a given rank $R$ decomposes the occupancy profile matrix into meaningful clusters. Here we applied two measurements 'sparseness'[78] and 'entropy'[79] to evaluate the clustering result. Average values were calculated from 100 times for non-negative matrix factorization runs at each given rank with random seed, which will ensure the measurements are stable.

Next, we used the coefficient matrix to associate modules with distinct cell clusters. In the coefficient matrix, each row represents a module and each column represents a cell cluster. The values in the matrix indicate the weights of clusters in their corresponding module. The coefficient matrix was then scaled by column (cluster) from 0 to 1. Subsequently, we used a coefficient > 0.1 (approximately the 95th percentile of the whole matrix) as threshold to associate a cluster with a module.

In addition, we associated each module with accessible elements using the basis matrix. For each element and each module, we derived a basis coefficient score, which represents the accessible signal contributed by all cluster in the defined module. In addition, we also implemented and calculated a basis-specificity score called 'feature score' for each accessible element using the 'kim' method[79]. The feature score ranges from 0 to 1. A high feature score means that a distinct element is specifically associated with a specific module. Only features that fulfil both of the following criteria were retained as module specific elements: (1) feature score greater than median + 3 standard deviations; (2) the maximum contribution to a basis component is greater than the median of all contributions (that is, of all elements of $W$).

### Identification of differentially accessible regions and definition of specificity score

To identify cCREs that were differentially accessible either in subtypes or in brain regions, we constructed a logistic regression model predicting cluster/region membership based on each cCRE individually and compares this to a null model with a likelihood ratio test. We used two functions 'fit_models' and 'compare_models' in R package Monocle3 (v.0.2.2)[80] to perform the differential test. We designed the full model as

$$\mathrm{logit}\left(P_{ij}\right) = a_j + m_j + r_j + \varepsilon_j$$

and a reduced mode as

$$\mathrm{logit}\left(P_{ij}\right) = a_j + r_j + \varepsilon_j$$

in which $P_{ij}$ represents the probability of $i$th site is accessible in the $j$th cell, $a$ is the $\log_{10}$-transformed total number of sites observed as accessible for the $j$th cell, $m$ is membership of the $j$th cell in either cluster or region being tested, $r$ is the replicate label for $j$th cell and $\varepsilon$ is an error term.

For each set of testing, between subtypes or between regions in cell type, we kept only cCREs that overlapped with peaks identified in corresponding cell types. A likelihood ratio test was then used to determine whether the full model (including cell cluster or region membership) provided a significantly better fit of the data than the reduced model. After correcting $P$ values using Benjamini–Hochberg method, we set an FDR cut-off as 0.001 to filter out significant differential cCREs.

The $\log_2$-transfomed fold change is used for two-group comparison, for multiple groups, we calculated a Jensen–Shannon divergence-based specificity score previously described[22] to better assign differential cCREs to cell type or brain region. The fraction of accessibility of each cluster $f$ was first calculated for each $i$th site. We normalized these scores by multiplying by corresponding scaling factors, which are considering different overall complexity across groups. To do so, median number of sites accessible $c$ in individual cells for each cluster was calculated and followed with $\log_{10}$-transforming. Then, we took the ratio of the average of $c$ (across all clusters) over the median accessibility in each cluster as scaling factor. These corrected fraction of accessibility for each cCRE was then converted to probability by scaling by groups. Then, we calculated Jensen–Shannon divergence between two probability distributions. For example, the probability distribution for the first cCRE as $d1$, to test whether this cCRE is specific in group 1, we assumed another probability distribution:

$$d2 = \begin{cases} 1, & \text{group 1} \\ 0, & \text{otherwise} \end{cases}$$

Function 'JSD' in R package 'philentropy' was used to calculate Jensen–Shannon divergence between these two probability distributions, and Jensen–Shannon-based specificity (JSS) scores was defined as:

$$\mathrm{JSS} = 1 - \sqrt{\text{Jensen Shannon divergence}}$$

For each group, we calculated the JSS score for every cCRE. To find a reasonable cut-off to determine restricted or general cCREs, we consider JSS scores from all cCREs that are not identified as differential accessible (from likelihood ratio test) as a background distribution, and JSS scores from cCREs that passed our likelihood ratio test threshold

and had positive values to be true positives. We set an empirical FDR cut-off where the type I error was no more than 5%.

Finally, the differential cCREs could be aligned to several cell types or brain regions based on the JSS score, we named the one can be assigned to only one type or region as region-specific or cell-type-specific cCREs.

## Comparison between the regional specificity of cell types defined by snATAC-seq data and the spatial ISH signals of cell-type-specific genes

To validate the regional specificity of cell types, we took advantage of the spatially mapped quantified ISH expression from ABA[44] in five matched major brain structures, isocortex, olfactory areas, hippocampal formation (HPF), striatum (STR), pallidum. We used the 'differential search' function to identify 10,269 genes with increased expression in these five brain regions compared to all brain regions 'grey matter' (expression level >1 and fold change > 1). We also identified cell-type-specific genes using Seurat[75] with default parameters for each joint cluster from integrative analysis (fold change > 1 and FDR < 0.05) (Extended Data Fig. 10). 505 cell-type-specific genes (range from 1 to 53, 15 on average) overlapped with the list of genes with increased expression in the five brain regions. For each cell type, we calculated the regional specificity score (see previous section: regional specificity of cell types) on the basis of the relative contribution from five brain regions estimated from snATAC-seq datasets, and also a coefficient of variation based on averaged normalized ISH signals of cell-type-specific marker genes. For each cell-type-specific gene, we calculated the PCC between cell composition in five brain structures and spatial expression levels across the five brain structures derived from ISH.

Because the astrocyte subtypes identified in our study were not resolved in scRNA-seq studies, we identified subtype-specific genes for astrocyte subtypes using chromatin accessibility from snATAC-seq using a likelihood ratio test. The cell-type-specific genes were filtered by FDR less than 0.001 from the likelihood ratio test and empirical FDR cut-off of no more than 5% for JSS scores. Then, we calculated the fraction of overlap between spatially mapped ISH genes from different brain structures and genes with astrocyte subtype-specific accessibility.

## Predicting enhancer–promoter interactions

First, co-accessible regions were identified for all open regions in each cell cluster (randomly selected 200 nuclei, and using all nuclei for cell cluster with <200 nuclei) separately, using Cicero[37] with following parameters: aggregation $k = 10$, window size = 500 kb, distance constraint = 250 kb. To find an optimal co-accessibility threshold for each cluster, we generated a random shuffled cCRE-by-cell matrix as background and identified co-accessible regions from this shuffled matrix. We fitted the distribution of co-accessibility scores from random shuffled background into a normal distribution model by using the R package fitdistrplus[81]. Next, we tested every co-accessibility pairs and set the cut-off at co-accessibility score with an empirically defined significance threshold of FDR < 0.01.

CCRE outside of ±1 kb of the TSS in GENCODE mm10 (v.16, RRID:SCR_014966)[38]. were considered distal. Next, we assigned co-accessibility pairs to three groups: proximal-to-proximal, distal-to-distal, and distal-to-proximal. In this study, we focus only on distal-to-proximal pairs. We further used RNA expression from matched T-types to filter out pairs that were linked to non-expressed genes (normalized UMI < 5).

We calculated PCC between gene expression and cCRE accessibility across joint RNA-ATAC clusters to examine the relationship between co-accessibility pairs. To do so, we first aggregated all nuclei or cells from scRNA-seq and snATAC-seq for every joint cluster to calculate accessibility scores ($\log_2(CPM)$) and relative expression levels ($\log_2(normalized UMI)$). Then, PCC was calculated for every cCRE–gene pair within a 1-Mb window centred on the TSS for every gene. We also

generated a set of background pairs by randomly selecting regions from different chromosomes and shuffling of cluster labels. Finally, we fit a normal distribution model and defined a cut-off at PCC score with empirically defined significance threshold of FDR < 0.01, to select significant positively correlated cCRE–gene pairs.

## Identification of candidate driver transcription factors

We used the Taiji pipeline[45] to identify candidate driver transcription factors in cell clusters. In brief, for each cell type cluster, we constructed the transcription factor regulatory network by scanning transcription factor motifs at the accessible chromatin regions and linking them to the nearest genes. The network is directed with edges from transcription factors to target genes. The weights of the genes in the network were determined on the basis of the RNA expression level (gene activity score for SGZ NIPCs only, because there is no corresponding T-type) of corresponding T-types. The weights of the edges were calculated by the relative accessibility of the promoters of the source transcription factors. We then used the personalized PageRank algorithm to rank the transcription factors in the network. The output of Taiji pipeline is transcription-factor-by-cell type matrix with PageRank scores. From the output matrix, we calculated coefficient of variation across cell types. To identify driver transcription factors, we used following criteria: (1) transcription factors have FDR less than 0.001; (2) transcription factors have coefficient of variant larger than the mean of coefficient of variant; (3) PageRank score should be ranked in the top 25% of all transcription factors for each cell type; (4) RNA expression level (CPM) is larger than 20, which we consider as an expressed transcription factor in corresponding cell type.

## Motif enrichment

We performed both de novo and known motif enrichment analysis using Homer (v.4.11, RRID:SCR_010881)[61]. For cCREs in the consensus list, we scanned a region of ±250 bp around the centre of the element. And for proximal or promoter regions, we scanned a region of ±1,000 bp around the TSS. Randomly selected background regions are used for motif discovery. To identify motif enriched in different cell types or brain regions, we use variable cCREs as input and invariable cCREs as background.

## GREAT analysis

Gene Ontology annotation of cCREs was performed using GREAT (version 4.0.4, RRID:SCR_005807)[82] with default parameters. Gene Ontology biological process was used for annotations.

## Gene Ontology enrichment

We perform Gene Ontology enrichment analysis using the R package Enrichr (RRID:SCR_001575)[83]. The gene set library 'GO_Biological_Process_2018' was used with default parameters. The combined score is defined as the $P$ value computed using the Fisher's exact test multiplied with the $z$-score of the deviation from the expected rank.

## GWAS enrichment

To enable comparison to GWASs of human phenotypes, we used lift-Over with settings '-minMatch=0.5' to convert accessible elements from mm10 to hg19 genomic coordinates[51]. Next, we reciprocal lifted the elements back to mm10 and only kept the regions that mapped to original loci. We further removed converted regions with lengths greater than 1 kb.

We obtained GWAS summary statistics for quantitative traits related to neurological disease and control traits (Supplementary Table 25): age first birth and number of children born[84], tiredness[85], Crohns disease[86], attention deficit hyperactivity disorder[87], allergy[88], birth weight[89], bipolar disorder[90], insomnia[91], sleep duration[92], neuroticism[93], coronary artery disease[94], rheumatoid arthritis[95], educational attainment[96], schizophrenia[97], age at menarche[98], tobacco use disorder

(ftp://share.sph.umich.edu/UKBB_SAIGE_HRC/, Phenotype code: 318)[99], intelligence[100], amyotrophic lateral sclerosis[101], anorexia nervosa[102] and height[103].

We prepared summary statistics to the standard format for linkage disequilibrium score regression. We used homologous sequences for each major cell types as a binary annotation, and the superset of all candidate regulatory peaks as the background control.

For each trait, we used cell-type-specific linkage disequilibrium score regression (https://github.com/bulik/ldsc) to estimate the enrichment coefficient of each annotation jointly with the background control[52].

## Fine mapping

We obtained 99% credible sets for schizophrenia from the Psychiatric Genomics Consortium website (https://www.med.unc.edu/pgc/). Potential causal variants with a posterior probabilities of association score larger than 1% were used for overlapping with cCREs.

## External datasets

The datasets used for intersection analysis area as follows: representative DNase hypersensitivity site regions for both hg19 and mm10 were obtained from SCREEN database (https://screen.encodeproject.org)[104,105]. ChromHMM[31,33] states for mouse brain were downloaded from GitHub (https://github.com/gireeshkbogu/chromatin_states_chromHMM_mm9), and coordinates of ChromHMM states were mapped using LiftOver (https://genome.ucsc.edu/cgi-bin/hgLiftOver) to mm10 with default parameters[51]. PhastCons[59] conserved elements were download from the UCSC genome browser (http://hgdownload.cse.ucsc.edu/goldenpath/mm10/phastCons60way/). CTCF-binding sites were downloaded from the Mouse Encode Project[31] (http://chromosome.sdsc.edu/mouse/). CTCF-binding sites from the cortex and olfactory bulb were used in this study. Peaks were extended ±500 bp from the loci of peak summits and mapped using LiftOver to mm10[51].

## Statistics

No statistical methods were used to predetermine sample sizes. There was no randomization of the samples, and investigators were not blinded to the specimens being investigated. However, the clustering of single nuclei on the basis of chromatin accessibility was performed in an unbiased manner, and cell types were assigned after clustering. Low-quality nuclei and potential barcode collisions were excluded from downstream analysis as outlined above.

## Reporting summary

Further information on research design is available in the Nature Research Reporting Summary linked to this paper.

## Data availability

Demultiplexed data can be accessed via the NEMO archive (NEMO, RRID:SCR_016152) at: https://assets.nemoarchive.org/dat-wywv153. Processed data are available on our web portal and can be explored here: http://catlas.org/mousebrain. Additional data are available in the NCBI Gene Expression Omnibus (GEO) under accession number GEO173650 and upon request.

## Code availability

Custom code and scripts used for analysis can be accessed at: https://github.com/yal054/snATACutils and https://github.com/r3fang/SnapATAC.

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

**Acknowledgements** We thank J. Huang and E. Lein for critical reading of the manuscript; R. Raviram, Y. Zhang, G. Li and J. Hocker for discussions; and all other members of the Ren laboratory for valuable inputs. We extend our gratitude to the QB3 Macrolab at UC Berkeley for purification of the Tn5 transposase. We thank CNS muSlice LLC for making the brain slicer used in this study. This publication includes data generated at the UC San Diego IGM Genomics Center utilizing an Illumina NovaSeq 6000 that was purchased with funding from a National Institutes of Health (NIH) SIG grant (S10 OD026929). This study was supported by NIH grant U19MH11483. J.R.E. is an Investigator of the Howard Hughes Medical Institute. Work at the Center for Epigenomics was also supported by the UC San Diego School of Medicine.

**Author contributions** Study supervision: B.R. Contribution to data analysis: Y.E.L., K.Z., R.F., Y.Q., O.P., Y.Y. Contribution to data generation: S.P., X.H., J.Y.H., X.W., D.G., S.K., J.L., A.P.-D., M.M.B., X.Y., N.K., M.M., Y.S. Contribution to web portal: Y.E.L., Z.Z., B.L. Contribution to data interpretation: Y.E.L., S.P., B.R., J.R.E., M.M.B., E.A.M., J.C., K.J.G., M.N. Contribution to writing the manuscript: Y.E.L., S.P., B.R. All authors edited and approved the manuscript.

**Competing interests** B.R. is a co-founder and consultant of Arima Genomics Inc. and co-founder of Epigenome Technologies. K.J.G. is a consultant of Genentech and shareholder in Vertex Pharmaceuticals. A. J.R.E is on the scientific advisory board of Zymo Research, Inc.

**Additional information**
**Correspondence and requests for materials** should be addressed to B.R.

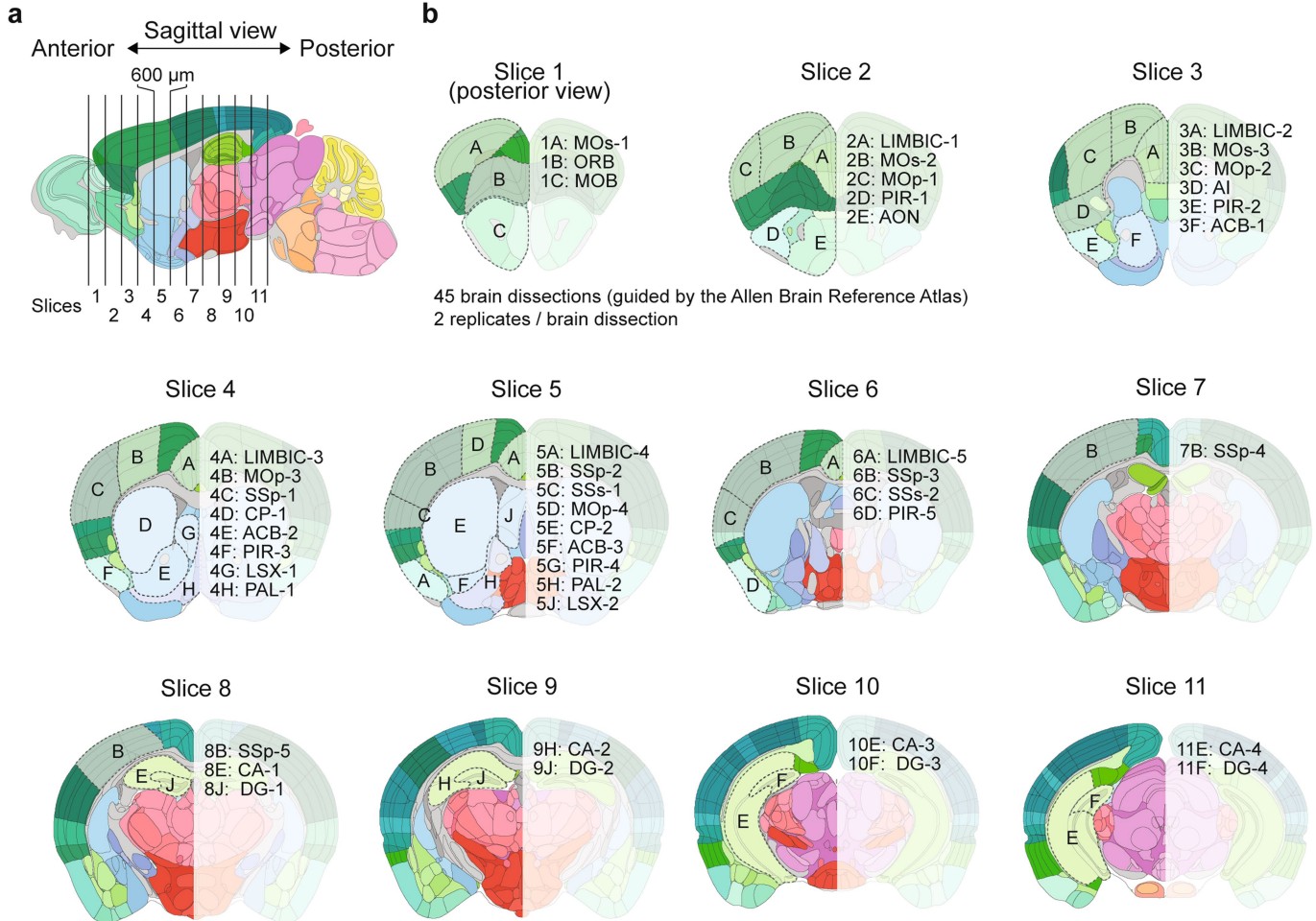

**Extended Data Fig. 1 | Anatomic maps of the 45 dissections in the adult mouse cerebrum. a**, Schematic of brain tissue dissection strategy. Mouse brains were cut into 600-µm-thick coronal slices. **b**, Brain regions dissected from each coronal slice are marked according to the Allen Brain Reference Atlas[28]. The frontal view of each slice from slices 1–11 is shown, with the dissected regions alphabetically labelled on the left, and the anatomic labelling listed on the right. A detailed list of the dissected regions and the full anatomic labelling can be found in Supplementary Table 1.

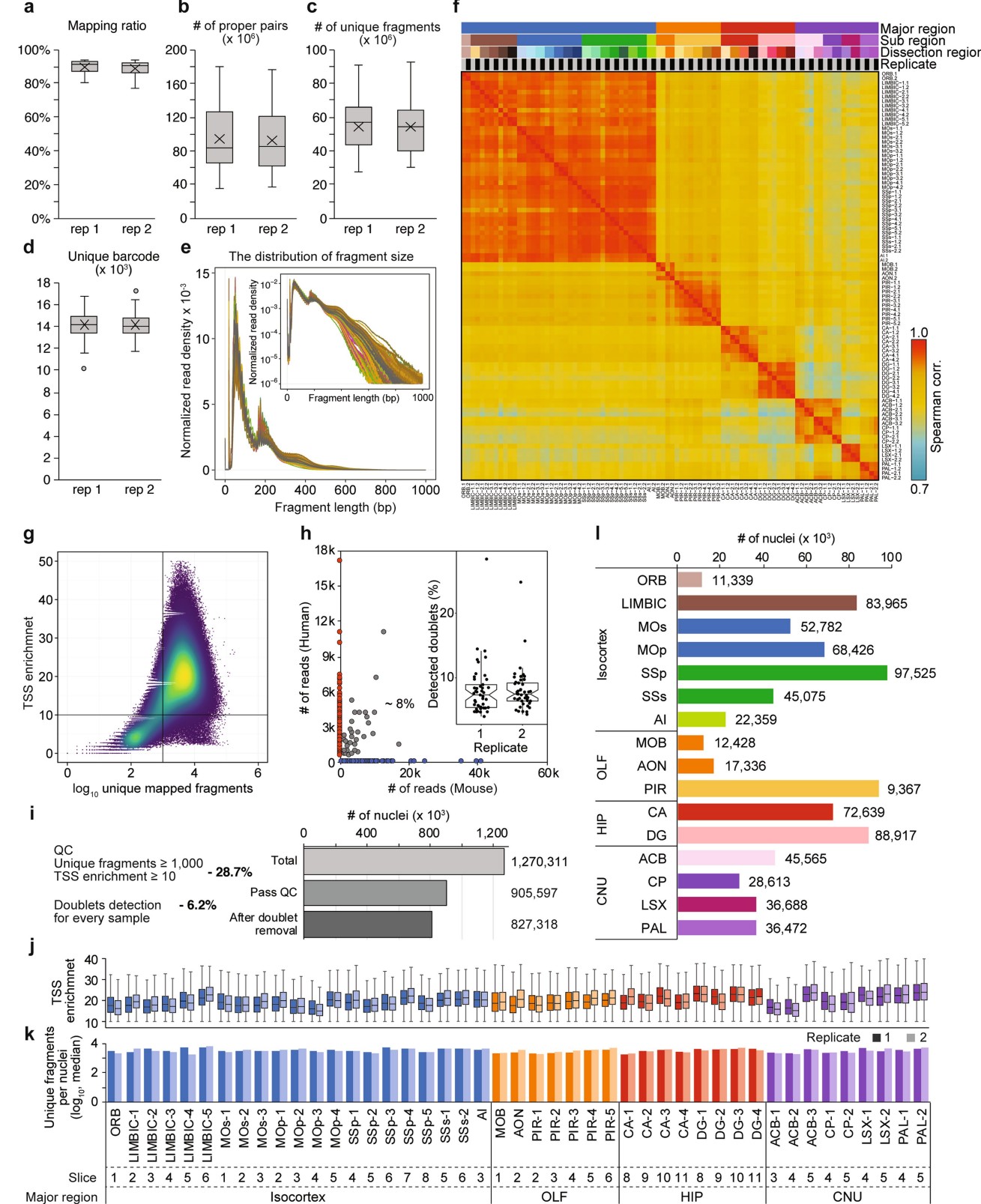

**Extended Data Fig. 2** | See next page for caption.

**Extended Data Fig. 2 | Quality control metrics of the snATAC-seq datasets.**
**a**, Box plots showing the distribution of mapping ratios (the fraction of the mapped sequencing reads) in replicates (rep) 1 and 2 of snATAC-seq experiments from each brain dissection. **b**, Box plots showing the distribution of the number of proper read pairs (reads are correctly oriented) in replicates 1 and 2 of snATAC-seq experiments. **c**, Box plots showing the distribution of numbers of unique chromatin fragments detected in replicates 1 and 2 of snATAC-seq experiments. **d**, Box plots showing the distribution of number of unique barcodes captured in replicates 1 and 2 of snATAC-seq experiments. **e**, Frequency distribution plot showing the fragment size distribution of each snATAC-seq dataset. **f**, Heat map showing the pairwise Spearman correlation coefficients between snATAC-seq datasets. The column and row names consist of two parts: brain region name and replicate label. **g**, Dot plot illustrating fragments per nucleus and individual TSS enrichment. Nuclei in top right quadrant were selected for analysis (TSS enrichment > 10 and > 1,000 fragments per nucleus). **h**, Fraction of cell collision estimated from species mix samples. Inset shows the fraction of potential barcode collisions detected in snATAC-seq libraries using a modified version of Scrublet[66]. **i**, Number of nuclei retained after each step of quality control. **j**, Distribution of TSS enrichment. **k**, The number of uniquely mapped fragments per nucleus for individual libraries. **l**, The number of nuclei passing quality control for subregions. All box plots are stylized as in Fig. 2c.

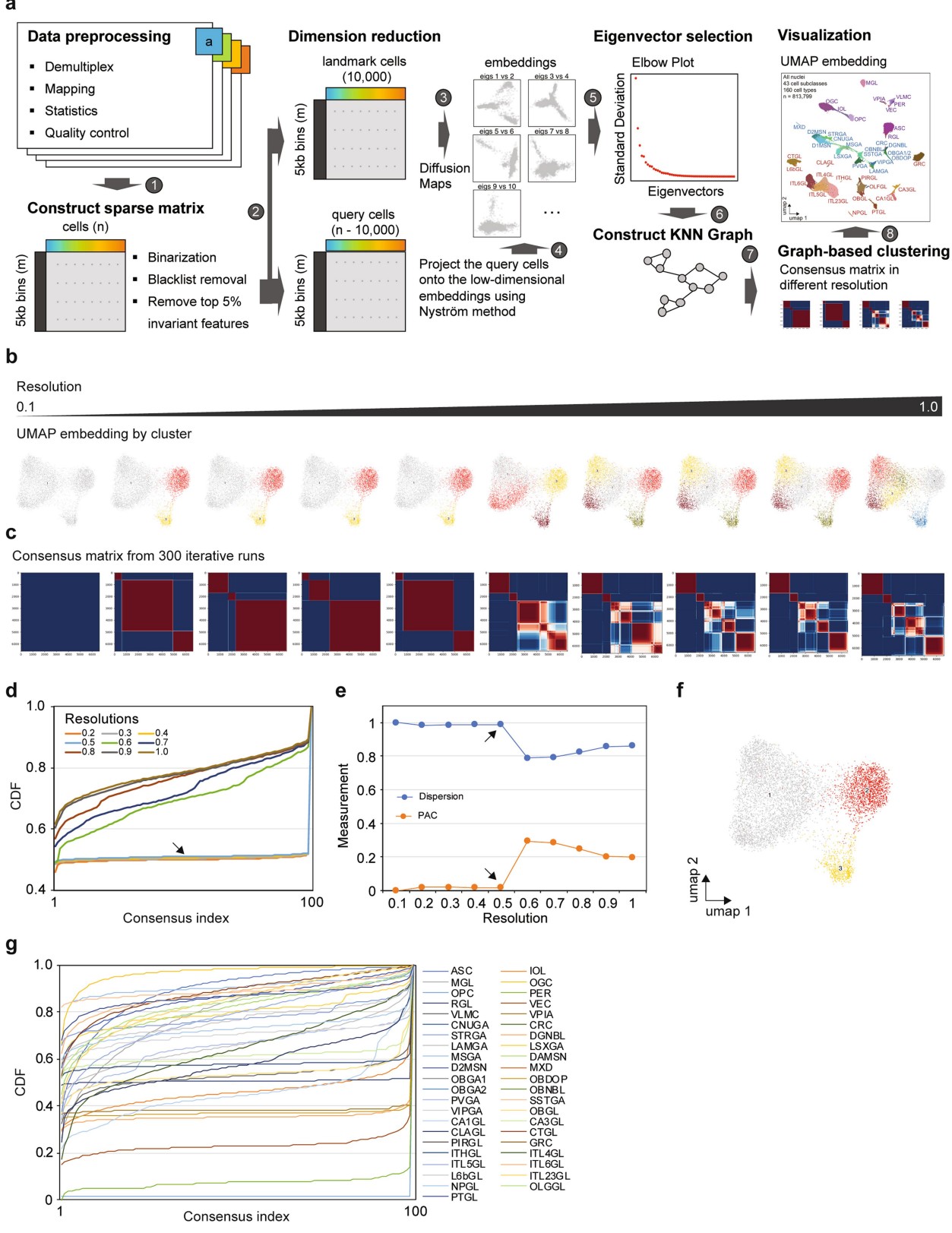

**Extended Data Fig. 3 | Cell clustering based on snATAC-seq data.**
**a**, Schematic diagram of the cell clustering pipeline. **b**, UMAP[58] embedding of a representative cell clustering at different resolutions from 0.1 to 1.0 using Leiden algorithm. **c**, Consensus matrix from 300 iterative clustering runs with different resolutions. **d**, Cluster stability at different resolutions was assessed using a CDF of consensus matrices. High values illustrate nuclei that clustered together in most cases. **e**, The PAC and dispersion coefficient at different resolutions. A low PAC and high dispersion coefficient indicates the best and most stable clustering. **f**, Optimal cell clustering result for a representative subclass (resolution = 0.5). **g**, The CDF curve of consensus matrix at an optimal resolution for every subclass.

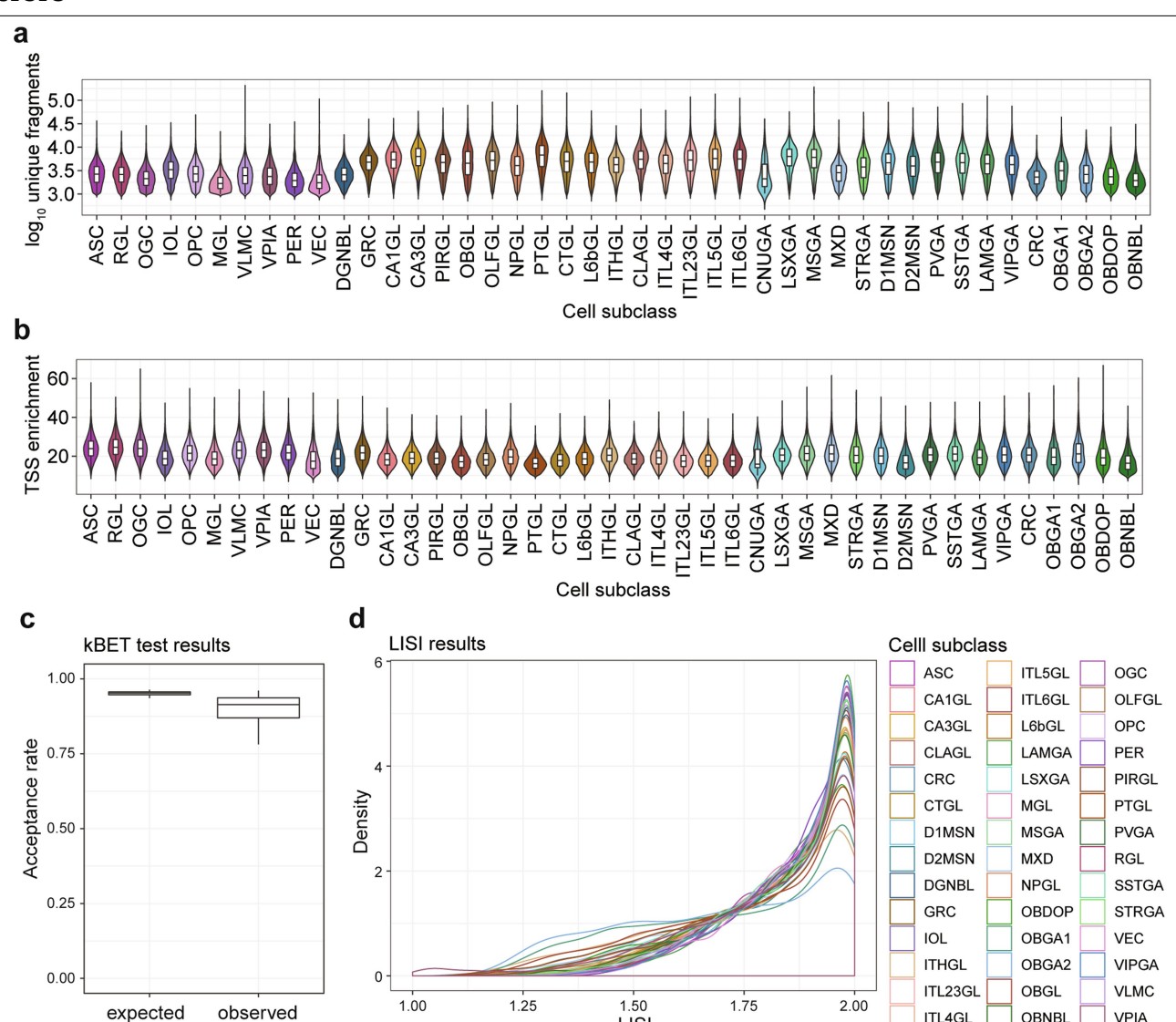

**Extended Data Fig. 4 | Summary statistics of snATAC-seq datasets in the current study. a**, Violin plots showing the log-transformed number of unique fragments per nucleus in each cell subclass identified. **b**, Violin plots showing the TSS enrichment in each nucleus of each subclass. **c**, Acceptance rate from *k*-nearest neighbour batch effect test (kBET)[106] for each subclass of cerebral cells. **d**, Distribution of the local inverse Simpson's index (LISI) scores[107] for cells in each subclass.

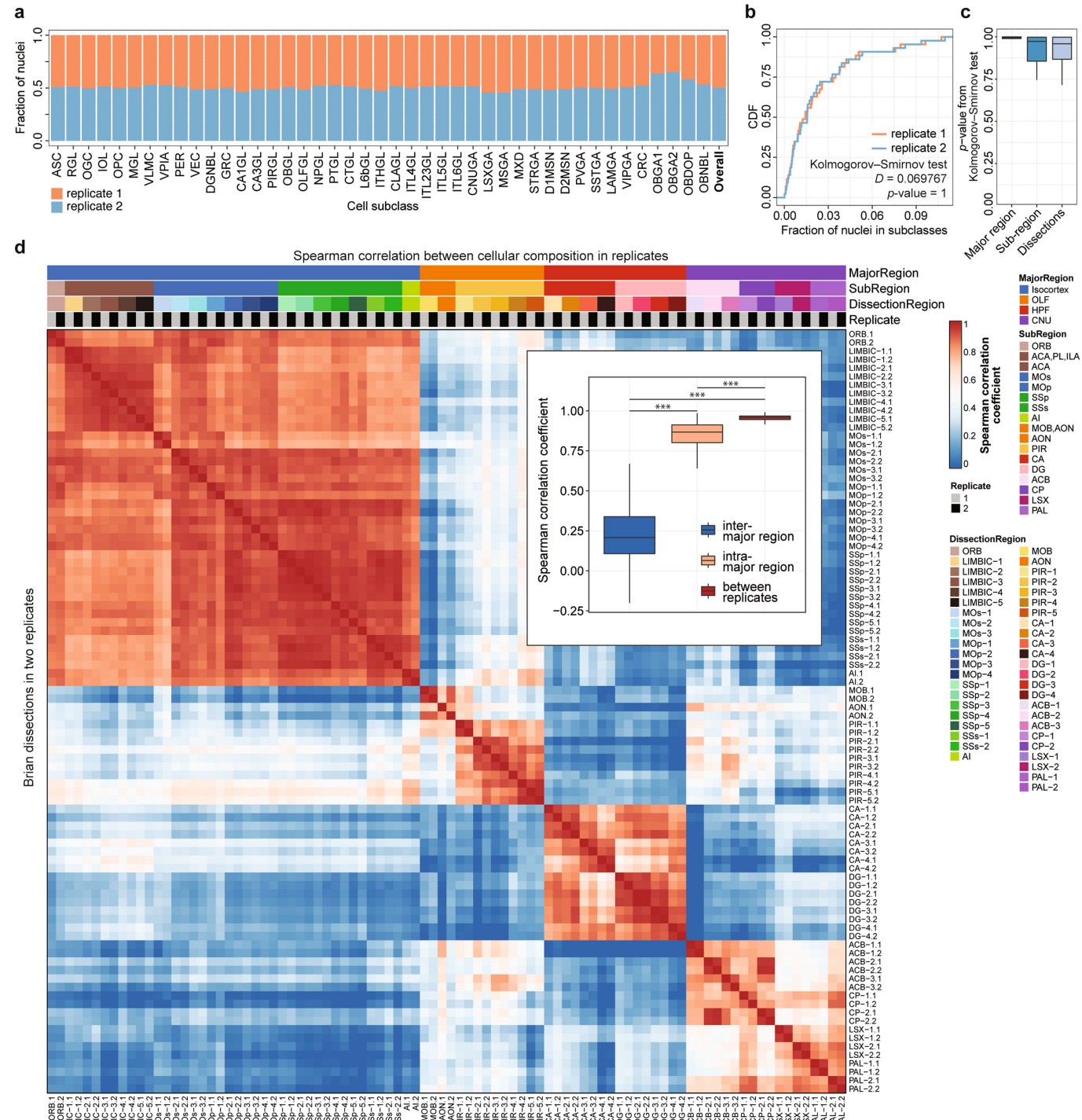

**Extended Data Fig. 5 | Reproducibility of the cell type composition of each brain region estimated from single-cell chromatin accessibility profiles. a**, Bar plot showing the fraction of nuclei from two biological replicates for each of the 43 subclasses of mouse cerebral cells discerned from snATAC-seq data. **b**, CDF plot showing the consistency of the estimated fractions of each subclass of cerebral cells between the two biological replicates. Kolmogorov–Smirnov test shows no significant difference between the biological replicates. **c**, Box plots of the *P* values of Kolmogorov–Smirnov tests illustrate consistent results between the two biological replicates for each subclass of cerebral cells

across major brain regions, sub-regions and brain dissections tested. **d**, Heat map showing the pairwise Spearman correlation coefficients of cell type composition between each replicate of brain dissections. The column and row names consist of two parts: brain region name and replicate label. For example, MOp-1.1 represents the replicate 1 of the first brain dissection of the primary motor cortex (MOp-1). The embedded box plot shows the distribution of Spearman correlation coefficients between two biological replicates, replicates from intra-major brain regions and inter-major brain regions. \*\*\**P* < 0.001, Wilcoxon rank-sum test. Box plots are stylized as in Fig. 2c.

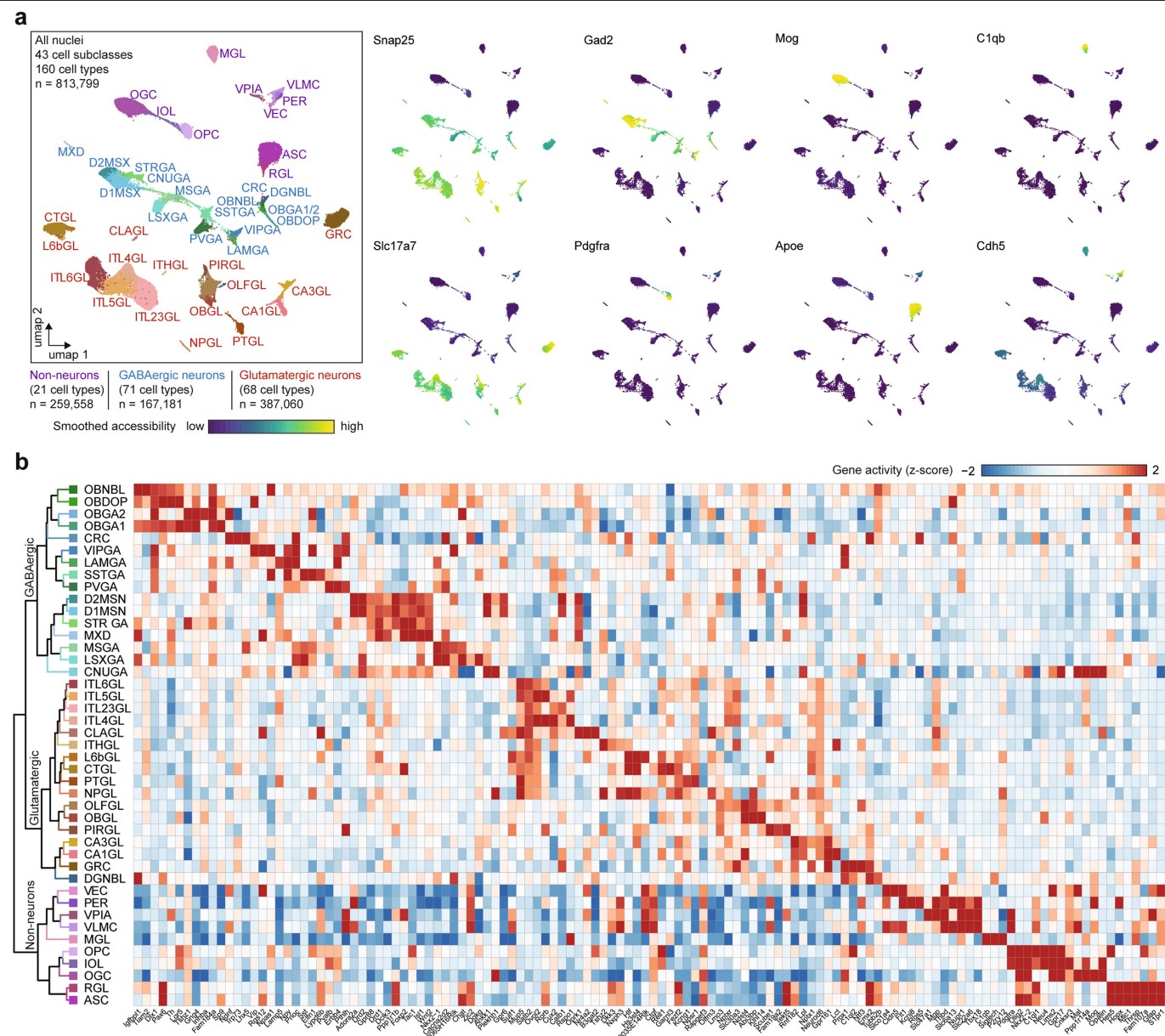

**Extended Data Fig. 6 | Maker genes used for annotation of different subclasses of cerebral cells. a**, Gene activity scores for marker genes used for subclass annotation. The UMAP[58] embedding from Fig. 1b is shown for reference. **b**, Heat map showing the gene activity scores of the marker genes (columns) used for subclass annotation across the 43 subclasses (rows). The colour gradient bar is at the top right.

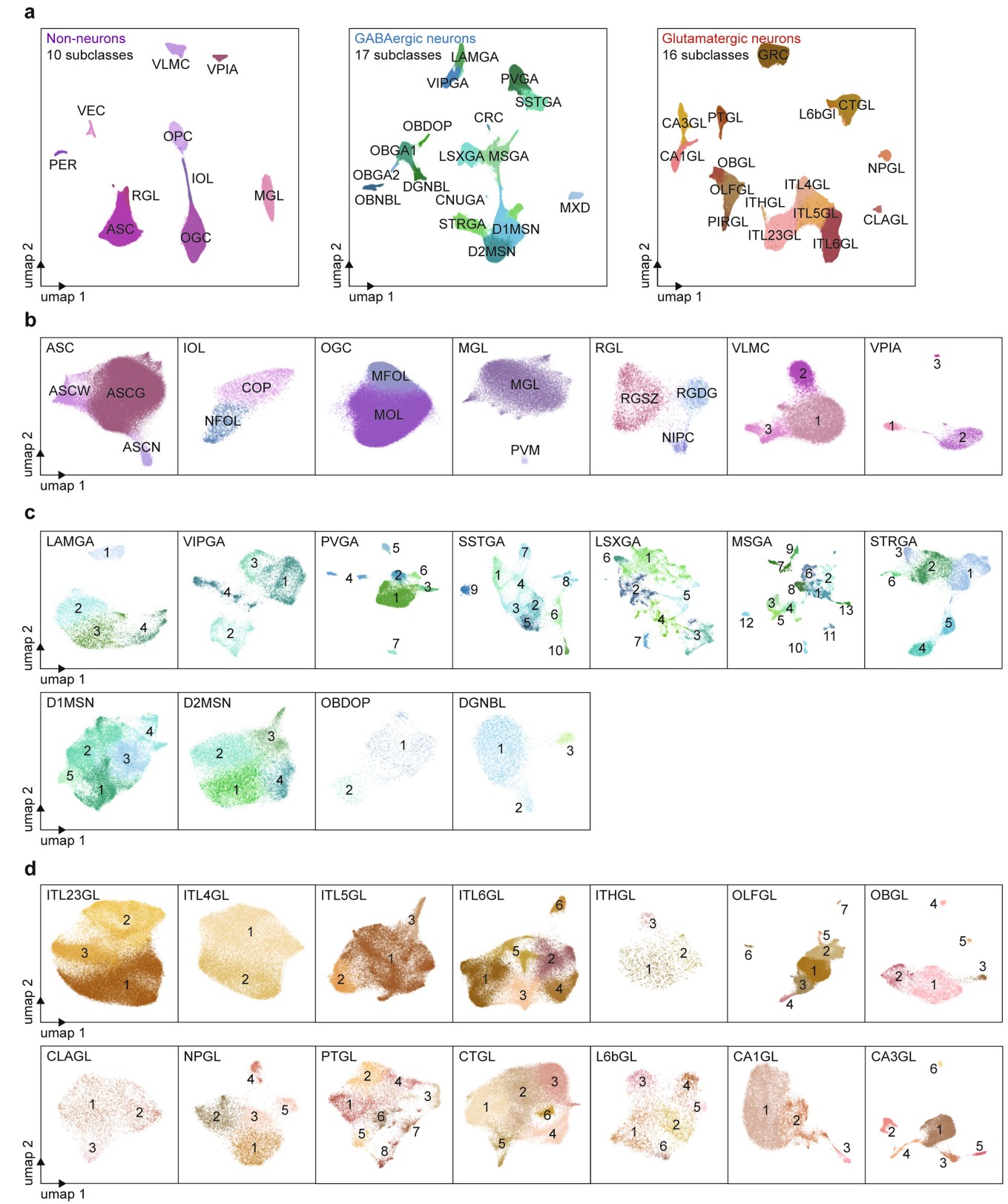

**Extended Data Fig. 7 | Iterative clustering identifies cell types for the subclasses of cerebral cells. a**, UMAP[58] embedding of the three classes of cerebral cells, namely non-neurons (left), GABAergic neurons (middle), and glutamatergic neurons (right). **b**–**d**, The subclasses of cerebral cells that can be further divided into cell types are shown for the non-neurons (**b**), GABAergic neurons (**c**) and glutamatergic neurons (**d**). UMAP embedding[58] is shown for each subclass, with the subclass label shown on the top left, and cell-type labels or cluster numbers on each cell type.

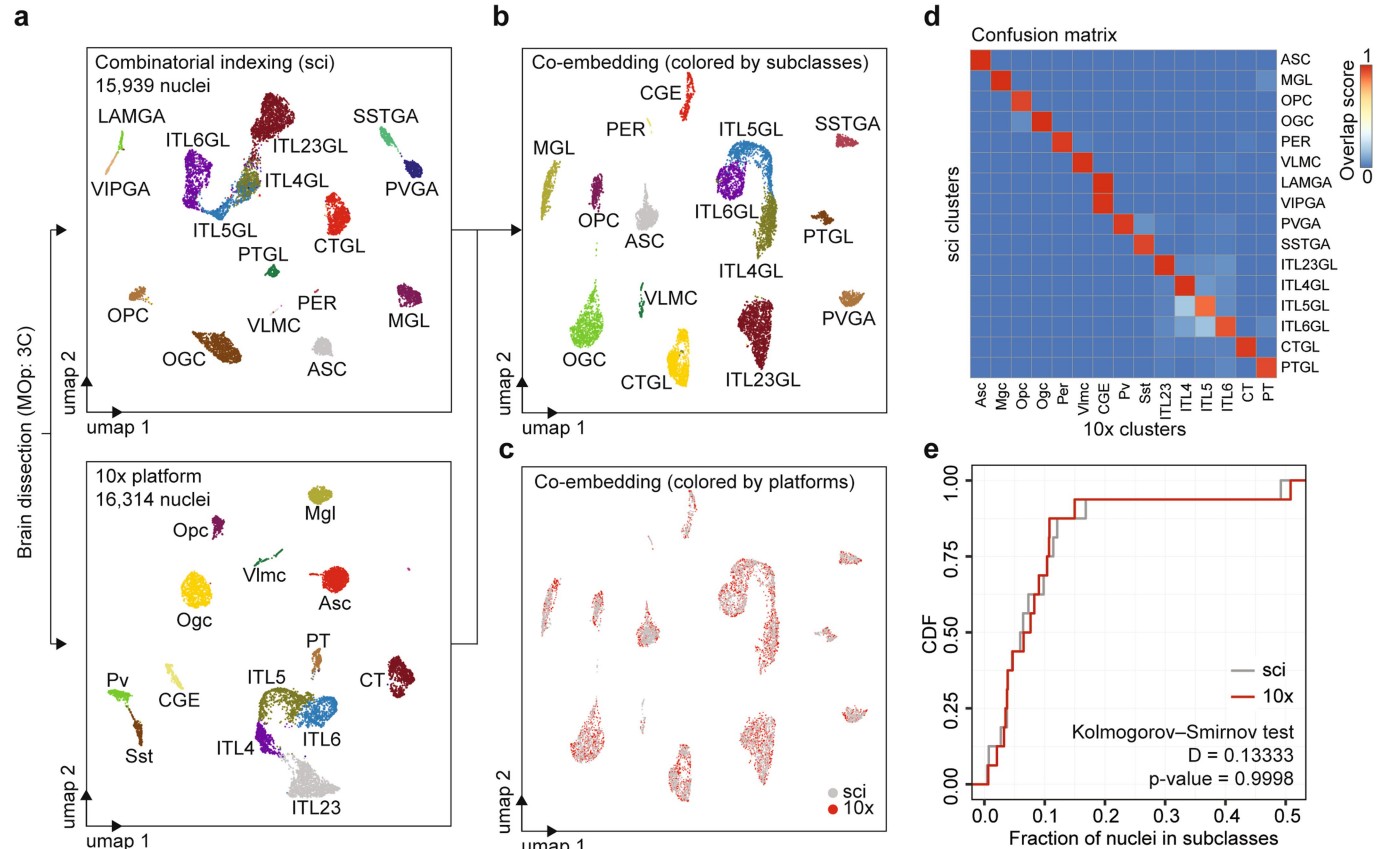

**Extended Data Fig. 8 | Comparison of cell type compositions in the mouse primary motor cortex determined by snATAC-seq using combinatorial indexing and droplet-based snATAC-seq (10x Genomics) platforms.** **a**, Individual clustering of snATAC-seq data generated using the single-cell combinatorial indexing (sci) and the droplet-based barcoding (10x) for the primary motor cortex (dissection: 3C). **b, c**, Co-embedding and joint clustering of snATAC-seq data generated from sci and 10x platforms. **b**, Dots are coloured by cell clusters. **c**, Dots are coloured by the experimental platforms (sci or 10x). **d**, Heat map illustrating the overlap between cell cluster annotations from both platforms. Rows show cell types from combinatorial barcoding; columns show cell types from the droplet-based platform. The overlap between the original clusters and the joint cluster was calculated (overlap score) and plotted on the heat map. **e**, CDF plot showing the fraction of nuclei in individual cell types for each platform.

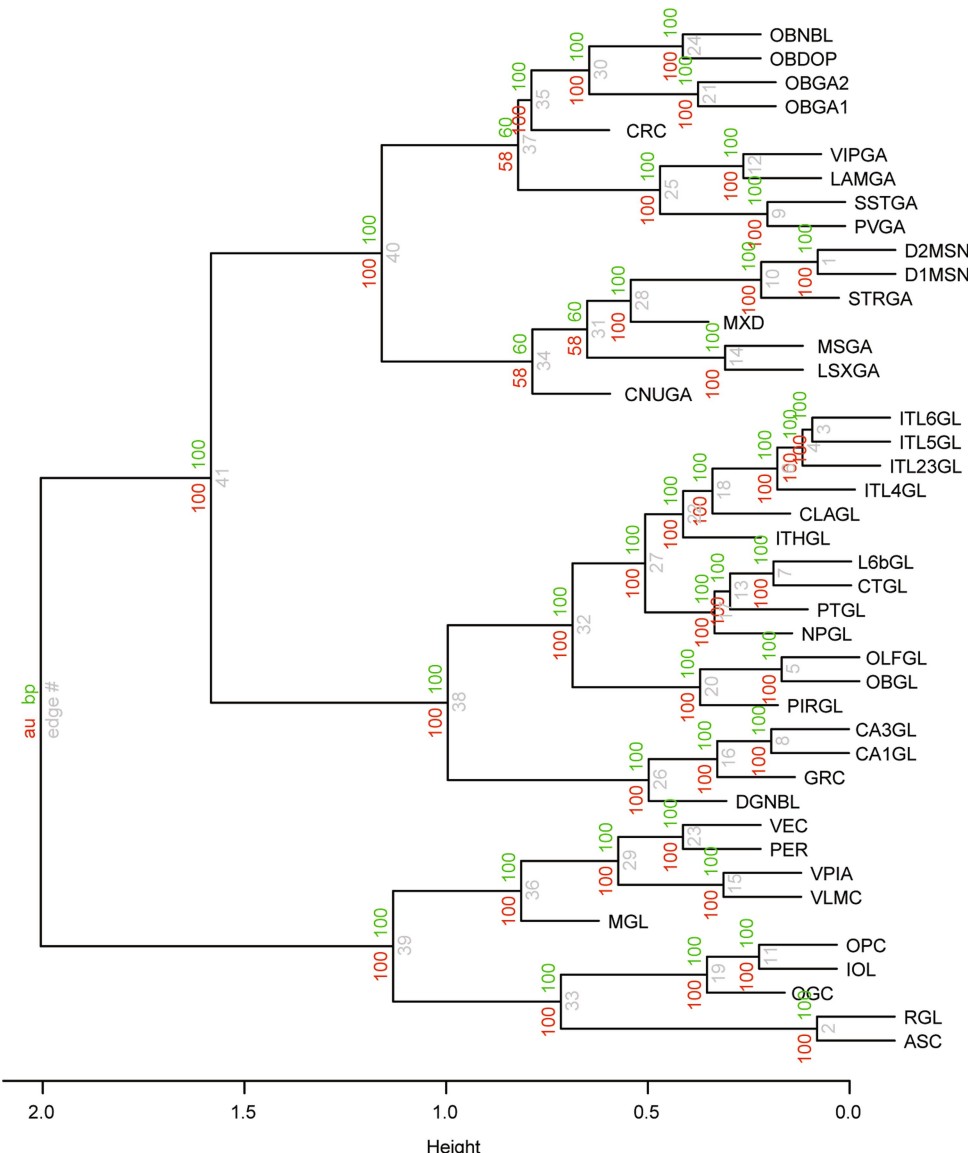

**Extended Data Fig. 9 | Hierarchical tree depicting the relationships between different subclasses of cerebral cells.** Dendrogram tree was constructed using 1,000 rounds of bootstrapping for the subclasses using R package pvclust[70]. Nodes are labelled in grey, approximately unbiased *P* values (in red) and bootstrap probability values (in green) are labelled at the shoulder of the nodes, respectively. A full list and a description of cell cluster labels are in Supplementary Table 3.

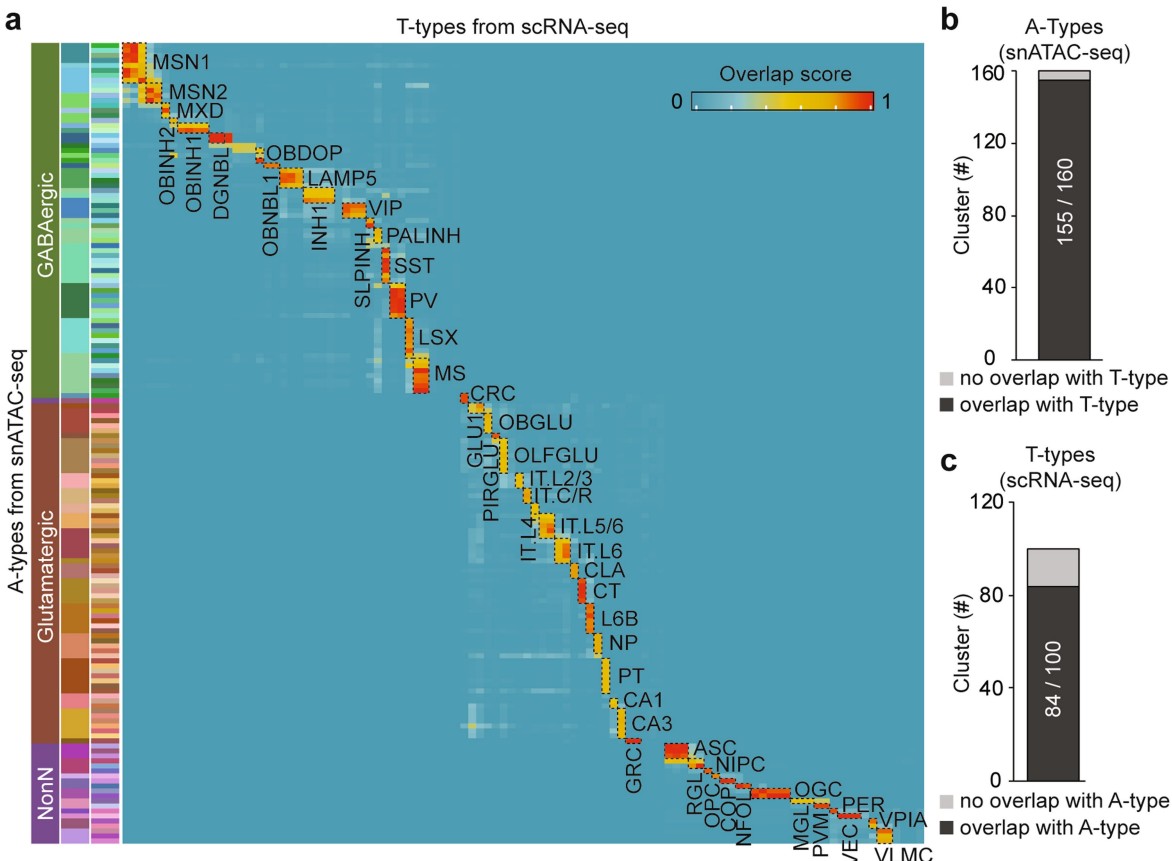

**Extended Data Fig. 10 | The chromatin-accessibility-based cell clustering matches transcriptomics-based cell taxonomy. a**, Heat map showing the similarity between A-type (accessibility) and T-type (transcriptomics) based cell cluster annotations. Each row represents an A-type cluster (a total of 160) and each column represents a T-type cluster (a total of 100). Similarity between original clusters and the joint cluster was calculated as the overlap score, which defined as the sum of minimal proportion of cells/nuclei in each cluster that overlapped within each co-embedding cluster. The overlap score varied from 0 to 1 and was plotted on the heatmap. Joint clusters with an overlap score of >0.5 are highlighted using black dashed line and labelled with RNA–ATAC joint cluster ID. For a full list of cell type labels and descriptions see Supplementary Table 4. **b, c**, Bar plots indicating the number of clusters that matched (dark grey) or did not match (light grey) clusters from the other modality. **b**, 155 out of 160 A-types had a matching T-type cluster. **c**, 84 out of 100 T-types had a matching A-type cluster.

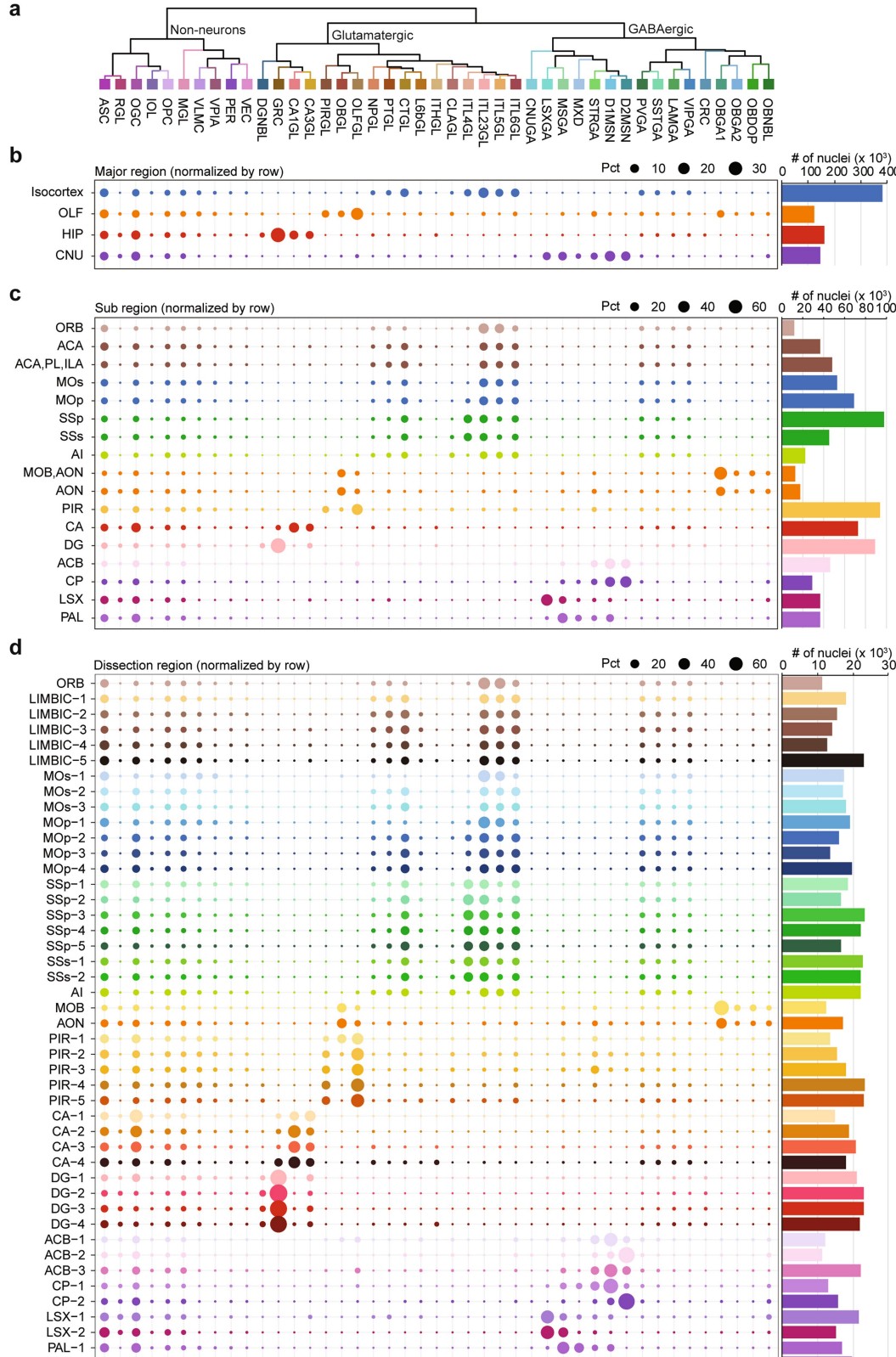

**Extended Data Fig. 11 | Cellular composition of brain regions, subregions and dissections. a**, Cluster dendrogram based on chromatin accessibility as in EDF8. **b**–**d**, Normalized percentages (pct) of each subclass in the four major regions (**b**), the subregions (**c**), and the dissected regions (**d**) are shown as different sized dots. The sizes of dots correspond to the percentage, and the colours of the dots indicate the major brain regions, subregions or dissections. Bar plots to the right show the total number of nuclei sampled for each region, subregion, or dissection.

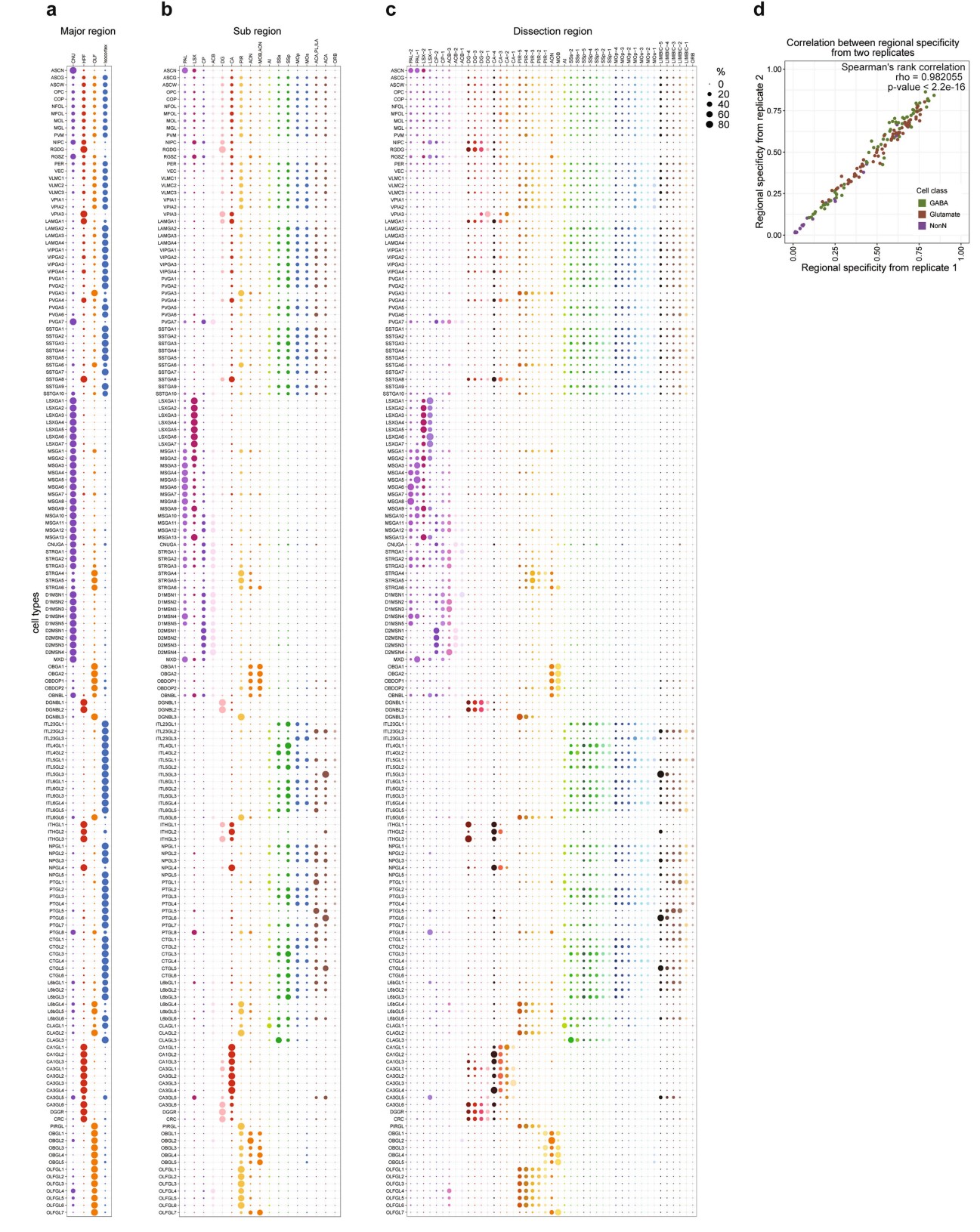

**Extended Data Fig. 12 | Distribution of each cerebral cell type across different brain regions, subregions and anatomical dissections.**
**a**–**c**, Normalized percentages of cells from each major brain region (**a**), subregion (**b**) and dissection (**c**) are shown as dots in cell types, with the sizes of the dots reflecting the percentages. **d**, Scatter plot showing the reproducibility of regional specificity of each cell type between two biological replicates of brain dissections.

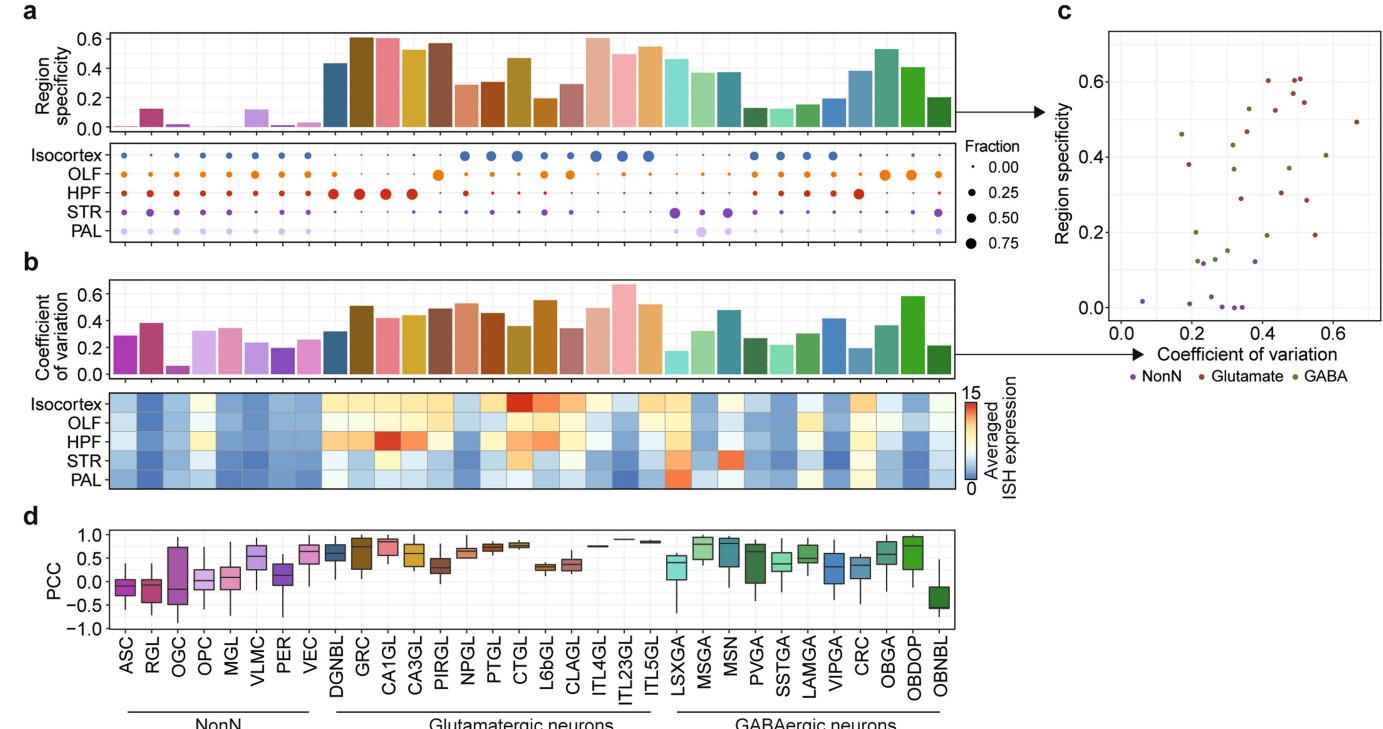

**Extended Data Fig. 13 | The regional specificity of cerebral cell types defined on the basis of snATAC-seq datasets is consistent with ISH patterns of cell-type-specific genes in the mouse brain. a**, Bar plots showing regional specificity of each cell type determined from relative contribution from five different brain regions, including isocortex, olfactory areas, hippocampal formation, striatum and pallidum. Dot plots showing percentages of each major brain region in the subclass of cerebral cells. The size of each dot reflects the relative contribution of the brain region as indicated by the legend to the right of the panel, and the colour of the dots indicates the brain region.
**b**, Bar plots showing the coefficients of variation calculated with the average normalized ISH signals of cell-type-specific marker genes for the corresponding cell subclasses across five brain regions. The heat map shows

the average normalized ISH signals of cell subclass-specific marker genes. The cell subclass-specific marker genes were identified for joint cell subclasses from RNA-ATAC integrative analysis (Extended Data Fig. 9). For a full list of ISH data of cell subclass-specific genes, see Supplementary Table 6. **c**, Scatter plot shows the correlation between the coefficients of variation of marker gene ISH signals and the regional specificity score calculated based on snATAC-seq data for 32 joint cell subclasses from RNA-ATAC integration analysis (Pearson correlation coefficients (PCC) = 0.55). **d**, Box plots show the Pearson correlation coefficients (PCC) calculated between cell composition across brain regions based on snATAC-seq data and the spatial distribution ISH signals of cell subclass-specific genes across the five main brain regions for each major brain cell subclass. Box plots are stylized as in Fig. 2c.

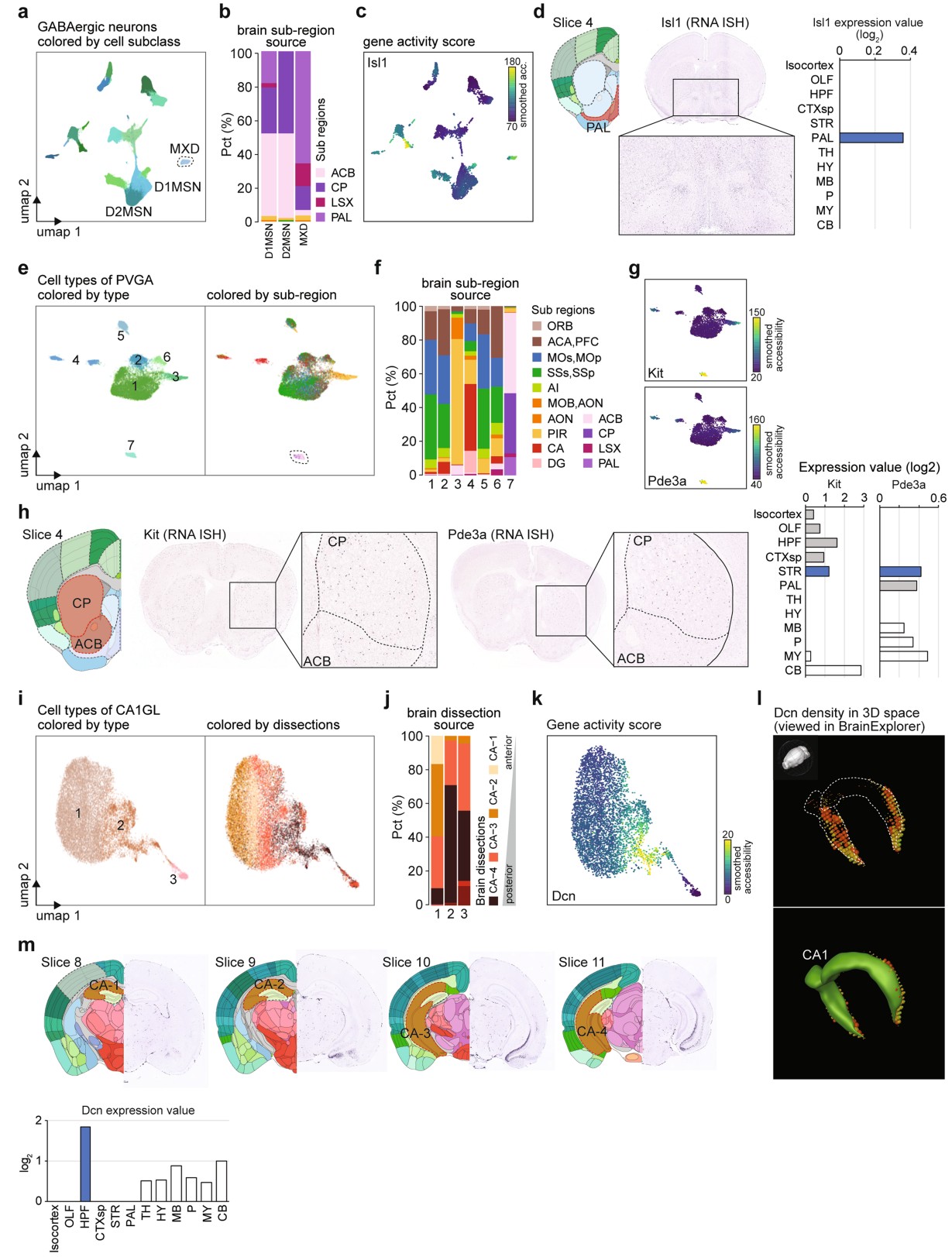

**Extended Data Fig. 14** | See next page for caption.

**Extended Data Fig. 14 | Brain region specificity of different subclasses and cell-types. a**, UMAP[58] embedding of the subclasses of GABAergic neurons. **b**, Sub-region composition of dopaminergic neurons, D1MSN and D2MSN, and MXD illustrates most MXD neurons were detected in the pallidum. **c**, Gene activity score for gene *Isl1* in GABAergic neurons predicts expression in MXD. **d**, RNA ISH and bar plot illustration of expression levels show the highest abundance of *Isl1* in the predicted region, pallidum (blue). Data and images were downloaded from © 2004 Allen Institute for Brain Science. Allen Mouse Brain Atlas. Available from: atlas.brain-map.org. **e**, UMAP[58] embedding of PVGA cell types. **f**, Sub-region composition for each PVGA cell type illustrates that majority of PVGA-7 were detected in nucleus accumbens and caudoputamen. **g**, Gene activity scores for *Kit* and *Pde3a* predict expression in one PVGA cell type (PVGA-7). **h**, RNA ISH and bar plot illustration of expression levels show expression of *Kit* and *Pde3a* in predicted regions, nucleus accumbens and caudoputamen. Predicted region in blue, other sampled regions in grey and non-sampled regions in white. **i**, UMAP[58] embedding of CA1 glutamatergic neurons (CA1GL) cell types. **j**, Dissection composition in each CA1GL cell type. **k**, Gene activity of gene *Dcn* in CA1GL cell types. **l**, Density of expression level of *Dcn* viewed in BrainExplorer (https://mouse.brain-map.org/static/brainexplorer) shows expression cornu ammonis field 1 (CA1). **m**, RNA ISH and bar plot show highest expression of *Dcn* CA1 and hippocampal formation. The predicted region is coloured in blue, other sampled regions in grey and non-sampled regions in white.

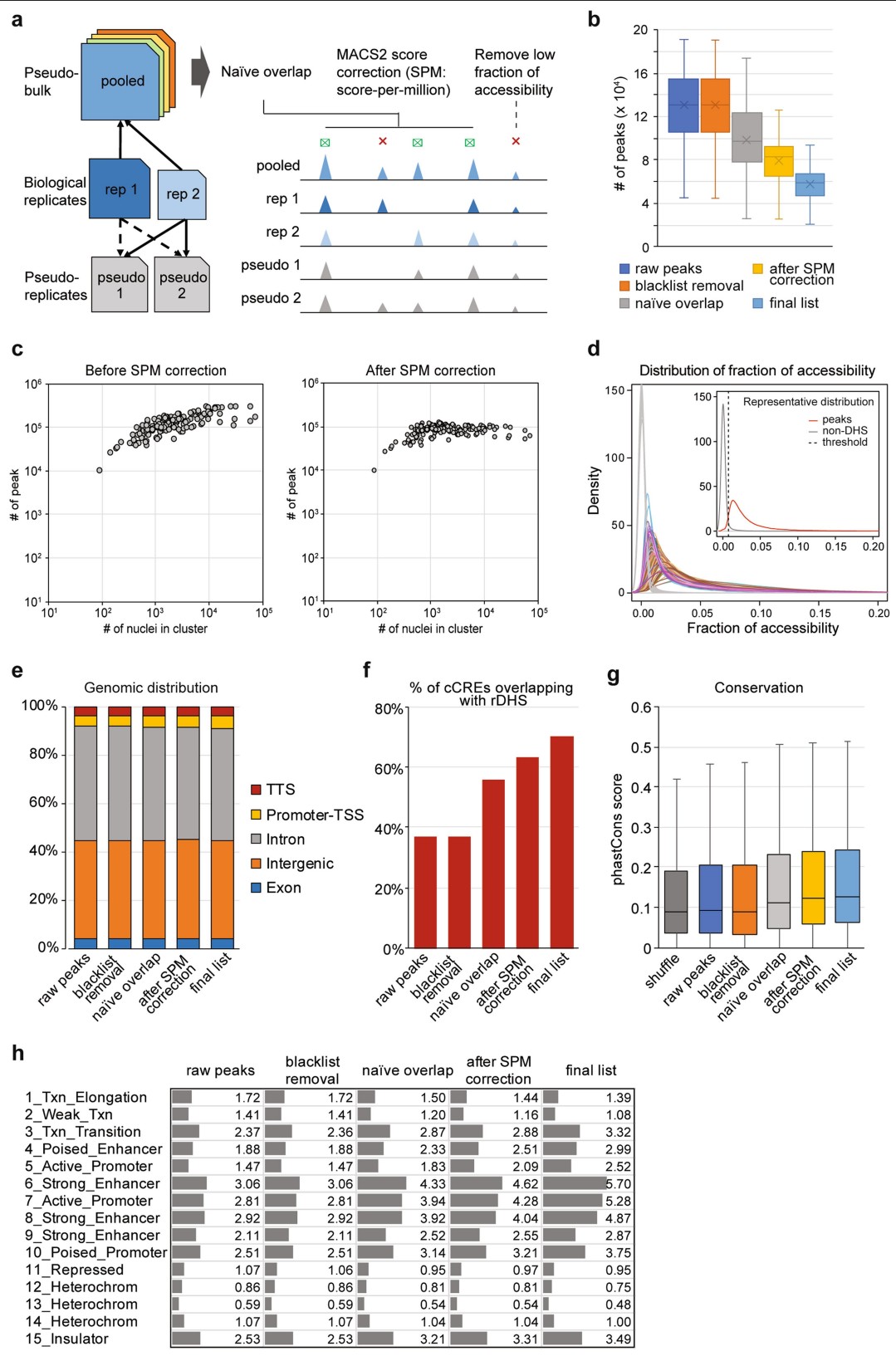

**Extended Data Fig. 15 | Statistics of peak calling from snATAC-seq data from each cell type. a**, Schematic diagram of peak calling and filtering pipeline. **b**, Number of peaks retained after each peak calling and filtering step. **c**, Scatter plots showing the relationship between the number of nuclei in each cluster and the number of peaks before score-per-million (SPM) correction (left) and after SPM correction (right). **d**, Density distribution plot showing the fraction of cells per cell type in which a peak was accessible and a corresponding background for each cell types. For each cell type, the background is defined as same number of non-DHS and non-peak regions randomly picked from genome. **e**, Fraction of different peak sets that overlap with annotated transcriptional start sites, introns, exons, transcriptional termination sites (TTS), and intergenic regions in the mouse genome. **f**, Fraction of different peak sets that overlap with DHSs[14]. **g**, Box plot showing sequence conservation in different peak sets and the control set. **h**, Enrichment analysis of different peak sets with a 15-state ChromHMM model in the mouse brain chromatin. All box plots are stylized as in Fig. 2c.

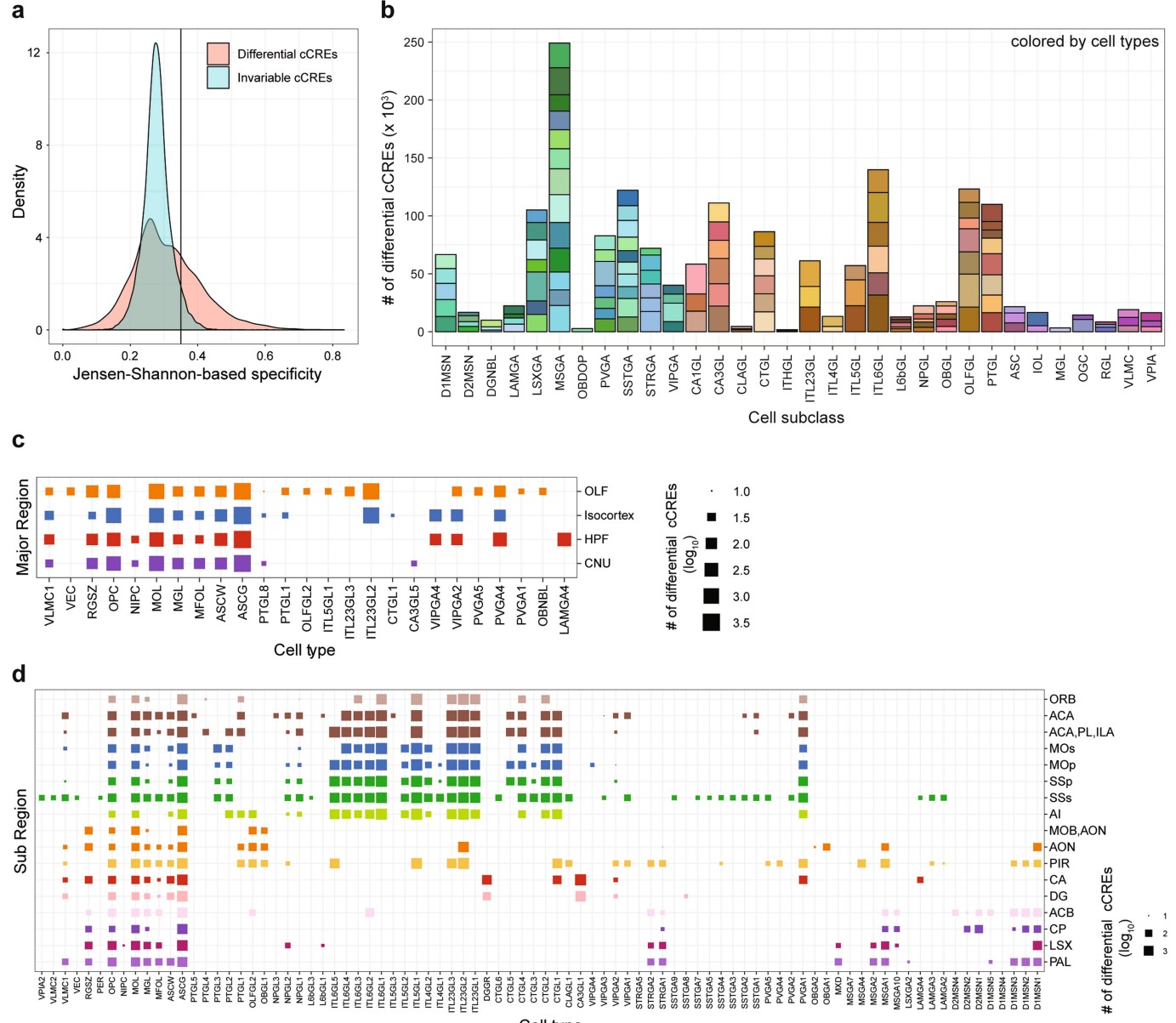

**Extended Data Fig. 16 | Differentially accessible cCREs across different cell types and brain regions. a**, Distribution plot of the Jensen–Shannon specificity scores of the differentially accessible cCREs and invariable cCREs. The vertical line shows the cutoff at FDR of 0.05. **b**, Stacked bar charts showing the number of differential cCREs between cell types for subclasses. **c**, A graph shows the numbers of differential cCREs in each major brain region. **d**, A graph shows the numbers of differential cCREs across subregions. The sizes of dots correspond to the number of differential cCREs (log₁₀-transformed) found in each cell type between brain regions, and the colours of the dots indicate the major brain regions in **c**, subregions in **d**.

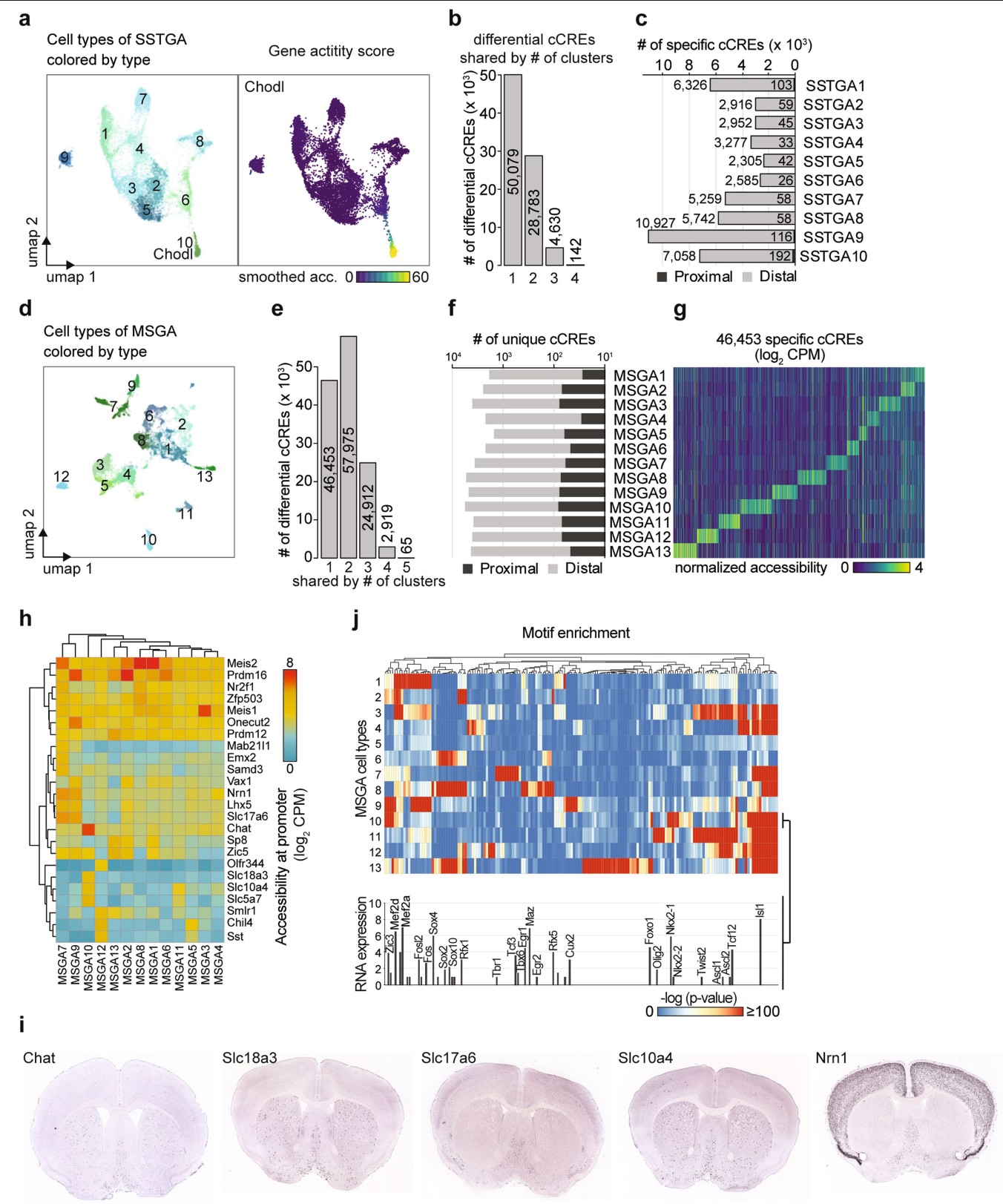

**Extended Data Fig. 17** | See next page for caption.

**Extended Data Fig. 17 | Variation of chromatin accessibility across different cell types within the subclasses of cerebral neurons. a**, UMAP[58] embedding and gene activity scores of *Chodl* show specificity for one SSTGA cell type. **b**, Bar chart showing the number of differentially accessible cCREs shared by 1, 2, 3 or 4 cell types of SSTGA. **c**, Bar chart showing the number of cell-type-specific cCREs, categorized by proximal (black) and distal regions (light grey). **d**, UMAP[58] embedding of MSGA cell types. **e**, Bar chart showing the number of differentially accessible cCREs shared by 1, 2, 3, 4 and 5 subtypes of MSGA. **f**, Number of specific accessible proximal and distal cCREs for each MSGA cell type. **g**, Heat map showing the normalized accessibility of the cell-type-specific cCREs across different MSGA cell types. **h**, Heat map showing the promoter accessibility at selected marker genes in each MSGA cell type. **i**, ISH data (https://portal.brain-map.org)[44] of the marker genes of each MSGA cell type. **j**, Motif enrichment of uniquely accessible cCREs and expression level of transcription factors in cholinergic neurons (MSGA10). The RNA expression of cholinergic neurons was downloaded from mouse brain atlas (http://mousebrain.org).

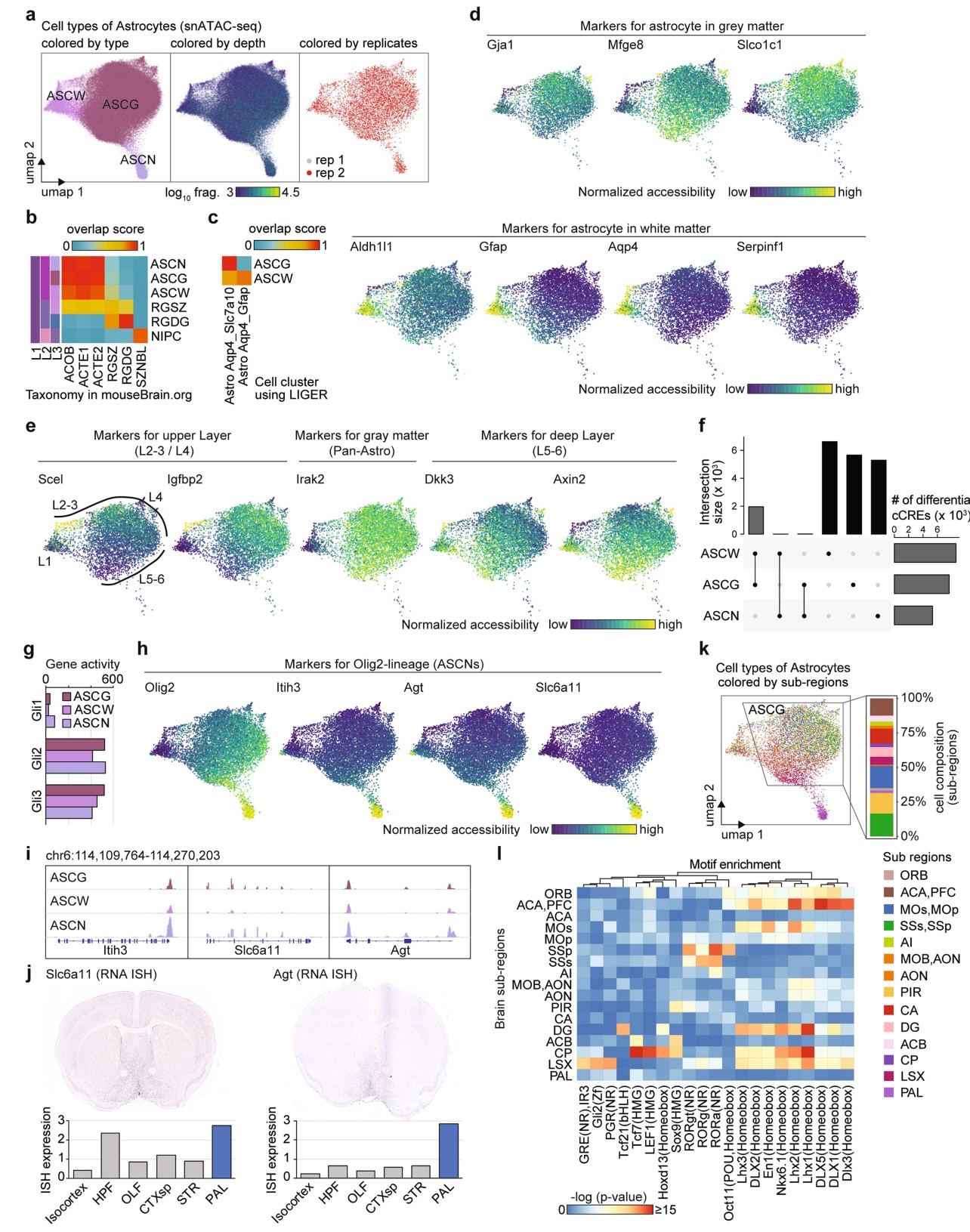

**Extended Data Fig. 18 |** See next page for caption.

**Extended Data Fig. 18 | Astrocyte cell types exhibit regional specificity and differential chromatin accessibility. a**, UMAP[58] embedding of astrocytes, coloured by cell type (left), fragment depth (middle), and replicate (right). **b**, Heat map illustrating the overlap between A-type and T-type cell cluster annotations. Each row represents a snATAC-seq subtype and each column represents scRNA-seq cluster[2] (http://mousebrain.org). The overlap between original clusters and the joint cluster was calculated (overlap score) and plotted on the heat map. **c**, Heat map showing overlap score with cell type defined by integration of multiple transcriptomic and epigenomic modalities[63]. **d**, Smoothed gene activity scores of marker genes for astrocytes from white and grey matter. **e**, Smoothed gene activity scores of representative cortical layer-specific marker genes for astrocyte. **f**, Upset plot showing intersections of differentially accessible cCREs between cell types. cCREs only presented in one cell type are defined as cell-type-specific cCREs. **g**, Gene activity score for GLI transcription factor family members in astrocyte cell types. **h**, Smoothed gene activity scores of *Oligo2, Itih3, Agt* and *Slc6a11* in astrocytes show stronger activity in ASCN. **i**, Genome browser tracks[60] of aggregate chromatin accessibility profiles for each astrocyte cell type at gene *Itih3*, *Slc6a11* and *Agt* locus. **j**, Views of ISH experiments from Allen Brain Atlas (atlas.brain-map.org) showing predominant expression of *Slc6a11* and *Agt* in the pallidum. Bar plot showing expression values from ISH experiments in brain structures. Data and images were downloaded from © 2004 Allen Institute for Brain Science[44]. Allen Mouse Brain Atlas. Available from: atlas. brain-map.org. **k**, UMAP embedding of astrocyte coloured by brain subregions. **l**, Motif enrichment in cCREs in ASCG with regional specificity.

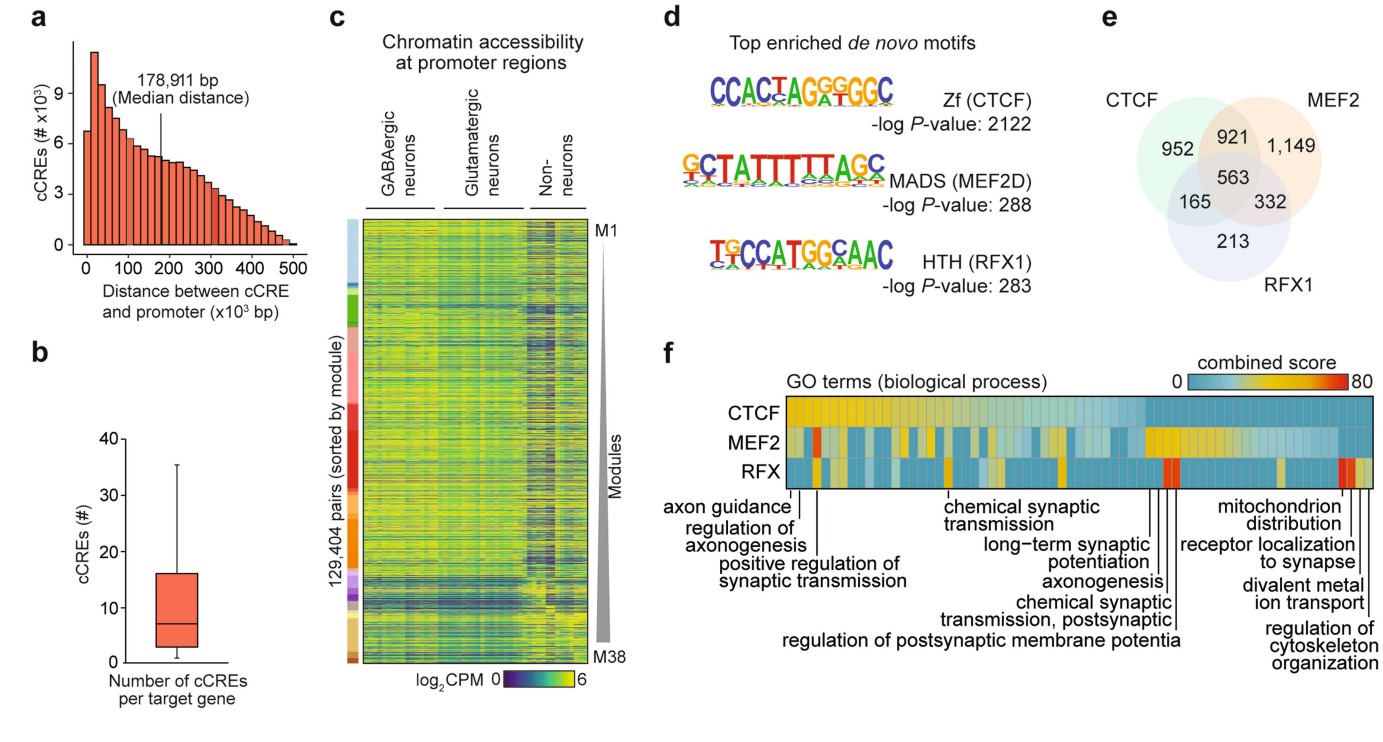

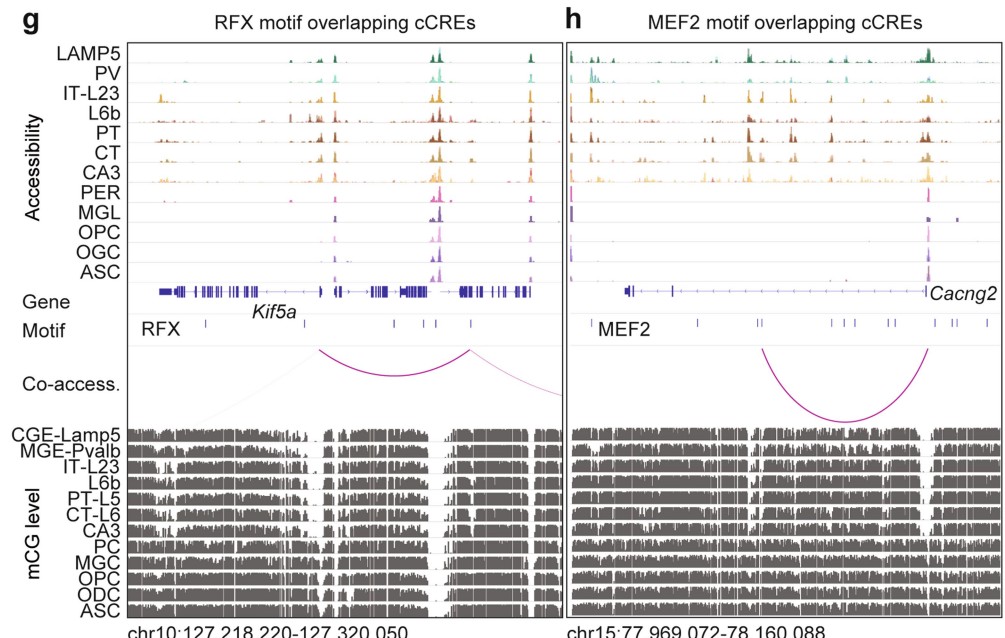

chr10:127,218,220-127,320,050 chr15:77,969,072-78,160,088

**Extended Data Fig. 19 | Characterization of predicted cCRE–target gene pairs. a**, Histogram illustrating the 1-D genomic distances between positively correlated distal cCRE and putative target gene promoters. **b**, Box plot showing that genes were linked with a median of 7 putative enhancers. Box plot is stylized as in Fig. 2c. **c**, Accessibility at promoter regions across RNA–ATAC joint cell types; order is as in Fig. 4c. **d**, Enrichment of sequence motifs for CTCF, MEF2 and RFX from de novo motif search in the putative enhancers of module M1 using HOMER[61]. For a full list, see Supplementary Table 21. **e**, Venn diagram illustrating the overlap of putative target genes of cCREs containing binding sites for MEF2, RFX and CTCF, respectively. **f**, Gene Ontology (GO) analysis of the putative target genes of each factor in module M1 was performed using Enrichr[83]. The combined score is the product of the computed P value using the Fisher exact test and the z-score of the deviation from the expected rank[83]. **g**, **h**, Examples of distal cCRE overlapping a RFX motif (**g**) or a MEF2 motif (**h**) and positively correlated putative target genes. Motifs were identified using de novo motif search in HOMER[61]. Genome browser tracks[60] showing chromatin accessibility, mCG methylation levels (see accompanying manuscript[29]) and positively correlated cCRE and gene pairs.

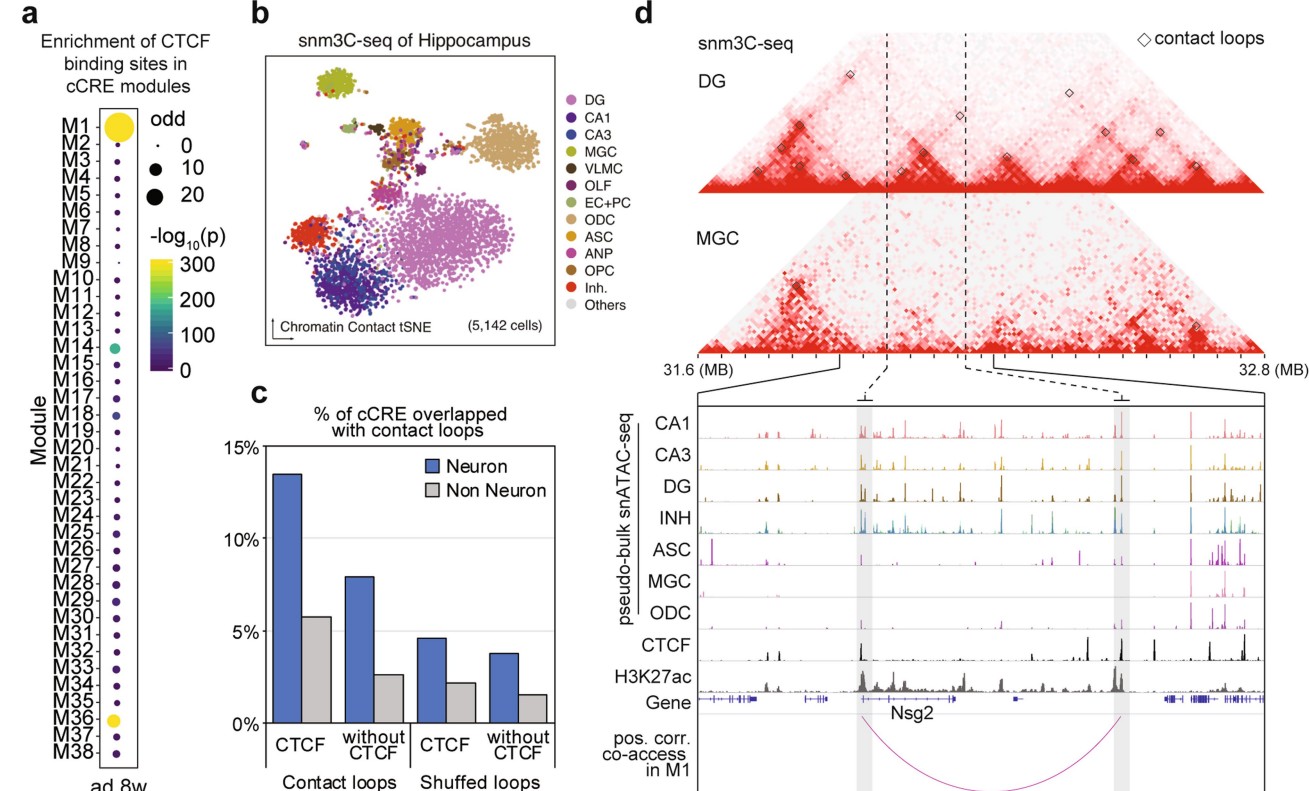

**Extended Data Fig. 20 | CTCF-associated cCRE–gene pairs. a**, Enrichment of CTCF binding sites from 8-week-old mouse forebrain[31] in cCRE modules. **b**, *t*-SNE of sn-m3C-seq cells (*n* = 5,142) coloured by clusters. This panel was replicated from the accompanying paper[29] for illustration purposes. **c**, Percentage of cCRE from module M1 with and without CTCF binding overlapping with contact loops identified from snm3C-seq in hippocampus[29]. Shuffled loops are generated by randomly flipping one of the anchors.

**d**, Genome browser track[60] view at the *Nsg2* locus as an example for CTCF dependent loops. Displayed are chromatin accessibility profiles for several neuronal and non-neuronal clusters, positively correlated cCRE–gene pairs, H3K27ac in postnatal day 0 (P0) mouse forebrain[108] and CTCF from in cortex from 8-week-old mouse[31]. The triangle heat map (top) shows chromatin contacts in DG neurons and MGC derived from snm3C-seq data (see accompanying manuscript[29]). DG-specific loops are highlighted by boxes.

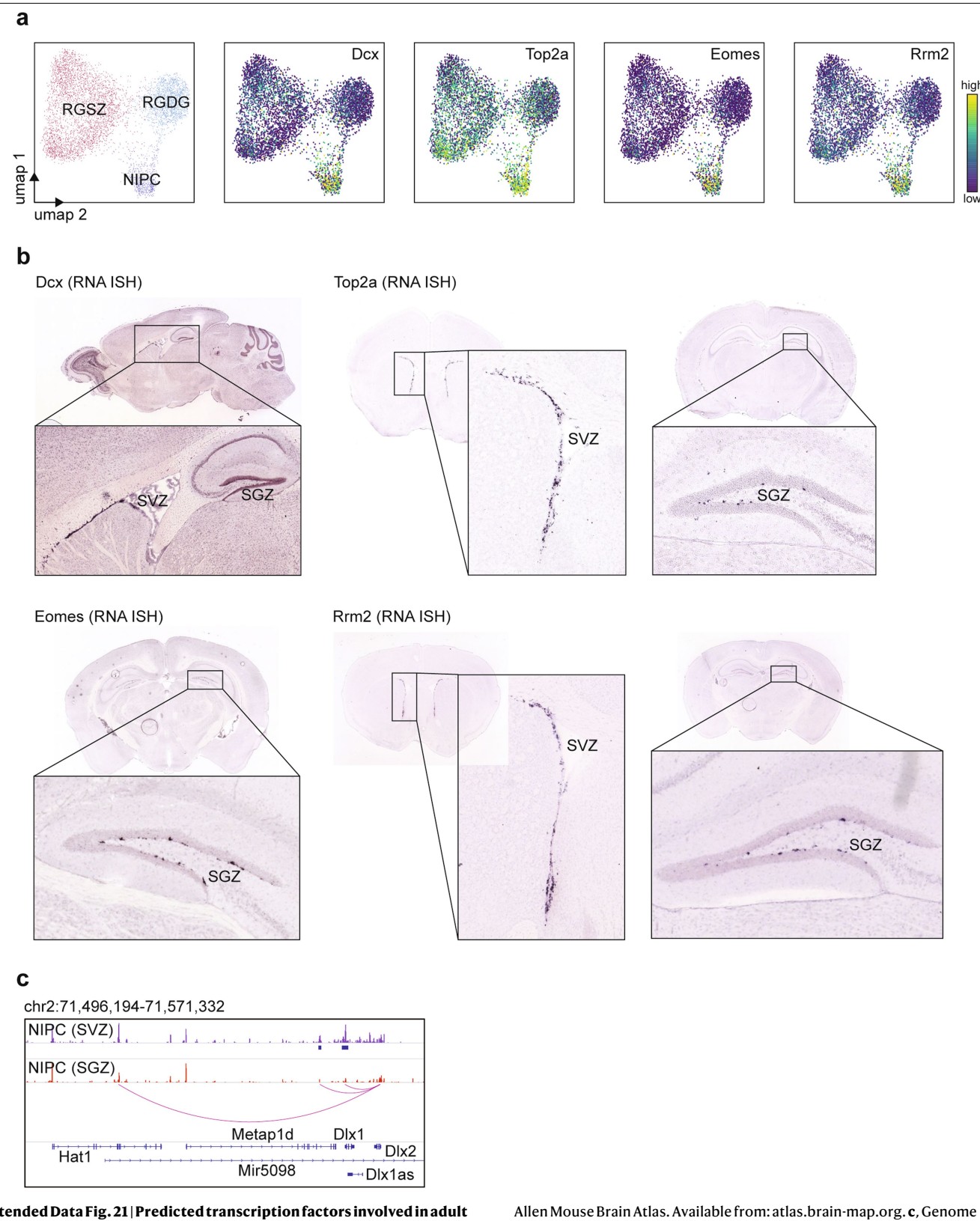

**Extended Data Fig. 21 | Predicted transcription factors involved in adult neurogenesis lineages. a**, Normalized accessibility at marker genes in NIPCs. **b**, RNA ISH shows expression levels of marker genes of NIPCs in the SGZ and SVZ. Images were downloaded from © 2004 Allen Institute for Brain Science[44]. Allen Mouse Brain Atlas. Available from: atlas.brain-map.org. **c**, Genome browser tracks[60] showing representative differentially accessible cCREs in NIPCs from SVZ.

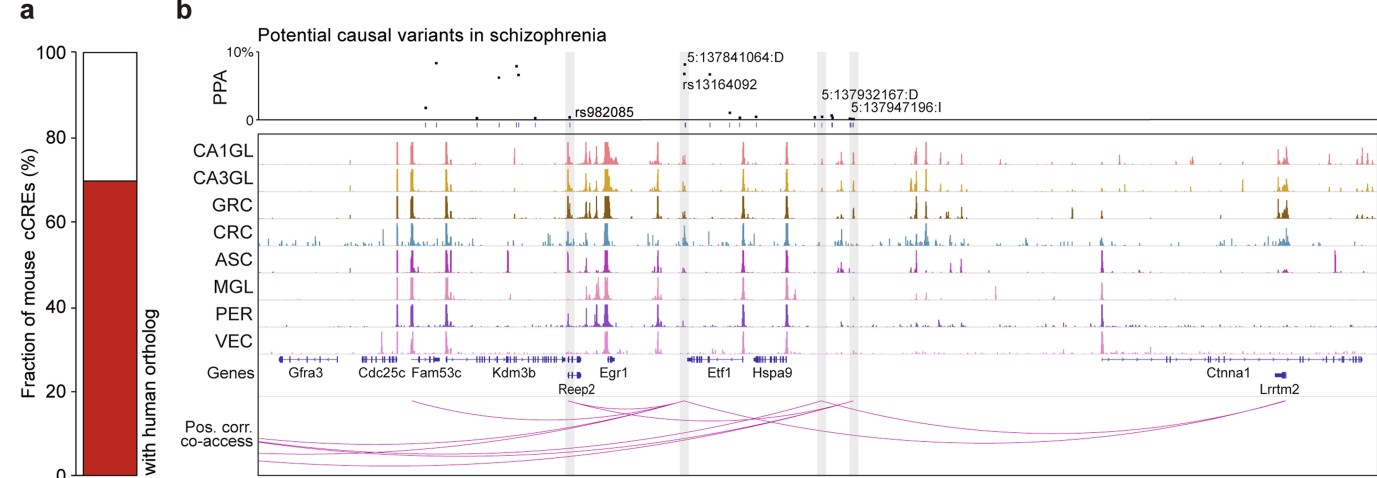

**Extended Data Fig. 22 | Mouse cerebral cCREs maps help to interpret potential casual risk variants of neurological diseases. a**, Bar plot showing the percentage of cCREs identified in the current study with homologous sequences in the human genome (using reciprocal homology search) (Methods). **b**, Chromatin accessibility profiles for several neuronal and non-neuronal cell types, and posterior probabilities of association (PPA) for potential casual variants surrounding the homologous region in the mouse genome to a schizophrenia-associated locus. Grey bars highlight cCREs overlapping potential causal variants. rsID and hg19 coordinates of overlapped variants are labelled. ':D' denotes that the alternative allele is a deletion; ':I' denotes an insertion. Predicted positively correlated cCRE–gene pairs are shown in red arcs.

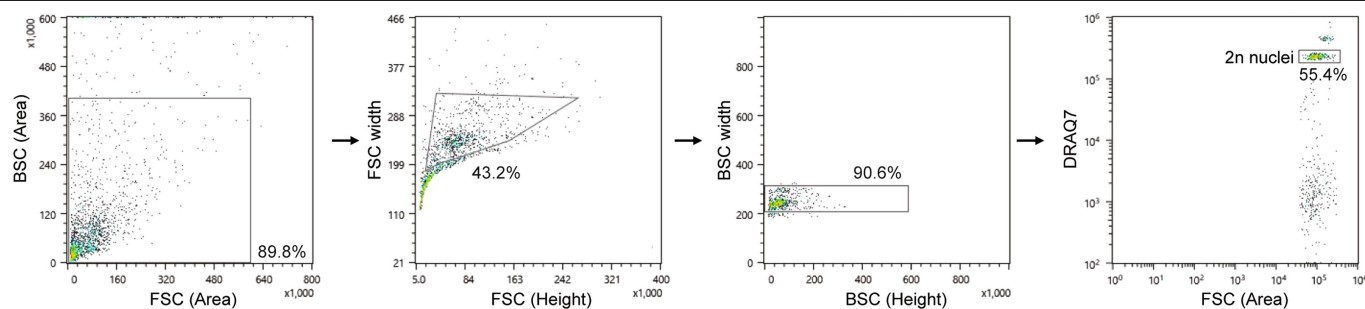

**Extended Data Fig. 23 | Nuclei gating strategy.** After tagmentation, nuclei were pooled and stained with DRAQ7. First, potential nuclei were identified using forward scatter (FSC) area and back scatter (BSC) area (left dot plot). Next, potential doublets were removed based on BSC and FSC signal width (two middle dot plots). Finally, 20 diploid nuclei (2*n*) were sorted into each well of eight 96-wells plates (right dot plot).

# natureresearch

| | |
|---|---|

# Reporting Summary

Nature Research wishes to improve the reproducibility of the work that we publish. This form provides structure for consistency and transparency in reporting. For further information on Nature Research policies, see Authors & Referees and the Editorial Policy Checklist.

## Statistics

For all statistical analyses, confirm that the following items are present in the figure legend, table legend, main text, or Methods section.

| n/a | Confirmed | |
|---|---|---|
| ☐ | ☒ | The exact sample size (*n*) for each experimental group/condition, given as a discrete number and unit of measurement |
| ☐ | ☒ | A statement on whether measurements were taken from distinct samples or whether the same sample was measured repeatedly |
| ☐ | ☒ | The statistical test(s) used AND whether they are one- or two-sided *Only common tests should be described solely by name; describe more complex techniques in the Methods section.* |
| ☒ | ☐ | A description of all covariates tested |
| ☐ | ☒ | A description of any assumptions or corrections, such as tests of normality and adjustment for multiple comparisons |
| ☐ | ☒ | A full description of the statistical parameters including central tendency (e.g. means) or other basic estimates (e.g. regression coefficient) AND variation (e.g. standard deviation) or associated estimates of uncertainty (e.g. confidence intervals) |
| ☐ | ☒ | For null hypothesis testing, the test statistic (e.g. $F$, $t$, $r$) with confidence intervals, effect sizes, degrees of freedom and $P$ value noted *Give P values as exact values whenever suitable.* |
| ☒ | ☐ | For Bayesian analysis, information on the choice of priors and Markov chain Monte Carlo settings |
| ☐ | ☒ | For hierarchical and complex designs, identification of the appropriate level for tests and full reporting of outcomes |
| ☐ | ☒ | Estimates of effect sizes (e.g. Cohen's *d*, Pearson's *r*), indicating how they were calculated |

*Our web collection on statistics for biologists contains articles on many of the points above.*

## Software and code

Policy information about availability of computer code

| | |
|---|---|
| Data collection | Sony Cell Sorter Software v2.1.2-5, Biomek Software 5.1 (library preparation) Illumina HiSeq2500 and HiSeq4000 instrument control software (sequencing) |
| Data analysis | Seurat (v3.1.2), sklearn (v0.22), bwa mem (v.0.7.17), Cicero (v1.3.4.5), HOMER (v4.11), liftOver (http://hgdownload.soe.ucsc.edu/admin/exe/linux.x86_64/), Enrichr (v2.1), GREAT (4.0.4), BEDTools (v2.25.0), ATACseqQC (1.8.5), pvclust (v2.2.0), MACS2 (v2.1.2), fitdistrplus (v1.0.14), LDSC (v1.0.1) https://github.com/yal054/snATACutils https://github.com/r3fang/SnapATAC |

For manuscripts utilizing custom algorithms or software that are central to the research but not yet described in published literature, software must be made available to editors/reviewers. We strongly encourage code deposition in a community repository (e.g. GitHub). See the Nature Research guidelines for submitting code & software for further information.

## Data

Policy information about availability of data

All manuscripts must include a data availability statement. This statement should provide the following information, where applicable:

- Accession codes, unique identifiers, or web links for publicly available datasets
- A list of figures that have associated raw data
- A description of any restrictions on data availability

Demultiplexed data can be accessed via the NEMO archive at: https://assets.nemoarchive.org/dat-wywv153
Processed data are available on our web portal and can be explored here: http://catlas.org/mousebrain
Additional data are available in the NCBI Gene Expression Omnibus (GEO) under accession number GEO173650 and upon request.

# Field-specific reporting

Please select the one below that is the best fit for your research. If you are not sure, read the appropriate sections before making your selection.

☒ Life sciences ☐ Behavioural & social sciences ☐ Ecological, evolutionary & environmental sciences

For a reference copy of the document with all sections, see nature.com/documents/nr-reporting-summary-flat.pdf

# Life sciences study design

All studies must disclose on these points even when the disclosure is negative.

| | |
|---|---|
| Sample size | No statistical methods were used to predetermine sample sizes. |
| Data exclusions | No samples were excluded.<br>For analysis only nuclei with > 1,000 reads/nucleus and transcriptional start site enrichment > 10 were selected.<br>Potential barcode collisions were excluded from analysis |
| Replication | Experiments were performed for 2 biological replicates for each dissected region |
| Randomization | There was no randomization of the samples |
| Blinding | Investigators were not blinded to the specimens being investigated. |

# Reporting for specific materials, systems and methods

We require information from authors about some types of materials, experimental systems and methods used in many studies. Here, indicate whether each material, system or method listed is relevant to your study. If you are not sure if a list item applies to your research, read the appropriate section before selecting a response.

### Materials & experimental systems

| n/a | Involved in the study |
|---|---|
| ☒ | ☐ Antibodies |
| ☒ | ☐ Eukaryotic cell lines |
| ☒ | ☐ Palaeontology |
| ☐ | ☒ Animals and other organisms |
| ☒ | ☐ Human research participants |
| ☒ | ☐ Clinical data |

### Methods

| n/a | Involved in the study |
|---|---|
| ☒ | ☐ ChIP-seq |
| ☐ | ☒ Flow cytometry |
| ☒ | ☐ MRI-based neuroimaging |

# Animals and other organisms

Policy information about studies involving animals; ARRIVE guidelines recommended for reporting animal research

| | |
|---|---|
| Laboratory animals | Adult C57BL/6J male mice were purchased from Jackson Laboratories. |
| Wild animals | No wild animals were used in this study |
| Field-collected samples | No filed-collected samples were used in this study |
| Ethics oversight | All experimental procedures using live animals were approved by the SALK Institute Animal Care and Use Committee under protocol number 18-00006. |

Note that full information on the approval of the study protocol must also be provided in the manuscript.

# Flow Cytometry

## Plots

Confirm that:

☒ The axis labels state the marker and fluorochrome used (e.g. CD4-FITC).

☒ The axis scales are clearly visible. Include numbers along axes only for bottom left plot of group (a 'group' is an analysis of identical markers).

☒ All plots are contour plots with outliers or pseudocolor plots.

☒ A numerical value for number of cells or percentage (with statistics) is provided.

## Methodology

| | |
|---|---|
| Sample preparation | Nuclei were stained with DRAQ7 (#7406, Cell Signaling) |
| Instrument | Sony SH800 |
| Software | SH800S software |
| Cell population abundance | NA |
| Gating strategy | Potential nuclei were first identified using FSC-Area and BSC-Area. Next doublets were removed based on BSC and FSC signal width. DRAQQ7 postive nuclei with 2n count were sorted |

☒ Tick this box to confirm that a figure exemplifying the gating strategy is provided in the Supplementary Information.

