## [Peer Review File · Nature]

Manuscript Title: An Atlas of Gene Regulatory Elements in Adult Mouse Cerebrum

Reviewer Comments & Author Rebuttals

Reviewer Reports on the Initial Version:

Referees' comments:

Referee #1 (Remarks to the Author):

Li et al. applied sn-ATAC-seq to different mouse brain regions, collecting single cell chromatin accessibility profiles for ~800K nuclei. The size of the dataset allows the authors to identify many distinct cell type clusters. They are able to convincingly match their cell type clusters to cell types identified by published sc-RNA-seq studies and analyze e.g. transcriptional regulators or cell type-specific heritability enrichment.

Due to its size the dataset is a valuable resource for the field. The study appears well designed and implemented, both experimentally and computationally. However, it would be nice to see a significant biological finding to justify publication in Nature, since neither the choice of profiled tissue and time point (see PMID: 30078704), the technique, the analysis tools nor the presented findings are novel (RFX has been previously linked to neuronal and synaptic functions; e.g. PMID: 32006715, PMID: 24924197, PMID: 18843046). As part of this, the authors could further validate some of the claims in the paper (e.g. CTCF's role at enhancers) or perform a more in-depth analysis on some of the data, e.g. looking at the differences between cells from the same cell type but different brain regions.

In addition, the following aspects should be addressed:

- The authors should provide more quantitative instead of qualitative statements. For example: What fraction of reads fall in peaks? How many cells were removed upon applying their read count cutoff? How many cells were removed in the scrublet procedure? How many mice were processed? How many false positive peaks were detected? Etc.
- There is a very stark separation for the neuronal cell types based on dissection region (Fig. 1C). Can it be excluded that this is some way due to a batch effect? It is not clear from the methods how the samples were processed: were different regions processed at the same time and were replicates processed at different times? It just says that 8 96-well plates were processed (15k nuclei at 100% recovery) – does this mean 8 96-well plates per region and replicate?
- It is not clear how the cell types were annotated. If this was only done based on marker gene accessibility, which marker genes were used? Only the 14 genes in Fig. 1D? It seems hard to annotate 43 cell types based on accessibility at 14 genes.
- How do the false positive peaks perform for the metrics shown in Figure 3A-D?

Minor points:

- There is no figure showing the subclustering results of the 160 sub cell types. Cell types can be subclustered into almost any number of subclusters, so without seeing the clustering results and how resolved they are, this number is pretty meaningless.
- The colors in the top panels of Figure 2A-C are hard to distinguish
- In Figure 2E,F which cutoff was applied to count as overlap?

Referee #2 (Remarks to the Author):

The manuscript describes a snATAC-seq analysis of the mouse cortex and adjacent brain regions and integrate the analysis of these data with previously published scRNA-seq. The authors sequence data from more than 800,000 nuclei, probably a record in this type of experiment, and identify 160 different cell types. Analysis of accessible regions allows the characterization of almost half a million enhancers, some of which associate with human mental disorders.

The experiments described here represent a technical tour-de-force. The data is of high quality and the computational analyses are carefully done. However, there is not much in terms of mechanistic insights gained for the data. This manuscript will be a good resource for those interested in mouse neurobiology and using mouse models to study mental diseases. I leave it up to the editors to decide if this type of information is sufficiently significant for publication in Nature.

Author Rebuttals to Initial Comments:

We thank the reviewers for their helpful comments. While both reviewers were positive on the quality and resource value of the datasets that we report in the current manuscript, they also commented on the lack of biological insight as a significant weakness. To address their concerns, we have carried out new analyses and made substantial revisions to the manuscript, which in our view have added considerably new dimensions to the study and further clarified the value of our atlas of *cis*-regulatory elements in the mouse cerebrum. Briefly, the following new sections were added in the revised manuscript:

1. Regional specificity of different brain cell types

Taking advantage of our high-resolution brain dissections we examined the regional specificity in chromatin accessibility of each major cell type and sub-type in the mouse cerebrum. We showed that most brain cell types exhibit strong regional specificity in their distribution. While this is expected for glutamatergic neurons, interestingly a large number of GABAergic neurons and even some glial cell types display regional specificity. This finding is further validated using *in-situ* hybridization (ISH) data from the Allen Brain Atlas (ABA).

We have added the following items to describe these additional findings:

Main text in lines 212-265

Figure 1f-g

Extended Data Figure 10, 11, 12

2. Open chromatin regions characterize distinct sub-types of neuronal and glial cells

We performed additional analysis to identify open chromatin regions and candidate *cis*-regulatory elements (cCREs) that define the regional specificity of neuronal and non-neuronal subtypes. We explored the two most diverse inhibitory neurons types, *Sst*-positive GABAergic neurons (SSTGA, 10 subtypes) and GABAergic neurons in the medial septal nucleus (MSGNA, 13 subtypes) in detail. We defined differentially accessible cCREs between the sub-types within each class. Motif analysis of these sub-type specific open chromatin uncovered potential TFs that could regulate cell-type specific gene expression in these rare neuronal sub-types.

Additionally, in order to understand the gene regulatory program in an astrocyte sub-type that displayed strong regional specificity, we performed transcription factor motif enrichment analysis. Open chromatin regions in these astrocytes were enriched for candidate transcription factors, in particular the hedgehog signaling effector *GLI*, which was suggested previously to play a role in maintaining astrocyte function.

We have added the following items to describe the findings:

Main text in lines 315-370

Figure 3

Extended Data Figure 14, 15, 16

Supplementary Table 11, 12, 13,14

3. Integrative analysis of neurogenesis in the adult mouse brain

We identified the neuronal intermediate progenitor cell (NIPC) populations in both the subgranular zone (SGZ) and the subventricular zone (SVZ), which give rise to excitatory and inhibitory neurons, respectively. We performed integrative analysis to dissect the regulatory programs in these cells underlying the differential cell fate, and identified known and novel lineage-driving transcription factor candidates by integrating motif enrichment in open chromatin

regions with gene expression. Differential accessible regions were identified and associated with expression of key regulators in neuronal intermediate progenitor cells (NIPCs) in both neurogenic zones.

We have added the following items in the revision:

Main text in lines 492-524

Figure 5

Extended Data Figure 19

Supplementary Table 23

4. Fine mapping analysis to reveal the likely causal risk variants of neuronal traits and psychiatric disorders.

To illustrate the utilities of the cell-type resolved maps of candidate cis regulatory elements (cCRE) in the mouse cerebrum, we carried out detailed analysis of the risk variants associated with several psychiatric disorders. In particular, we focused on schizophrenia (SCZ), the trait that showed substantial association with cCREs identified in a number of neuronal types. We found that ~26.9% of the likely causal variants of SCZ mapped through the fine-mapping analysis fall within the cCREs defined in this study, providing strong evidence for the model that dysregulation of neuronal gene expression contributes to SCZ.

We have added the following items in the revision:

Main text in lines 568-586

Figure 6c

Responses to specific reviewer comments are detailed in bold below.

Reviewer 1

Li et al. applied sn-ATAC-seq to different mouse brain regions, collecting single cell chromatin accessibility profiles for ~800K nuclei. The size of the dataset allows the authors to identify many distinct cell type clusters. They are able to convincingly match their cell type clusters to cell types identified by published sc-RNA-seq studies and analyze e.g. transcriptional regulators or cell type-specific heritability enrichment. Due to its size the dataset is a valuable resource for the field. The study appears well designed and implemented, both experimentally and computationally. However, it would be nice to see a significant biological finding to justify publication in Nature, since neither the choice of profiled tissue and time point (see PMID: 30078704), the technique, the analysis tools nor the presented findings are novel (RFX has been previously linked to neuronal and synaptic functions; e.g. PMID: 32006715, PMID: 24924197, PMID: 18843046). As part of this, the authors could further validate some of the claims in the paper (e.g. CTCF's role at enhancers) or perform a more in-depth analysis on some of the data, e.g. looking at the differences between cells from the same cell type but different brain regions.

Response:

We thank the reviewer for the positive remarks on the experimental design, the computational analysis and the value of the resource that we report. We are also very grateful for the constructive comments on further improvement with regard to the biological insights gained from the current study. In response to the reviewer's concerns and suggestions, we performed substantial additional analyses to describe the regional specificity of different neuronal and glial cell types, characterize the transcriptional regulatory sequencing underlying the specific subtypes of neurons and glial cells, interpret the likely causal variants of several psychiatric disorders including schizophrenia, and dissect gene regulatory programs during adult neurogenesis and oligodendrocyte maturation. As outlined in the beginning of this response letter, these new findings have been added to the revised manuscript.

Major point 1

In addition, the following aspects should be addressed:

- The authors should provide more quantitative instead of qualitative statements. For example: What fraction of reads fall in peaks? How many cells were removed upon applying their read count cutoff? How many cells were removed in the scrublet procedure? How many mice were processed? How many false positive peaks were detected? Etc.

Response:

Thank you for this suggestion! We have revised the texts thoroughly to add more quantitative statements wherever appropriate. We also reorganized Extended Data Figure 2, 3, 5, and 13, to provide more statistics for experimental samples as well as quality metrics for nuclei and peaks.

What fraction of reads fall in peaks?

Response:

In median, 77.6% of unique mapped fragments fall in cCREs. We are providing this additional info here as Fig R1 and in the manuscript as new Extended Data Figure 5.

Figure R1: Detailed statistics for cell types

a Violin plots show log₁₀ transformed unique fragments across major types. **b** Violin plots show TSS enrichment across major types.

How many cells were removed upon applying their read count cutoff? How many cells were removed in the scrublet procedure?

Response:

We applied multiple filtering steps during the quality control of nuclei. In total, we profiled 1,270,311 nuclei from all brain samples (Figure R2). In the first step, 28.7% of nuclei were removed as they failed to pass initial quality control (>1,000 fragments per nucleus and/or TSS enrichment > 10). Next, 6.2% of the nuclei were removed, due to likely barcode collision events, detected using a procedure modified from Scrublet¹. Finally, ~2% of nuclei were further removed during the clustering process by manual inspection since they form tiny clusters (less than 1% in each round of clustering) that might be noise.

We added the following sentence in the main text:

“Altogether, 28.7% nuclei of low-quality and 6.2% potential doublets were removed from further analysis (Extended Data Figure 3c).”

Figure R2: Number of nuclei retained after each step of quality control.

How many mice were processed?

Response:

For each dissected brain regions, 3-31 mice were processed, we have added this information in the sample metadata table in Supplementary Table 1, Column E. We added more details about tissue preparation and nuclei isolation in the methods part:

“Regions were pooled from 3-31 animals of the same sex to obtain enough nuclei for single nucleus ATAC-seq for each biological replica, and two biological replicas were processed for each dissection region.”

How many false positive peaks were detected?

Response:

In Extended Data Figure 13 (Fig R3 in this response letter), we illustrate the strategy that we used to identify candidate *cis*-regulatory elements (cCREs). We first applied MACS2 on aggregated profiles of each cluster to identify open chromatin regions. We selected 881,611 peaks from different cell types that were detected in both biological replicates or pseudo-replicates. We found that read depth or cluster size can affect MACS2 peak calling scores due to the nature of the Poisson distribution test in MACS2, which will introduce bias when we apply a constant cutoff. Ideally, we would perform a reads-in-peaks normalization between clusters, but in practice, this type of normalization is not possible because we don't know how many peaks are accessible in each cell cluster. For these reasons, we used “score per million” (SPM) to correct this issue. Finally, we selected regions that were identified as open chromatin in a significant fraction of the cells in each sub-type. By applying above filtering steps, we removed ~44% of the peaks as potential false positives and end up with a union of 491,818 open chromatin regions.

We modified a paragraph in the main text as below:

“To identify the cCREs in each of the 160 A-types defined from chromatin landscapes, we aggregated the snATAC-seq profiles from the nuclei comprising each cell cluster and determined the open chromatin regions with MACS2 and filtering steps (Extended Data Figure 13a). We then selected the genomic regions mapped as accessible chromatin in both biological replicates, finding an average of 93,775 (range from 50,977 to 136,962) sites (500-bp in length) in each sub-type (Extended Data Figure 13b). To account for different performance due to read depth and/or number of nuclei in individual clusters, we filtered reproducible peaks based on a corrected MACS2 score (Extended Data Figure 13c, see Methods). We further selected the elements that were identified as open chromatin in a significant fraction of the cells in each sub-type (FDR >0.01, zero-inflated Beta model, Extended Data Figure 13d, see Methods), resulting in a union of 491,818 open chromatin regions. These cCREs occupied 14.8% of the mouse genome (Supplementary Table 6 and 7).”

And, we provided more detailed descriptions in the method section.

Figure R3: Statistic for peak calling. **a** Schematic diagram of peak calling and filtering pipeline. **b** Number of peaks retained in each peak calling and filtering steps. **c** Scatter plots show the relationship between number of nuclei in cluster and number of peaks before score-per-million correction (SPM, left panel) and after SPM correction (right). **d** Distribution of fraction of accessibility in peaks for each cell type and its corresponding background. For each cell type, the background is defined as same number of non-DHS and non-peak regions randomly picked from genome.

The authors should provide more quantitative instead of qualitative statements.

Response:

We have revised the main texts throughout to provide more quantitative statement wherever possible. In particular, in Extended Data Figure 2 (Fig R4 in this response letter), we showed that snATAC-seq libraries are of good quality, on average more than 50 million fragments are kept for downstream analysis, on average 14 thousand barcodes are detected for each library. The distributions of fragment size show a clear periodic pattern resulting from nucleosome protection, which is a hallmark of high-quality ATAC-seq libraries. Excellent correlation between biological replicates and similar brain regions also indicates high reproducibility and robustness of the experiments.

We have added sentences in the main text as follows:

“The libraries were sequenced, and the reads were deconvoluted based on nucleus-specific barcode combinations and sequencing reads showed nucleosome-like periodicity (Extended Data Figure 2a-e). Excellent correlation between datasets from similar brain regions (0.92-0.99 for isocortex; 0.89-0.98 for OLF; 0.79-0.98 for CNU; 0.88-0.98 for hippocampus) and between

biological replicates (0.98 in median, range from 0.95 to 0.99) indicated high reproducibility and robustness of the experiments (Extended Data Figure 2f).”

Figure R4: Quality metrics of snATAC-seq libraries. **a** Mapping ratio for each sample. **b** Number of proper read pairs for each sample. **c** Number of unique fragments for each sample. **d** Fragment size distribution of each snATAC-seq library. **e** Number of unique barcodes captured by each snATAC-seq. **f** Spearman correlation matrix of snATAC-seq libraries. The box is drawn from lower quartile (Q1) to upper quartile (Q3) with a horizontal line drawn in the middle to denote the median, and an average marker, whiskers with maximum 1.5 IQR.

In Extended Data Figure 3 (Fig R5 in this response letter), we show the metrics and criteria used for quality control of nuclei. We selected nuclei with at least 1,000 fragments that displayed high enrichment in the annotated transcriptional start sites (TSSe > 10). Potential barcode collision or doublets were examined and removed for each sample set. Altogether, 28.7% nuclei of low-quality and 6.2% potential doublets were removed from further analysis.

Figure R5: Quality control of snATAC-seq datasets. **a** Dot-plot illustrating fragments per nucleus and individual TSS (transcriptional start site) enrichment. Nuclei in the upper right quadrant were selected for analysis. **b** Fraction of cell collision from human-mouse mix experiment. Inside panel shows fraction of potential barcode collisions detected in snATAC-seq libraries using a modified version of Scrublet¹; the box is drawn from lower quartile (Q1) to upper quartile (Q3) with a horizontal line drawn in the middle to denote the median, whiskers with maximum 1.5 IQR). **c** Number of nuclei retained after each step of quality control. **d** Number of nuclei passing quality control for sub-regions. **e** Distribution of TSS enrichment (the box is drawn from lower quartile (Q1) to upper quartile (Q3) with a horizontal line drawn in the middle to denote the median, whiskers with maximum 1.5 IQR) and **f** number of uniquely mapped fragments/nucleus for individual libraries.

Major point 2

- There is a very stark separation for the neuronal cell types based on dissection region (Fig. 1C). Can it be excluded that this is some way due to a batch effect?

Response:

We thank the reviewer for raising this important point. The stark separation based on dissection region was evident for a few major types like granular cell (DGGR) from hippocampus, which is restricted to the dentate gyrus (DG), and similarly for CA1GL and CA3GL in cornu ammonis field (CA) and three neuron types (PIRGL, OLFGL, and OBGL) in the olfactory area. The intratelenchephalic (IT) excitatory neurons from different cortical layers also show spatial gradients, which is consistent with results from DNA methylation analysis (companion paper, Liu, Zhou et al. bioRxiv. 2020²).

We used two approaches to assess the potential batch effects between biological replicates and found that they are negligible. First, we employed the k-Nearest neighbor batch-effect test (kBET)³, which measures batch mixing at the local level (Figure R6a). We calculated kBET acceptance rate for each major type. An acceptance rate close to 1 suggests that the clustering result is likely not due to a batch effect. The observed acceptance rate for major types was close to 1 and comparable to the expected background (Figure R6b). Second, we used a metric, the Local Inverse Simpson's Index (LISI) proposed by Korsunsky et al.⁴, to assess potential confounding factors due to batch and cell type mixing. In the case of LISI integration (iLISI) to measure batch mixing, the index is computed for batch labels, and a score close to the expected number of batches denotes good mixing (Figure R6c). In our case, we detected for cell types an iLISI score close to 2 which denotes good mixing of batches/replicates (Figure R6d).

Figure R6: Statistic for each cell type.

a Schematic diagram of k-nearest neighbor batch effect test (kBET). b Acceptance rate from kBET test for each major type. c Schematic diagram of local inverse Simpson's index (LISI). d Distribution of LISI scores for cells in major cell types.

We added a sentence in the main text:

“The clustering result was robust to variations of sequencing depth, signal-to-noise ratios between brain regions and replicates, demonstrated using both the K-nearest neighbor batch effect test (kBET) and local inverse Simpson's index (LISI) analysis (Extended Data Figure 5).”
And, we provided batch effect test in Extended Data Figure 5.

It is not clear from the methods how the samples were processed: were different regions processed at the same time and were replicates processed at different times?

Response:

Libraries for replicates were generated on different days from different pools of mice. We have added this information in the sample metadata table in Supplementary Table 1, Column D. Initially we generated a library for one region per day. Starting on 190207 (YY/MM/DD) we generated libraries for two samples in parallel. In these instances, we processed the first replicate for two regions together on one day and the libraries for the second replicate on another day, for example, libraries for the first replicate for regions 11E and 11F were generated on 190214 (CEMBA190214_11E, CEMBA190214_11F) and the libraries for the second replicate were generated on 190305 (CEMBA190305_11E, CEMBA190305_11F).

It just says that 8 96-well plates were processed (15k nuclei at 100% recovery) – does this mean 8 96-well plates per region and replicate?

Response:

Yes, for each region and replicate, 20 nuclei were sorted per well of 8 96 well plates (768 wells, 15,360 nuclei total). If processing two samples per day, tagmentation was performed with different sets of barcodes in separate 96 well plates. After tagmentation nuclei from individual plates were pooled together. From each of the two samples, 20 nuclei were sorted per well for a total of 40 nuclei into 8 96 well plates (768 wells, 30,720 nuclei total, 15,360 nuclei per sample). We added more detailed description in method.

Major point 3

- It is not clear how the cell types were annotated. If this was only done based on marker gene accessibility, which marker genes were used? Only the 14 genes in Fig. 1D? It seems hard to annotate 43 cell types based on accessibility at 14 genes.

Response:

Thank you for this comment! We first distinguished cell clusters into three major classes, non-neurons, GABAergic neurons, and Glutamatergic neurons, by checking chromatin accessibility at the promoter and gene body (gene activity) of 8 marker genes. We used *Snap25* to separate neurons with non-neurons (Extended Data Figure 6, here as Figure R7a). We also checked several well-known non-neuron marker genes to confirm the annotation, including *Pdgfra* for OPC, *Mog* for OGC, *ApoE* for astrocytes, *C1qb* for MGL, *Cdh5* for endothelial cells. *Slc17a7* and *Gad2* were used to further distinguish GABAergic and Glutamatergic neurons.

To annotate each major type, at least three previously reported marker genes were used. We showed marker genes accessibility in Extended Data Figure 6 (here as Figure R7b) and generated one Supplementary Table 4 to list 135 marker genes with the description of the cluster names and references to the marker gene information.

Figure R7: Marker genes used for cell type annotation.

a Marker genes used for annotation of major classes. **b** Marker genes used for major type annotation. See also Supplementary Table 4.

we added a sentence in the main text:

“..., which were annotated based on chromatin accessibility in promoters and gene bodies of at least 3 known marker genes.”

Major point 4

- How do the false positive peaks perform for the metrics shown in Figure 3A-D?

Response:

Thank you for this question! We performed the same analysis for different peak sets, including raw peak set, peak set after blacklist removal, peak set after naïve overlap, peak set after score-per-million (SPM) correction and filtering, and final peak set. Although the genomic distribution of different peak sets doesn't show much difference (Figure R8a), we found that the percentage of peaks from the final list showed the highest overlap with representative DNase hypersensitivity sites (rDHS), highlighting that multiple filtering steps help increase the fraction of reliable peaks (Figure R8b). Besides, the final peak set was more conserved (phastCons score) than other peak sets (Figure R8c) and showed higher enrichment for active *cis*-regulatory regions such as strong enhancers and active promoters (Figure R8d).

Figure R8: Statistic for peak calling. **a** Fraction of the different peak sets and identified cCREs (final set) that overlap with annotated transcriptional start sites (TSS), introns, exons, transcriptional termination sites (TTS), and intergenic regions in the mouse genome. **b** Venn diagram showing the overlap between different peak sets and DNase hypersensitive sites (DHS) from developmental and adult mouse tissue from the SCREEN database. **c** Box-Whisker plot showing that sequence conservation measured by PhastCons score in different peak sets, cCREs (final set), and random genomic sequences (the box is drawn from lower quartile (Q1) to upper quartile (Q3) with a horizontal line drawn in the middle to denote the median, whiskers with maximum 1.5 IQR). **d** Fold enrichment of different peak sets and cCREs (final set) with a 15-state ChromHMM model in the mouse brain chromatin.

These analyses are included in Extended Data Figure 13, and we modified the main text for this part, from line 281 to 293 in main text.

Minor point 1

- *There is no figure showing the subclustering results of the 160 sub cell types. Cell types can be subclustered into almost any number of subclusters, so without seeing the clustering results and how resolved they are, this number is pretty meaningless.*

Response:

Thank you for the suggestion! We now include two new figures to show the sub-clustering results of the 160 sub cell types (Extended Data Figure 4, 7).

The Extended Data Figure 4 (Figure R9) describes our workflow and provides more detailed statistics. We performed graph-based clustering in different resolutions to find more stable/optimized clustering results (Figure R9a). For sub-clustering for every major type, we examined in a total of 10 different resolutions from 0.1 to 1 (Figure R9b). We constructed a consensus matrix from 100 iterative runs of Leiden clustering with randomized starting seed s in different resolution, in which each element represents the fraction of observations two nuclei are part of the same cluster (Figure R9c). A perfectly stable matrix would consist entirely of zeros and ones, meaning that two nuclei either cluster together or not in every iteration. The relative stability of the consensus matrices can be used to infer the optimal resolution. Let M^s denote the $N \times N$ connectivity matrix resulting from applying Leiden algorithm to the dataset D^s with different seeds. The entries of M^s are defined as follows:

$$M^s(i, j) = f(x) = \begin{cases} 1, & \text{if single nucleus } i \text{ and } j \text{ belong to the same cluster} \\ 0, & \text{otherwise} \end{cases}$$

Let I^s be the $N \times N$ indicator matrix where the (i, j) -th entry is equal to 1 if nucleus i and j are in the same perturbed dataset D^s , and 0 otherwise. Then, the consensus matrix C is defined as the normalized sum of all connectivity matrices of all the perturbed D^s .

$$C(i, j) = \left(\frac{\sum_{s=1}^S M^s(i, j)}{\sum_{s=1}^S I^s(i, j)} \right)$$

The entry (i, j) in the consensus matrix is the number of times single nucleus i and j were clustered together divided by the total number of times they were selected together. The matrix is symmetric, and each element is defined within the range $[0, 1]$. We examined the cumulative distribution function (CDF) curve (Figure R9d) and calculated the proportion of ambiguous clustering (PAC) score to quantify stability at each resolution (Figure R9e). The resolution with a local minimum of the PAC scores denotes the parameters for the optimal clusters (Figure R9e,f). In the case these were multiple local minimal PACs, we picked the one with higher resolution. Another measurement is dispersion coefficient (DC), which reflects the dispersion (ranges from 0 to 1) of the consensus matrix M from the value 0.5. The closer to 1 is the DC, the more perfect is consensus matrix, and thus the more stable is the clustering (Figure R9e). In a perfect consensus matrix, all entries are 0 or 1, meaning that all connectivity matrices are identical. The DC is defined as:

$$\rho = \frac{1}{n^2} \sum_{i=1} \sum_{j=1} 4 * \left(M(i, j) - \frac{1}{2} \right)^2 .$$

Finally, for every cluster, we tested whether we could identify differential features compared to all other nuclei (background) and the nearest nuclei (local background) using the function 'findDAR'.

More detailed description can be found in section “clustering and cluster annotation” in method. To help the general audience better access these clusters, we provided UMAP embedding for both major and sub-types in a new Extended Data Figure 7 (here as Figure R10) and on our web portal (<http://catlas.org/mousebrain/>).

Figure R9: Strategy for cell clustering. a Schematic diagram of cell clustering pipeline. b UMAP embedding of representative cell clustering at different resolutions from 0.1 to 1.0 using Leiden algorithm. c. Consensus matrix from 100 iterative clustering runs at different resolution d. to measure the stability of clustering at different resolution, we plot the cumulative distribution function (CDF) of consensus matrices. In the CDF curve of a consensus matrix, the lower left portion represents sample pairs rarely clustered together, the upper right portion represents those almost always clustered together, whereas the middle segment represents those with ambiguous assignments in different clustering runs. e The proportion of ambiguous clustering (PAC) score measure quantifies middle segment for each resolution. It is defined as the fraction of cell pairs with consensus indices falling in the interval $(u_1, u_2) \in [0, 1]$ where u_1 is a value close to 0, and u_2 is a value close to 1 (in this study, $u_1=0.1$, and $u_2=0.9$). A low value of PAC indicates a flat middle segment, and a low rate of discordant assignments across permuted clustering runs. We infer the optimal cell clustering by the resolution having the lowest PAC. f Optimal cell clustering result for a representative major type.

Figure R10: UMAP embedding for major and sub types. **a** UMAP embedding of major types for non-neurons (left), GABAergic neurons (middle), and Glutamatergic neurons (right). The major types that can be subdivided into sub-types are displayed for non-neurons in **b**, GABAergic neurons in **c**, and glutamatergic neurons in **d**.

Minor point 2

- The colors in the top panels of Figure 2A-C are hard to distinguish

Response:

We deleted these panels since it was not legible in UMAP space. Instead, we provide Supplementary Table 5 for more details about the matching between cell types. Considering limited space left for new analysis, we decided to move other panels to Extended Data Figure 9.

Minor point 3

- In Figure 2E,F which cutoff was applied to count as overlap?

Response:

The cutoff for the overlap score was 0.5. We modified one sentence in the main text:

“By applying an overlap score cutoff at 0.5, for 155 of 160 types defined by snATAC-seq (A-Type), we could identify a corresponding cell cluster defined using scRNA-seq data (T-Type, Fig. 2a, b); conversely, for 84 out of 100 T-types we identified one, or in some cases more, corresponding A-types (Fig. 2a, c).”

Referee 2

The manuscript describes a snATAC-seq analysis of the mouse cortex and adjacent brain regions and integrate the analysis of these data with previously published scRNA-seq. The authors sequence data from more than 800,000 nuclei, probably a record in this type of experiment, and identify 160 different cell types. Analysis of accessible regions allows the characterization of almost half a million enhancers, some of which associate with human mental disorders.

The experiments described here represent a technical tour-de-force. The data is of high quality and the computational analyses are carefully done. However, there is not much in terms of mechanistic insights gained for the data. This manuscript will be a good resource for those interested in mouse neurobiology and using mouse models to study mental diseases. I leave it up to the editors to decide if this type of information is sufficiently significant for publication in Nature.

Response:

We thank the reviewer for the positive remarks regarding the impact of our manuscript and providing helpful comments to improve our study. In response to this and the other reviewer's concerns and suggestions, we performed substantial additional analyses to describe the regional specificity of different neuronal and glial cell types, characterize the transcriptional regulatory sequencing underlying the specific sub-types of neurons and glial cells, interpret the likely causal variants of several psychiatric disorders including schizophrenia, and dissect gene regulatory programs during adult neurogenesis and oligodendrocyte maturation. As outlined in the beginning of this response letter, these new findings have been added to the revised manuscript.

References:

- 1 Wolock, S. L., Lopez, R. & Klein, A. M. Scrublet: Computational Identification of Cell Doublets in Single-Cell Transcriptomic Data. *Cell Syst* **8**, 281-291 e289, doi:10.1016/j.cels.2018.11.005 (2019).
- 2 Liu, H. *et al.* DNA Methylation Atlas of the Mouse Brain at Single-Cell Resolution. *bioRxiv* (2020).
- 3 Buttner, M., Miao, Z., Wolf, F. A., Teichmann, S. A. & Theis, F. J. A test metric for assessing single-cell RNA-seq batch correction. *Nat Methods* **16**, 43-49, doi:10.1038/s41592-018-0254-1 (2019).
- 4 Korsunsky, I. *et al.* Fast, sensitive and accurate integration of single-cell data with Harmony. *Nat Methods* **16**, 1289-1296, doi:10.1038/s41592-019-0619-0 (2019).

Reviewer Reports on the First Revision:

Referees' comments:

Referee #1 (Remarks to the Author):

The authors have addressed our comments on adding more quantitative statements to our satisfaction.

To address our other major request of providing some biological insight that would distinguish this resource from previously published datasets on single-cell chromatin accessibility of adult mouse brain, the authors chose to analyze cell type compositions across different dissected regions and identify some region-specific cCREs.

One challenge with snATAC-seq and related single-cell combinatorial indexing-based methods is the very high sample loss during the library preparation procedure. This makes it challenging to draw accurate conclusions on the cell type proportions present in the input material. The authors provide no information on how reproducible their cell type proportion calculations are, e.g., do they hold true across biological replicates?

Unfortunately, some of the ISH data shown as validation is not very convincing (e.g. Olig2 expression in Fig. 3m is not limited to ASCN cell type in Palladium, but broadly present across the whole specimen, and it is not clear if this is because this gene is expressed in many different cell types or the ASCN is not predominantly present in the palladium). Since this is the main additional analysis the authors have added, it would be important that they validate a subset of their findings. Some options could be looking at ISH of highly subtype-specific marker gene combinations (as determined either from snATAC-seq or in the matched sc-RNA-seq data), to validate region-specific distribution of some cell (sub)types. Ideally this would be performed in a quantitative manner for all cell subtypes (e.g. correlation of subtype-specific marker gene combination distributions by ISH and relative contribution of subtypes to sub-regions as determined in Fig. 1f), rather than showing one or two hand-picked examples. Could ISH datasets such as from the Allen Brain atlas be leveraged to facilitate this?

Minor point:

This sentence is unclear to us: '13.5% of these CTCF-bound cCREs were in spatial proximity to the predicted target genes, evidenced by overlapping with chromatin contact loop anchors identified in the mouse hippocampus using single nucleus chromatin organization analysis'.

It is not clear to us how overlap with any loop anchor in the genome is related to target gene proximity.

Referee #2 (Remarks to the Author):

My original impression of the manuscript was that the work described and the computational analyses of the complex datasets were carefully done and interpreted and that the manuscript would be an important resource. My only concern was the lack of biological insights. The authors have now addressed this concern by revising the manuscript considerably and including additional analyses that strengthen the biological significance of the conclusions. The second reviewer of the manuscript had some insightful questions that the authors have also addressed, I think quite satisfactorily. I'm now comfortable recommending this manuscript for publication in Nature.

Author Rebuttals to First Revision:

We are very grateful for the constructive comments from the reviewers. While both reviewers were positive on the additional analyses that have strengthened the biological significance of the conclusions, they also commented on some important aspects of the previous submission, including concerns about potential sampling bias from combinatorial barcoding based snATAC-seq method, and lack of systematic validation regional specificity of the mouse brain cell types that we described. To address these concerns, we have carried out new analyses as described below. We also made corresponding revisions to the manuscript which are highlighted in Red font.

Point-to-point Responses

Referee 1

General Comments.

To address our other major request of providing some biological insight that would distinguish this resource from previously published datasets on single-cell chromatin accessibility of adult mouse brain, the authors chose to analyze cell type compositions across different dissected regions and identify some region-specific cCREs.

Response:

We would like to thank the reviewer for the helpful comments on the previous submission! Addressing these comments has led to significant improvement of our work.

Major points 1

One challenge with snATAC-seq and related single-cell combinatorial indexing-based methods is the very high sample loss during the library preparation procedure. This makes it challenging to draw accurate conclusions on the cell type proportions present in the input material. The authors provide no information on how reproducible their cell type proportion calculations are, e.g., do they hold true across biological replicates?

Response:

This is an important point, and we have taken several approaches to address it. Specifically, we looked closely at the reproducibility of cell type fractions estimated from snATAC-seq data by examining cells profiled from the two biological replicates. First, for each of the 43 major cell clusters, we found that the fraction of cells contributed by each biological replicate was approximately equal or close to 50% (0.51 ± 0.04 , see Figure R1a). Second, we also compared the cumulative distribution function (CDF) plot between two biological replicates, and performed a Kolmogorov-Smirnov (K-S) test. We did not observe a significant difference in the proportion of nuclei globally between biological replicates (Figure R1b). We observed the same trend across all major brain regions, sub-regions, and dissections (Figure R1c). Third, we analyzed the consistency in cellular composition between the two biological replicates of each of the 45 brain dissections profiled in the current study. We found an excellent Spearman

correlation coefficient (SPCC) between replicates (0.800-0.991, 0.954 on average), significantly higher than the SPCC calculated between replicates from different brain regions (intra- and inter- major regions, Figure R1d). Fourth, we also observed excellent SPCCs in regional specificities computed across the cerebrum between the two biological replicates, further underpinning the reproducibility of cell-type proportions (Figure R1e). We have added the information to the Extended Data Figure 6.

Figure R1: Cell-type proportions and regional specificity estimated from snATAC-seq data are consistent between the two biological replicates

a Bar plot shows the fraction of nuclei in biological replicates across 43 major cell types discerned from snATAC-seq data of the adult mouse cerebrum. **b** A cumulative distribution function (CDF) plot shows the fraction of nuclei in major types from two biological replicates. Kolmogorov-Smirnov test (K-S test) shows no significant difference in cell-type proportion between both biological replicates. **c** Boxplots of the p -values of K-S tests illustrate consistent results between biological replicates for each of the major brain regions, sub-regions, and brain dissections in the

present study. **d** A heatmap shows the pair-wise spearman correlation coefficients (SPCC) of cell type compositions of between two replicates of brain dissections. The column and row names are composed of two parts: brain dissection and replicate label. For example, MOp-1.1 represents the replicate 1 of the first brain dissection of the primary motor cortex (MOp-1). The embedded boxplot shows the distribution of SPCCs between two biological replicates, replicates from intra-major brain regions and inter-major brain regions. ***, p -value<0.001, Wilcoxon rank-sum test. **e** A scatter plot shows the cell-type regional specificity between two biological replicates calculated for each brain dissection.

Lastly, to address potential sampling bias and selective loss of cell populations in snATAC-seq using combinatorial barcoding (*sci*), we also performed snATAC-seq using a droplet-based platform from 10x Genomics (*10x*) for two biological replicates of the primary motor cortex (dissected region: 3C). We compared these data to the combinatorial barcoding datasets from the same dissection. The numbers of nuclei passing quality control for both methods were comparable (combinatorial barcoding: 15,939 nuclei, 10x: 16,314). Co-embedding of all datasets showed that the chromatin accessibility profiles and cell clusters from both platforms were in excellent agreement across cell types (Figure R2a-c). This was further shown by a confusion matrix comparing the similarity between clusters derived from the combinatorial barcoding and the 10X platform, respectively (Figure R2d). Further, we did not observe a significant difference in cell type composition between the two platforms (Figure R2e), with the exception of one small population of vascular cells (VLMC, 326 (*10x*), 155 (*sci*)).

Taken together these additional data demonstrate that cell type proportion measurements estimated based on clustering of snATAC-seq data were robust and reliable.

Figure R2: Comparison of cell-type compositions in the mouse primary motor cortex determined using single nucleus combinatorial indexing ATAC-seq and droplet based single nucleus ATAC-seq (10x Genomics) platforms.

a Individual clustering and of snATAC-seq data generated using the combinatorial barcoding (sci) and a commercial droplet-based platform (10x) for primary motor cortex (dissection: 3C). **b, c** Co-embedding and joint clustering of sci and 10x data shows excellent agreement between the two platforms. **b**, Dots are colored by cell cluster, **c**, Dots are colored by the experimental platforms. **d** Heatmap illustrating the overlap between cell cluster annotations from both platforms. rows: cell types from combinatorial barcoding; columns: cell types from the droplet-based platform. The overlap between the original clusters and the joint cluster was calculated (overlap score) and plotted on the heatmap. **d** Cumulative distribution function (CDF) plot showing the fraction of nuclei in individual cell types for each platform. **e** Bar plot showing contribution of nuclei from each platform to cell clusters.

Major points 2

Unfortunately, some of the ISH data shown as validation is not very convincing (e.g. Olig2 expression in Fig. 3m is not limited to ASCN cell type in Palladium, but broadly present across the whole specimen, and it is not clear if this is because this gene is expressed in many different cell types or the ASCN is not predominantly present in the palladium). Since this is the main additional analysis the authors have added, it would be important that they validate a subset of their findings.

Some options could be looking at ISH of highly subtype-specific marker gene combinations (as determined either from snATAC-seq or in the matched sc-RNA-seq data), to validate region-specific distribution of some cell (sub)types. Ideally this would be performed in a quantitative manner for all cell subtypes (e.g. correlation of subtype-specific marker gene combination distributions by ISH and relative contribution of subtypes to sub-regions as determined in Fig. 1f), rather than showing one or two hand-picked examples. Could ISH datasets such as from the Allen Brain atlas be leveraged to facilitate this?

Response:

This is also a very important point and we have performed extensive additional analysis to fully address this concern. Specifically, following this reviewer's suggestion, we have systematically verified the regional distribution of the identified snATAC-seq cell clusters in the mouse brain by comparing to the *in situ* hybridization (ISH) data of 505 cell-type specifically expressed genes across five major brain regions, namely isocortex, olfactory areas (OLF), hippocampal formation (HPF), striatum (STR), pallidum (PAL), using the ISH data downloaded from the Allen Brain Atlas.

To select the cell type specific genes, we first used the "Differential Search" function to select 10,269 genes from the Allen Brain Atlas that were preferentially expressed in these 5 brain structures compared to all brain regions (expression level > 1 and fold change > 1). Of these genes, 505 were also identified as cell-type-specific (out of 1,513) based on the single cell RNA-seq data in the joint RNA-ATAC clusters in the present study (fold change > 1 and FDR < 0.05, Extended Data Figure 10). Altogether, we defined between 1 and 53 genes for each cell type, with an average of 15 genes (Supplementary Table 6).

Next, we calculated the regional specificity of each cell type by evaluating the relative contribution from five brain regions (Figure R3a). To independently evaluate the specificity of

each cell type the ISH data, we assessed the variation in gene expression across the five brain regions, by calculating the coefficient of variation (CV) based on the average normalized ISH signals of cell-type specific marker genes for each cell type¹ (Figure R3b). We observed a general trend between the regional specificity score calculated from cell type composition and the coefficients of variation of marker gene expression from ISH (Figure R3c, Pearson correlation coefficients (PCC)=0.55). Consistent with previous findings, non-neuronal cell types generally showed low regional specificity in both ISH and snATAC-seq based analyses (Figure R3a, b). By contrast, most GABAergic and glutamatergic neurons showed moderate to high regional specificity based on both ISH based and snATAC-seq based analysis (Figure R3a, b). For example, cell-type-specific genes for granular cells (GRC) and cornu ammonis field 1 and 3 neurons (CA1GL and CA3GL) showed a high correlation with genes expressed predominantly in the hippocampal formation (HPF), cell-type-specific genes for intra-telencephalic (ITL23GL, ITL4GL, ITL5GL) neurons showed higher correlation with genes expressed in the isocortex and cell-type-specific genes for medium spiny neurons with genes expressed in the striatum (STR; Figure R1a-d). To assess the consistency for each marker gene, we calculated the Pearson correlation coefficients (PCC) between the distribution of that cell type across the five brain regions and the ISH signals for each of the marker genes in that cell type across the same brain regions (Figure R3d). Although not every marker gene shows consistent regional expression, this analysis result indicates a general concordance between ISH expression and regional specificity based on snATAC-seq in major cell types (Figure R3d). We note that the current ISH data still lack cellular resolution, and future assays using high throughput and super resolution imaging tools such as MERFISH or seqFISH would likely provide more informative comparisons.

We are providing this additional info (Figure R3), as part of the Extended Data Figure 13.

Figure R3: Correlation between regional specificity of cell clusters and regional expression of cell-type specific genes.

a Bar-plots show regional specificity based on relative contribution from different brain regions for cell types. Dot plot shows cell-type composition in five major brain structures, including isocortex, olfactory areas (OLF), hippocampal formation (HPF), striatum (STR), pallidum (PAL). The size of each dot reflects the relative contribution of a cell type in the brain region as indicated by the legend to the right of the panel, and the color of the dots indicates the brain region. **b** Bar-plots show the coefficients of variation (CV) based on the average expression of cell-type-specific marker genes from ISH in cell types. The heatmap shows the average expression levels of cell-type-specific marker genes in each cell type. For a full list of ISH expression in cell-type specific genes see Supplementary Table 6. **c** Scatter plot shows the correlation between the coefficients of variation of marker gene expression patterns by ISH and the regional specificity calculated based on snATAC-seq data. **d** Boxplots show the Pearson correlation coefficient (PCC) calculated between cell composition across brain regions based on snATAC-seq data and the spatial distribution patterns of cell-type-specific genes across the five main brain regions based on *in situ* hybridization (ISH) dataset for each major brain cell type.

We next examined the regional specificity of a few sub-types of glial cells. Since the subtypes identified in the present study were not resolved in scRNA-seq studies, we identified subtype-specific genes for astrocyte subtypes using primarily snATAC-seq data. Then, we calculated the fraction of overlap between spatially mapped ISH genes from different brain structures and subtype-specific genes. For sub-type ASCN, we found the highest overlap with genes predominantly expressed in the pallidum (PAL) (Figure R4a). We now provide ISH views for additional top subtype-specific genes, *Slc6a11*, *Itih3*, and *Agt* (Figure R4b-d). We previously found the highest chromatin accessibility at the *Olig2* locus in the ASCN subtype and thus used the name previously reported as *Olig2*-lineage derived mature astrocytes². We agree with the reviewer that the ISH illustrates that *Olig2* is present in the pallidum but is also expressed in other regions of the brain. Indeed, *Olig2* is expressed across the brain in several cell types including oligodendrocytes and oligodendrocytes precursor cells. Unfortunately, the ISH data do not reveal the cell-type responsible for this signal due to lack of cellular resolution. Here, spatial transcriptomics data would be crucial to identify and localize the cell-types expressing *Olig2* in the pallidum.

We are providing this additional info here as Figure R4, and added panel a, b to Figure 3 and panel c, d to Extended Data Figure 18.

Figure R4: Detailed statistics for cell types

a Heatmap shows the fraction of overlap between spatially mapped genes from in situ hybridization in different brain structures and specific genes in sub-types of astrocytes. The p-value was computed using the Fisher's exact test, ***, p-value<0.0001. Views of In-situ hybridization (ISH) experiment from Allen Brain Atlas (ABA, www.brain-map.org), showing the gene expressed of gene Slc6a11 in **b**, Itih3 in **c**, and Agt in **d** from pallidum (PAL). Bar-plot shows expression value in brain structures from ISH experiments. Images downloaded from ABA (www.brain-map.org).

Minor point:

This sentence is unclear to us: '13.5% of these CTCF-bound cCREs were in spatial proximity to the predicted target genes, evidenced by overlapping with chromatin contact loop anchors identified in the mouse hippocampus using single nucleus chromatin organization analysis'. It is not clear to us how overlap with any loop anchor in the genome is related to target gene proximity.

Response:

The folding of the chromatin fiber can bring distant functional elements, such as promoters and enhancers, into spatial proximity, which could be detected by chromosome capture assays such as Hi-C, and PLAC-seq. In the present study, we used the list of anchors of chromatin loops identified in different brain cell types in the mouse hippocampus (sn-m3C-seq; see companion paper ³ to infer the target promoters of CTCF-bound cCREs.

To make it more accessible, we have changed the sentence in the manuscript:

'Second, we found that 13.5% of these CTCF-bound cCREs were in spatial proximity with the predicted target gene promoters in neuronal cells in the mouse hippocampus, as evidenced by chromatin loops detected from single nucleus chromatin organization analysis ³.

Referee 2

General Comments.

My original impression of the manuscript was that the work described and the computational analyses of the complex datasets were carefully done and interpreted and that the manuscript would be an important resource. My only concern was the lack of biological insights. The authors have now addressed this concern by revising the manuscript considerably and including additional analyses that strengthen the biological significance of the conclusions. The second reviewer of the manuscript had some insightful questions that the authors have also addressed, I think quite satisfactorily. I'm now comfortable recommending this manuscript for publication in Nature.

Response:

We thank the reviewer for appreciating the impact of our manuscript and providing helpful comments that significantly improved our study.

References

- 1 Lein, E. S. *et al.* Genome-wide atlas of gene expression in the adult mouse brain. *Nature* **445**, 168-176, doi:10.1038/nature05453 (2007).
- 2 Tatsumi, K. *et al.* Olig2-Lineage Astrocytes: A Distinct Subtype of Astrocytes That Differs from GFAP Astrocytes. *Front Neuroanat* **12**, 8, doi:10.3389/fnana.2018.00008 (2018).
- 3 Liu, H. *et al.* DNA Methylation Atlas of the Mouse Brain at Single-Cell Resolution. *bioRxiv*, doi:10.1101/2020.04.30.069377 (2020).

Reviewer Reports on the Second Revision:

Referees' comments:

Referee #1 (Remarks to the Author):

The authors have adequately addressed our comments. Although we don't need to re-review it, we would ask that the authors include the 10X/sn comparisons in the manuscript or supplement. These will be very useful for the field.

Author Rebuttals to Second Revision:

N/A